# Chromatin spatial analysis by METALoci unveils sex-determining 3D regulatory hubs

Mammalian sex is determined by opposing networks of ovarian and testicular genes that are well characterized; however, its epigenetic regulation is still largely unknown. Here we explore the 3D chromatin landscape of sex determination in vivo by profiling fluorescence-activated cell-sorted embryonic mouse gonadal populations in both sexes before and after sex determination. Through conventional Hi-C analyses, we show that chromatin structures, particularly topologically associating domains, remain largely unchanged during sex determination, suggesting a preformed configuration. We further integrate Hi-C data with ChIP-seq experiments using METALoci, a spatial autocorrelation analysis that identifies three-dimensional (3D) regulatory hubs across the genome. We uncover a prominent rewiring of chromatin interactions during sex determination, affecting the 3D regulatory hubs of hundreds of genes that display time-specific and sex-specific expression. By combining predictive approaches and validations in transgenic mice, we identify a 3D regulatory hub for the protesticular gene *Fgf9*. The deletion of this gonad-specific hub allows mutant mice to survive through development, overcoming lung lethality associated with *Fgf9* loss of function while exhibiting male-to-female sex reversal. Through the reconstruction of gene regulatory networks, we identify a function for *Meis* genes, which act redundantly to specify sexual identity during ovarian and testicular development. Our results underscore the dynamic role of the 3D genome during sex determination, highlighting the potential of epigenomic approaches to uncover regulators of developmental processes.

Reproduction is a fundamental aspect of life that depends on the differentiation of compatible sexes. In mammals, sex is determined by a balanced network of ovarian-promoting and testicular-promoting factors[1]. Before sex determination, gonads from both sexes are bipotential, as they can either develop as ovaries or testes. In XY individuals, the gene encoding sex-determining region Y protein (*Sry*) tilts this balance; its expression in the supporting lineage results in the activation of its direct downstream target, the protesticular gene encoding SRY-box transcription factor (TF) 9 (*Sox9*)[2–6]. SOX9 interacts with the fibroblast growth factor 9 (FGF9) morphogen to propagate the male-determining signal to the entire gonad, suppressing ovarian-specific genes and promoting testicular development. In XX individuals, which lack *Sry*, ovarian development is actively driven by the expression of the ovarian-determining factor Wilms tumor suppressor (WT1)-KTS isoform, as well as of several members of the Wnt pathway, such as R-spondin 1, Wnt family member 4 or β-catenin, and TFs such as forkhead box L2 (FOXL2) or runt-related TF 1 (RUNX1)[7–12]. Subsequently, sex-determining signals induce changes on cell differentiation, hormone synthesis and, ultimately, a physical and behavioral transformation of the entire organism. Decades of research have revealed several genes associated with sex determination, yet its epigenetic regulation remains largely unknown.

✉e-mail: blanche.capel@duke.edu; martirenom@cnag.eu; dario.lupianez@csic.es

In vertebrates, gene expression is controlled by *cis*-regulatory elements (CREs), which serve as binding platforms for TFs[13]. However, to exert their function, CREs may move into physical proximity with their target genes, a process mediated by the three-dimensional (3D) folding of the chromatin. Chromosome conformation capture methods, particularly Hi-C[14], revealed that vertebrate genomes fold into distinct levels of organization[15–17]. At the megabase scale, genomes segregate into active (A) and inactive (B) compartments, reflecting the clustering of loci according to epigenetic state. At the submegabase scale, genomes organize into topologically associating domains (TADs), which represent large genomic regions with increased interaction frequencies. At a lower scale, CRE and promoters interact, resulting in gene expression patterns with marked cell type specificity. Emerging evidence shows that CRE–promoter interactions occur in the context of highly connected 3D regulatory hubs[18], whose nature remains an active area of research.

Noncoding mutations affecting either CRE function or 3D chromatin organization can cause disease[19–21] or drive species adaptation[22–24]. Noncoding mutations also have been associated with variations in sex determination, including duplications or deletions at the *SOX9* locus in persons with sex reversal[25] or an inversion at the *FGF9* TAD associated with ovotesticular development in female moles[22]. Recent studies started to explore the epigenetic regulation of sex determination globally[26–29], yet the lack of information on 3D chromatin organization has limited progress. This impacts our capacity to genetically diagnose differences in sex development (DSD), a group of conditions that alter reproductive capacities in humans[30].

Here, we explore the 3D regulatory landscape of mammalian sex determination in vivo by generating high-resolution chromatin interaction maps of the mouse gonadal supporting lineage, before and after sex determination, in both sexes. Conventional Hi-C analysis revealed that chromatin structures, particularly TADs, remain largely invariable during sex determination. We integrate Hi-C and epigenetic data using METALoci, a spatial autocorrelation framework that quantifies regulatory environments across the genome in an unbiased manner. These analyses revealed prominent changes in 3D regulatory hubs with sex and temporal specificity. We use METALoci as a predictive tool and validate it at the *Fgf9* locus by identifying a noncoding regulatory region, deletion of which led to male-to-female sex reversal in transgenic mouse models. This deletion uncoupled the perinatal lethality associated with *Fgf9* loss of function in the lung, thus allowing the study of gonadal phenotypes postnatally. Lastly, we reconstruct gene regulatory networks and identify a role for *Meis* genes during sex determination. We demonstrate in transgenic mice that, during both ovarian and testicular development, *Meis1* and *Meis2* act redundantly and are essential to specify sexual identity. Our results highlight the important role of 3D chromatin and epigenetic regulation in sex determination, a process of critical relevance for species reproduction.

## Results

### Changes in 3D chromatin structures are moderate during sex determination

We explored the 3D regulatory landscape of sex determination in the gonadal supporting lineage of both sexes, before and after commitment to female or male fate (XX E10.5, XX E13.5, XY E10.5 and XY E13.5; Fig. 1a). We used mouse lines expressing cell-specific markers to isolate gonadal populations by fluorescence-activated cell sorting (FACS). Progenitor supporting cells were isolated from both sexes at E10.5, using an *Sf1–eGFP* line[31]. At this stage, before *Sry* expression, *Sf1–eGFP* cells are bipotential. At E13.5, a *Sox9–eGFP* line was used to isolate Sertoli cells from developing testes[31], while a *Runx1–eGFP* line was used to obtain their ovarian counterparts, the granulosa cells[11].

The 3D chromatin interactions from isolated cell populations were profiled at high resolution, using a low-input Hi-C protocol[32] and generating between 750 and 950 million valid pairs per sample

(Supplementary Table 1). We subsequently sought to identify changes in 3D chromatin organization during sex determination. First, we identified compartments at 100-kb resolution and observed high correlation between biological replicates (Fig. 1b and Extended Data Fig. 1a). As expected, A compartments were enriched in H3K27ac, open chromatin and increased gene density, in contrast to B compartments (Extended Data Fig. 1b). Across samples, the genomic proportion assigned to A compartments fluctuated between 42–50% and 50–58% for B compartments (Extended Data Fig. 1c). Yet, dissimilarity index analyses revealed increased compartment correlation between male and female cells before sex determination (Fig. 1c and Extended Data Fig. 1d), reflecting higher similarities at the bipotential stage. Correlations decreased after sex was determined and cells progressed toward the granulosa or Sertoli cell fate. Comparisons to public Hi-C datasets[33] denoted higher similarities between gonadal cell types than against other cell types, such as mouse embryonic stem cells (mES cells) or neural cells (Extended Data Fig. 1e). We next identified compartment switches by performing pairwise comparisons (Extended Data Fig. 2a–c). We observed that compartment switches involved 7.4–8.9% of the genome, correlating well with expected changes in gene expression (that is, A to B: decreased expression, B to A: increased expression; Fig. 1d and Extended Data Fig. 2c). An exception was observed in the comparison between female and male bipotential stages, where only 5.3% of the genome varied (Extended Data Fig. 2c). As *Sry* is still not active at this developmental timepoint, sexual dimorphism in compartments may be induced by the different sex–chromosome complements[34]. Interestingly, early sex-specific variation in compartments was not associated with transcriptional changes (Fig. 1d). This may suggest that variations in 3D chromatin organization might precede changes in gene expression, as for other differentiation processes[35]. Next, we focused on changes at the TAD level. We calculated insulation scores for each sample, identifying 6,179 TAD boundaries (Fig. 1e). Metaplot analysis revealed that insulation increased during the transition from bipotential to differentiated stages (Fig. 1f), as described in other biological systems[33]. Yet, this increase in insulation was of low magnitude for most regions, as pairwise comparisons revealed that only 1.49–1.84% of TAD boundaries changed insulation significantly between sexes or time points (Fig. 1g). Moreover, manual inspection revealed that most changes resulted from quantitative variation in insulation rather than de novo formation or disappearance of TAD boundaries (Extended Data Fig. 2d).

In summary, our analyses revealed moderate variation in 3D chromatin organization during sex determination, reflected by high degree of conservation in TAD structures and compartment changes that increased through differentiation.

### Rewiring of 3D regulatory hubs in a time-specific and sex-specific fashion

Our analyses of TAD dynamics suggest that, at a large scale, 3D chromatin organization is formed before sex determination. Yet, compartment analyses suggest that some genomic regions undergo epigenetic changes during this process, potentially reflecting regulatory variation at other scales. However, such changes are difficult to detect with conventional Hi-C data analysis, as those usually focus on predefined chromatin structures such as compartments, TADs or loops and may overlook other types of changes relevant for gene regulation. To address these limitations, we developed METALoci, an unbiased approach to determine regulatory activity locus by locus without prior data assumptions.

METALoci relies on spatial autocorrelation analysis, classically used in geostatistics[36,37] to describe how the variation of a variable depends on space at global and local scales (for example, identifying contamination hotspots within a city[38]). We repurposed this analysis to infer gene regulatory activity, as CREs and genes cluster in the 3D nuclear space while displaying similar epigenetic properties. Similarly

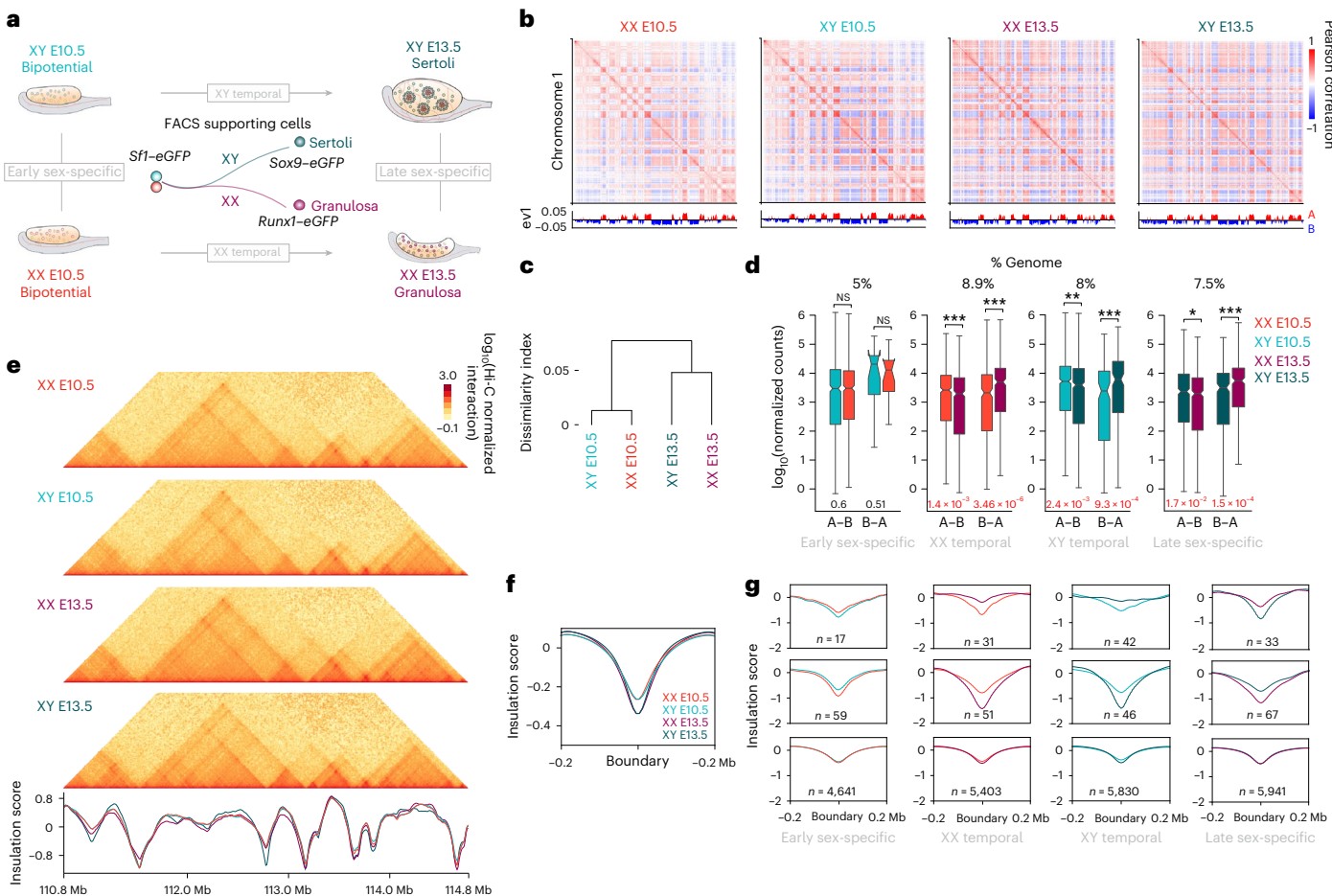

**Fig. 1 | Moderate changes in compartment and TAD organization during sex determination. a**, Experimental setup of FACS-sorted gonadal populations. **b**, Compartment analyses showing eigenvectors for chromosome 1 in different samples. Eigenvectors are computed from the matrices of the physical interaction between pairs of loci across the chromosome. **c**, Dissimilarity index in A/B compartments between different samples. **d**, Gene expression levels for genes that switched compartments. Note that significant changes in gene expression occur in all comparisons, except in XY E10.5 versus XX E10.5. Expression is shown as DESeq2-normalized counts. Differential expression was tested using a two-sided Mann–Whitney *U*-test. *P* values are shown below each box plot; red indicates significance (adjusted *P* < 0.05). NS, not significant. Box plots show the median, interquartile range (IQR; 25th–75th percentiles) and whiskers (±1.5× the IQR). Gene counts: early sex-specific A–B/B–A = 522/13, XX temporal = 751/304, XY temporal = 747/129, late sex-specific = 557/266. **e**, Top: Hi-C maps of the *Sox9* locus in different samples. Bottom: insulation scores for the same genomic region. **f**, Insulation score at TAD boundaries. Note the increase in insulation during the transition from bipotential to Sertoli or granulosa cells. **g**, Pairwise comparison of insulation scores at boundaries. The top two rows correspond to boundaries that change insulation between samples, while the bottom row depicts stable boundaries. Note that stable boundaries are more abundant in any comparison.

to METALoci, other methods have used graph theory to link spatial measurements (such as 3C-based approaches) with linear genomic measurements (such as ChIP-seq), including Canvas[39] and ChAseR[40]. In these approaches, networks are used to analyze Hi-C data by representing chromatin fragments as nodes and interactions as edges, constructing genome-wide networks on the basis of significant interactions. METALoci differs from these approaches in several ways. First, its primary aim is to dissect functional correlations relevant to the resident genes within a genomic domain rather than identifying correlated signals genome-wide. Second, METALoci is unbiased, as it does not rely on identifying significant interactions or known genomic features such as compartments, TADs or loops. Third, METALoci produces easily interpretable results at each locus, visually and qualitatively guiding biologists in generating insights and formulating new testable hypotheses. Fourth, METALoci is computationally fast, conceptually straightforward and easily explainable.

METALoci consists of four steps (Fig. 2a). First, a genome-wide Hi-C normalized matrix is taken as input and top interactions selected (Fig. 2b). Second, selected interactions are used to build a graph layout (equivalent to a physical map) using the Kamada–Kawai algorithm[41] with

nodes representing bins in the Hi-C matrix and the two-dimensional (2D) distance between the nodes being inversely proportional to normalized Hi-C interaction frequencies (Fig. 2b). Third, epigenetic and genomic signals, measured as coverage per genomic bin (for example, H3K27Ac ChIP-seq signal), are mapped into the graph layout nodes (Fig. 2c). Fourth, a measure of autocorrelation (specifically, the local Moran's I (LMI)[36,37]) is used to identify nodes and their neighborhoods (other genomic bins within a specified 2D distance in the graph layout) with similar epigenetic and genomic signals (Fig. 2c). METALoci categorizes each genomic bin according to its signal status and of its surrounding neighborhood (Fig. 2d). Specifically, a genomic bin categorized as high–high (HH) is enriched for signal but also other bins in spatial proximity (Fig. 2d). In contrast, bins marked as low–low (LL) represent those depleted of signal in both the corresponding bin and its spatial neighborhood. High–low (HL) and low–high (LH) represent bins that are enriched in signal but not their neighborhood and vice versa. Lastly, the group of genomic bins that are spatially contiguous and statistically significant for the spatial signal enrichment (that is, those classified as significant HH bins, including their direct neighbors in the graph layout; Fig. 2d, red) are named 'metaloci'. Importantly, METALoci quantifies the

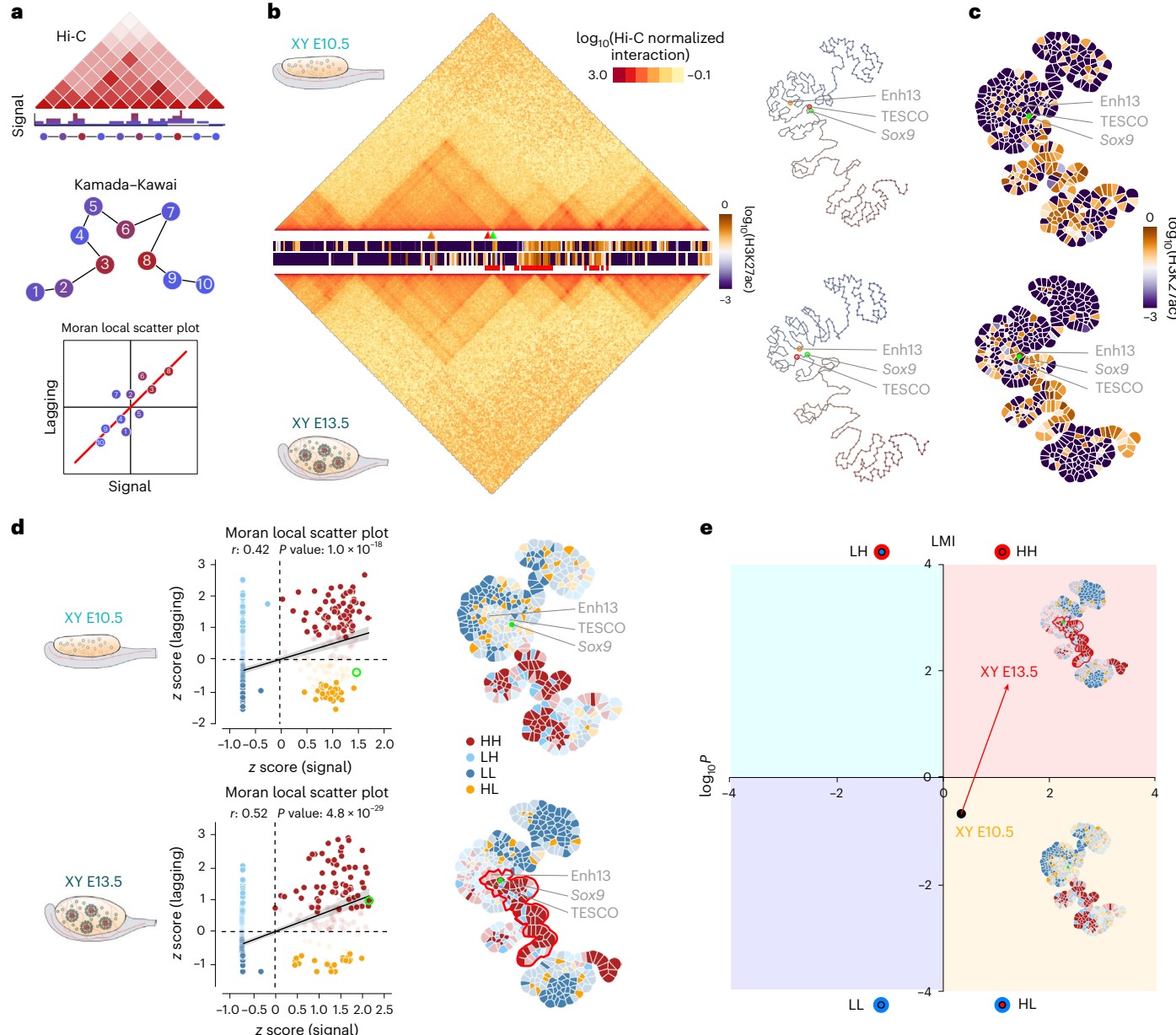

**Fig. 2 | Quantification of regulatory activity at individual loci using METALoci.**
**a**, Schematic METALoci pipeline (Methods). **b**, Left: Hi-C data of the Sox9 locus centered at chr11:110,780,000–114,770,000 coordinates and for XY E10.5 and XY E13.5 cells. H3K27ac ChIP-seq tracks are displayed between Hi-C maps, displaying signal intensity in yellow or blue color code (the top track corresponds to XY 10.5 and the bottom track corresponds to XY 13.5). The positions of the *Sox9* promoter (green arrowhead), Enh13 (orange arrowhead) and TESCO (red arrowhead) are highlighted. Squared red marks under H3K27ac track indicate the noncontinuous metaloci detected for the *Sox9* locus at XY E13.5. Right: Taking as input Hi-C data, a 2D layout is generated using the Kamada–Kawai algorithm. The layout highlights the *Sox9* locus (green circle) and the Enh13 (orange circle) and the TESCO (red circle) enhancers. **c**, H3K27ac signal is mapped into the graph layout and represented as a Gaudí plot (Methods). **d**, Left: LMI scatter plot where each point representing a node in the graph layout is placed within the four quadrants of the LMI (that is, HH, LH, LL and HL). Points with solid color are statistically significant (*P* < 0.05). The LMI scatter plot can be interpreted as the correlation between local signal at the node (*x* axis) of interest and that of the neighborhood

of the node (*y* axis). For example, if a genomic bin is high in H3K27ac and its neighborhood is also high, the node is placed in the HH quadrant and colored red. The point of the node containing the *Sox9* locus is highlighted with a green circle. Right: Gaudí plot highlighting in space the classification of each bin into the LMI quadrants with solid color indicating statistical significance (*P* < 0.05). *Sox9* HH metaloci outlined in red. For the regression plot, significance was assessed using a parametric regression test (*t*-test on the regression coefficient). For the LMI in the Gaudí plot, significance was assessed using a Monte Carlo permutation test of spatial randomness. **e**, LMI transition (gene transition) for the *Sox9* locus from a nonsignificant HL to a significant HH enhancer hub during the differentiation of Sertoli cells (XY E10.5 to XY E13.5). A gene transition is the length (in arbitrary units) of the vector connecting the LMI and *P* value coordinates between two time points. In the example, the *Sox9* gene transition (red arrow) was 2.29. A positive gene transition indicates that the resulting vector points toward the HH quadrant. A negative gene transition indicates that the vector points toward the LL quadrant. For the LMI in the Gaudí plot, significance was assessed using a Monte Carlo permutation test of spatial randomness.

spatial autocorrelation of the input signal for each genomic bin, which facilitates direct comparisons between datasets.

For each gene of the mouse genome, we applied METALoci to reconstruct its 3D regulatory hubs (named metaloci) during sex

determination. With that purpose, we integrated Hi-C datasets with H3K27ac ChIP-seq signal[27], marking active promoters and enhancers. Thus, HH metaloci for H3K27ac can be considered 3D regulatory hubs activating the expression of resident genes. To evaluate the accuracy

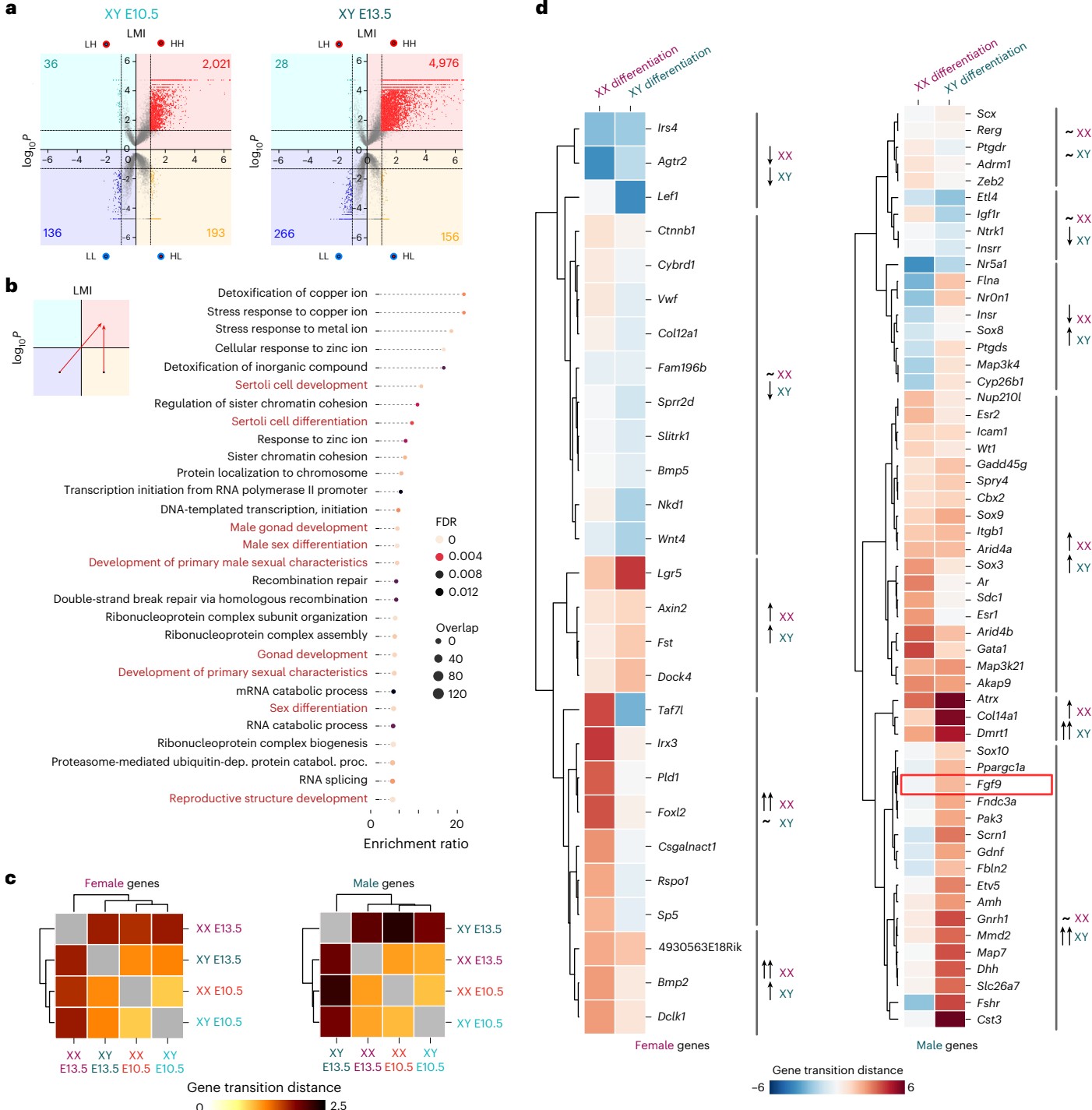

**Fig. 3 | METALoci captures extensive rewiring of 3D enhancer hubs during sex differentiation. a**, LMI quadrants for XY E10.5 (left) and XY E13.5 cells (right) for 24,027 annotated gene promoters in the mm10 reference genome. Quadrants include the total number of statistically significant genes in each quadrant. Note the increased numbers after differentiation in the HH and LL, denoting simultaneous activation and repression of genes during differentiation. *P* values for LMI were assessed using a Monte Carlo permutation test of spatial randomness. **b**, GO biological process enrichment analysis for genes that transition from LL or HL to HH during Sertoli cell differentiation. GO term enrichment significance was assessed using a hypergeometric test, with multiple testing corrections using Benjamini–Hochberg FDR. dep. dependent; catabol., catabolic; proc., process. **c**, Mean absolute gene transition for genes acquiring female-specific and male-specific expression during sex differentiation[43]. The gene transitions are larger for 'male' genes in XY cells upon differentiation compared to 'female' genes in XX cells. **d**, Individual gene transitions for each female-specific and male-specific genes during sex determination. Genes are grouped by unsupervised clustering based on gene transitions from E10.5 to E13.5 in female and male differentiation. The red rectangle highlights *Fgf9*, a male gene that gains prominent regulatory activity during XY differentiation and whose 3D regulatory hub is subsequently validated (Fig. 4).

of METALoci in detecting regulatory hubs, we first analyzed the *Sox9* locus, which is directly activated by SRY and essential to trigger Sertoli cell differentiation[5]. METALoci recapitulated known changes in *Sox9* regulation (Fig. 2b–d). At XY E10.5, the *Sox9* promoter remained in an inactive status and not associated with a HH metaloci. At XY E13.5, when Sertoli cells differentiate, the *Sox9* gene changed its regulatory status to HH, with both its promoter and environment enriched for H3K27ac signal (Fig. 2d). These changes parallel the transcriptional dynamics of

*Sox9*, which is expressed at low levels at the bipotential stage in both sexes but subsequently activated in Sertoli cells[31]. Our analysis also captured dynamic interactions of the *Sox9* promoter with two known enhancers, TESCO[42] and Enh13 (ref. 5) (Extended Data Fig. 3). Across the four quadrants of METALoci analysis, *Sox9* showed an HL-to-HH transition (here called 'gene transition'; Methods) in XY cells between E10.5 and E13.5, consistent with its known activation (Fig. 2e). Conversely, we captured regulatory changes associated with female differentiation. Specifically, the *Bmp2* promoter, classified as LL at early stages, gained an active environment in granulosa cells (HH). This involved a large contiguous patch of H3K27ac enrichment from two spatial proximal metaloci, including a known enhancer[27] (Extended Data Fig. 4).

We next explored genome-wide changes in 3D regulatory hubs during sex determination. We observed that, independent of sex, the number of genes categorized as HH doubled through differentiation, reflecting the activation of transcriptional programs (Fig. 3a and Extended Data Fig. 5a). Concomitantly, genes categorized as LL also doubled, suggesting the repression of additional pathways. Functional enrichment analyses revealed that genes transitioning from HL or LL toward HH during Sertoli cell differentiation (XY E10.5 to XY E13.5) were associated with relevant terms for this biological process (Fig. 3b). Specifically, we observed an overrepresentation (false discovery rate (FDR) < 0.01) of biological terms related to Sertoli cell differentiation, male sex development and RNA processing. Despite RNA processing signatures, an enrichment in sex-specific processes was not observed during female differentiation (XX E10.5 to XX E13.5) (Extended Data Fig. 5b), which could reflect the molecular similarity between granulosa cells and bipotential progenitors at this developmental stage[31].

To further assess METALoci's capacity to identify known sex determination biology, we curated from the literature[43] a list of 27 and 55 genes associated with female and male gonad development, respectively. The analysis of these genes with dimorphic expression revealed regulatory changes upon differentiation in both sexes. During the transition from bipotential to granulosa cells, many female-specific genes acquired an active regulatory environment, while their male-specific counterparts either lost it or displayed minor changes (Extended Data Fig. 5c). The opposite trend was observed during Sertoli cell differentiation, with the acquisition of active regulatory activity for many male-specific genes and a general loss in their female-specific counterparts (Extended Data Fig. 5c). Further analysis revealed that the magnitude of gene transitions, measured as mean absolute length of the vectors connecting gene states at E10.5 and E13.5 (Extended Data Fig. 5c), was moderate for sex-specific genes at early stages but increased with differentiation (Fig. 3c). Interestingly, changes in the regulatory environment were more prominent for male-specific than female-specific genes during the differentiation into the corresponding sex. A detailed comparison of individual gene transitions revealed distinct mechanisms governing how sex-specific genes acquire dimorphic expression patterns (Fig. 3d). Most male genes (46 of 55) gained an active regulatory environment during Sertoli cell differentiation. In contrast, this mechanism was not as common for female-specific genes during granulosa cell differentiation (14 of 27). Interestingly, the loss in regulatory activity for genes upregulated in the opposite sex was higher during male differentiation (13 of 27) than during female differentiation (8 of 55). These results suggest that, at this stage, the male differentiation program is sustained by more pronounced regulatory changes than the female program.

Overall, our METALoci analysis revealed a prominent rewiring of 3D regulatory hubs during sex determination, remodeling the chromatin landscape of hundreds of genes (Supplementary Data 1).

## In silico perturbations analyses reveal a noncoding region downstream of the *Fgf9* gene associated with male-to-female sex reversal

Next, we explored whether METALoci could be used to identify critical regulatory regions by taking the Sox9 locus as a control test, given its extensive characterization during sex determination. We computationally scanned the entire locus and estimated the effect of 50-kb deletions in the 3D regulatory hubs analyzed by METALoci (Extended Data Fig. 3 and Methods). Specifically, we assessed whether deleting five bins of 10 kb at a time in a moving window of one-bin steps over the studied region would disrupt the *Sox9* HH metaloci in XY E13.5. To do so, we computationally removed the selected bins, their Hi-C interactions and H3K27ac signal and recomputed the *Sox9* metaloci. If the resulting metaloci was perturbed by the simulated deletion, we considered that the bins removed were important for *Sox9* metaloci. *Sox9* is flanked by an upstream >1-Mb gene desert enriched in dozens of ATAC-seq and H3K27Ac ChIP-seq peaks that have been individually studied for regulatory activity. Importantly, our in silico perturbation analysis marked as relevant the only two elements demonstrated to be functional during gonadal development, the Enh13 and TESCO enhancers[5,42] (Extended Data Fig. 3).

Next, we followed the same approach to investigate *Fgf9*, a protesticular gene encoding a morphogen that upregulates *Sox9* in developing testis and inhibits the female pathway. Our simulations revealed that the *Fgf9* promoter transitions from HL to HH between E10.5 and E13.5 in XY cells, consistent with its expression pattern (Fig. 3d and Extended Data Fig. 4). *Fgf9* ablation results in male-to-female sex reversal in transgenic mice[44] and gains in copy numbers have been identified in persons with 46 XX sex reversal[45]. Yet, despite the critical role of *Fgf9* in controlling sex determination, nothing is known about its regulation. In XY bipotential cells, when *Fgf9* is expressed at low levels, we observed that critical regulatory regions were predicted to proximal to the gene (Fig. 4a). In XY Sertoli cells, however, we observed a switch toward distal regulation that was concomitant with *Fgf9* upregulation[31]. Importantly, the putative critical regions at the *Fgf9* locus differ from the narrow and well-defined predictions from the *Sox9* locus, as they were broader. These regions involved interactions with other nearby promoters and TAD boundaries and included several ATAC-seq/H3K27ac ChIP-seq peaks. Specifically, our analyses identified a noncoding region located approximately 250 kb downstream of *Fgf9*, whose deletion was predicted to be disruptive of its HH metaloci. Interestingly, the human homologous region contains gene-wide association study (GWAS) hits associated with abnormal testosterone levels[46–48], a phenotype consistent with *FGF9* altered expression (Supplementary Table 2).

To validate these predictions, we generated a 306-kb homozygous deletion (*Δ306*) within the 1.15-Mb TAD of *Fgf9* in mES cells and subsequently derived transgenic mice. This deletion included most of the *Fgf9* predicted downstream regulatory region with no annotated genes or regulatory elements in gonads (Fig. 4a). RNA-seq analyses revealed a twofold downregulation of *Fgf9* in XY E13.5 mutants gonads, as well as downregulation of other male-specific markers (Fig. 4b,c and Extended Data Fig. 6a). Concomitantly, the ovarian program was activated, reflected by the upregulation of female-specific genes (Fig. 4b and Extended Data Fig. 6a). Meiosis, the first molecular signature of ovarian development[49], was activated with the upregulation of markers like *Stra8* (Extended Data Fig. 6b). At E14.5, gonads from XY mutant mice displayed two distinct phenotypes; they developed as ovotestes or ovary-like gonads (Fig. 4d). The ovotestis phenotype featured testicular tissue at the center of the gonad and ovarian tissue at the poles (Extended Data Fig. 6c). Immunofluorescence analyses confirmed the presence of male markers such as SOX9 in testicular tissue, as well as female markers such as FOXL2 in ovarian regions (Extended Data Fig. 6c). Similar patterns were observed in ovary-like mutant gonads, albeit with an increased content in ovarian tissue. These results denoted the initial activation of the male program but a failure in propagating the testis-determining signal to the gonad (Extended Data Fig. 6c). The expression of SYCP3 in mutant gonads confirmed meiosis initiation, as shown in RNA-seq experiments (Extended Data Fig. 6c). The two phenotypes observed in *Δ306* mutants mirror those described for the full *Fgf9* knockout (KO)[44], albeit with

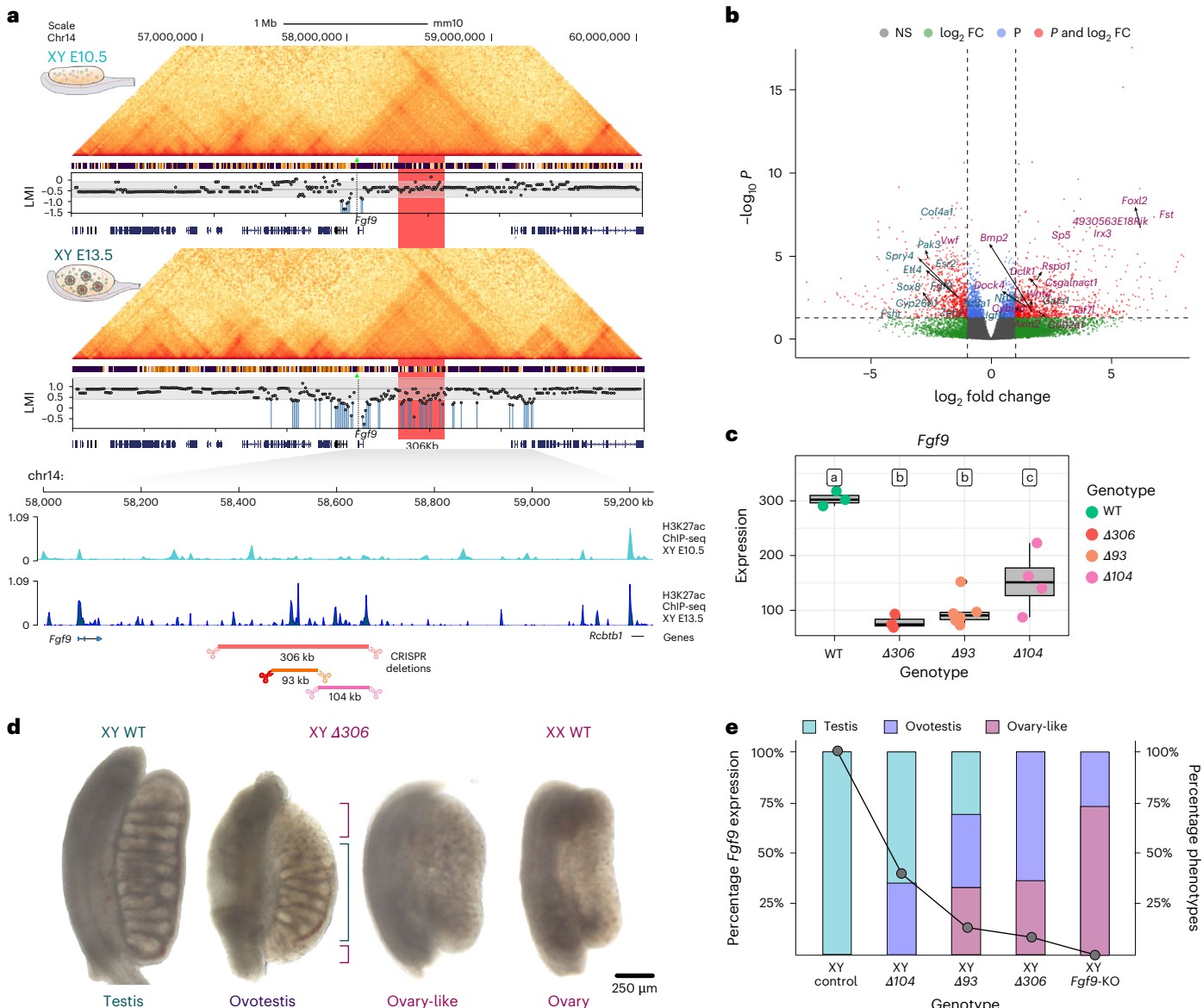

**Fig. 4 | A noncoding region at the *Fgf9* locus is associated with male-to-female sex reversal. a**, Top: predictive scanning analysis at the *Fgf9* locus. Hi-C and ChIP-seq tracks are displayed for XY E10.5 and XY E13.5. Each dot in the scatter plot indicates the value of LMI for *Fgf9* after the deletion of a particular bin set in the scanned region. Vertical blue lines mark regions whose deletion is predicted to decrease the LMI of the metaloci below 1 s.d. (below gray area; Methods). The red transparent shape indicates the region containing most regulatory potential within the *Fgf9* TAD. Bottom: zoomed-in view of the *Fgf9* TAD region. H3K27 ChIP-seq tracks are shown for XY E10.5 and E13.5 cells (Sertoli). Note the abundance of ATAC peaks across the adjacent gene desert to *Fgf9*. Deleted regions in mutant mice are indicated below. **b**, Volcano plot of RNA-seq from XY E13.5 *Δ306* mutant and control gonads. Dysregulated genes associated to female and male differentiation[43] are indicated. Differential expression was assessed using DESeq2 (two-sided Wald test with Benjamini–Hochberg-adjusted *P* values). FC, fold change. **c**, Expression levels of *Fgf9* in control WT and mutant (*Δ104*, *Δ93* and *Δ306*) gonads. Box plots show normalized DESeq2 expression

between different genotypes. Expression values of individual samples are depicted with points colored according to the genotype. The results of pairwise Wald tests (two-sided) are shown using compact letter display. Briefly, the differences in expression of a particular gene across two genotypes are statistically indistinguishable if one letter is shared (FDR-corrected *P* values < 0.05). Comparisons (log$_2$ fold change, adjusted *P*): WT–*Δ306* (1.95, 4.96 × 10$^{-7}$), WT–*Δ93* (1.64, 2.84 × 10$^{-7}$), WT–*Δ104* (1.00, 1.75 × 10$^{-2}$), *Δ306*–*Δ93* (−0.31, 0.689), *Δ306*–*Δ104* (−0.95, 0.0618) and *Δ93*–*Δ104* (−0.64, 0.103). A positive log$_2$ fold change indicates higher expression in the first condition. Box plots display the median (center line), the IQR (25th–75th percentiles) and whiskers (±1.5× the IQR). **d**, E14.5 gonads of *Δ306* mutants and controls (*n* = 20). Note the two phenotypes: ovotestis (13/20) and ovary-like (7/20). The green bracket indicates the testicular portion in the center of the ovotestis. The purple bracket indicates the ovarian portion at the poles of the ovotestis. **e**, Correlation between *Fgf9* expression and gonadal phenotypes in control WT and mutants (*Δ104*, *Δ93* and *Δ306*). Data from *Fgf9*-KO mutants correspond to the analyses reported in a previous study[44].

differences in their frequency. While an ovarian-like phenotype was more often observed in *Fgf9*-KO mice, most *Δ306* mutants developed ovotestes, consistent with the residual *Fgf9* expression (Fig. 4d,e).

Next, we generated two additional mouse models carrying smaller deletions within the 306-kb region (*Δ93* and *Δ104*; Fig. 4a). RNA-seq analyses of mutant gonads revealed reduced *Fgf9* gonadal expression upon the 93-kb deletion, with similar levels as for *Δ306* mutants

(Fig. 4c and Extended Data Fig. 6b), indicating that this genomic region accounts for most of the *Fgf9* regulatory potential. Interestingly, the 104-kb deletion also reduced *Fgf9* expression, albeit to more moderate levels (Fig. 4c and Extended Data Fig. 6b). Importantly, the observed variations in *Fgf9* expression were reflected at the phenotypical level. Most *Δ93* mutants developed ovotestis or ovary-like gonads, similar to their *Δ306* counterparts (Fig. 4e and Extended Data Fig. 7a). However,

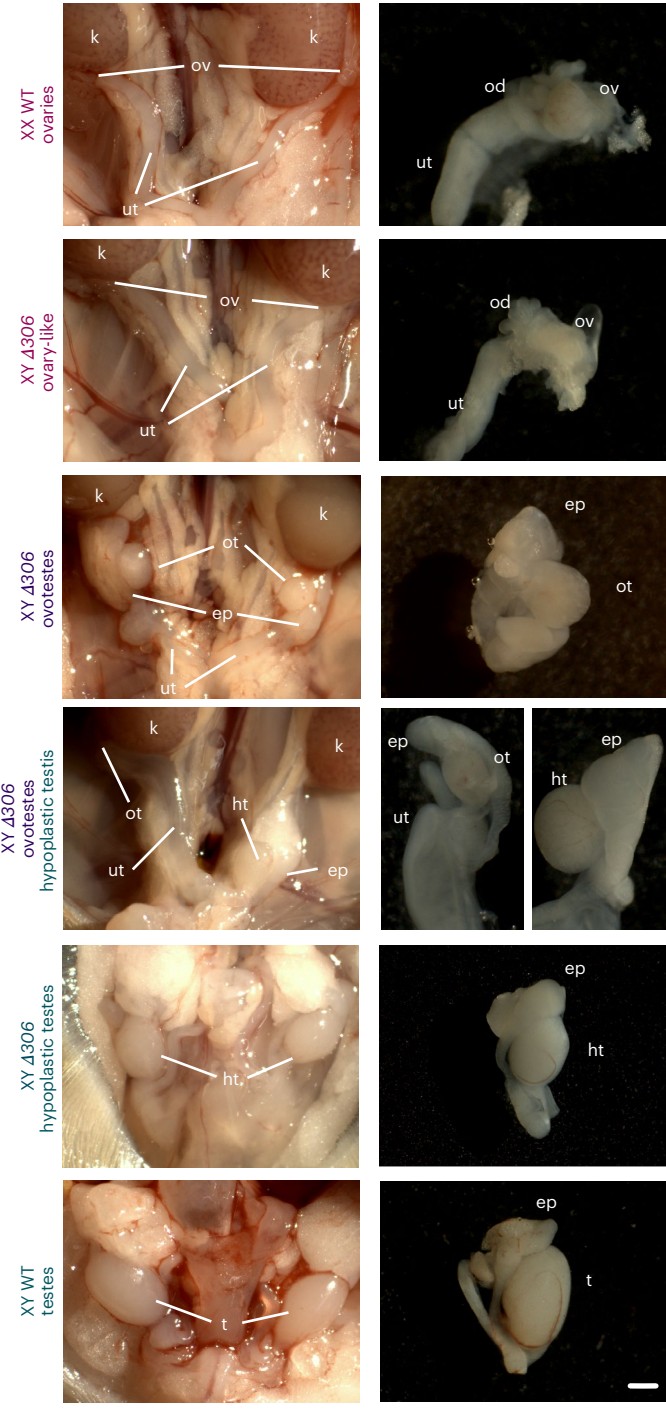

XX WT ovaries

XY Δ306 ovary-like

XY Δ306 ovotestes

XY Δ306 ovotestes hypoplastic testis

XY Δ306 hypoplastic testes

XY WT testes

**Fig. 5 | Phenotypical analysis of Δ306 mutants at postnatal stages.** Left: urogenital tracts (the digestive tract was removed for better visualization). Right: gonadal phenotypes. Note that ovary-like and ovotestes develop close to the kidneys, while testes are descended. ep, epididymus; t, testes; ht, hypoplastic testes; ut, uterine horns; k, kidneys; ot, ovotestes; od, oviducts; ov, ovary-like gonad. Scale bar, 75 μm (urogenital tract pictures), 200 μm (top four gonad pictures) and 100 μm (bottom two gonad pictures).

some Δ93 mutants also developed testes, a phenotype never observed on Δ306 mutants, indicating that sex reversal is less severe. In contrast, ovary-like gonads were not observed for Δ104 mutants, which exclusively displayed phenotypes ranging from testes to ovotestes. Principal component analyses (PCAs) of RNA-seq data from our entire collection of mutants revealed that most variation (54.08%) is explained by expression changes in genes associated with sex-related processes

(Extended Data Fig. 7c,d). Wild-type (WT) controls clustered together in the PCA space, consistent with their testicular phenotypes. Conversely, Δ306 and Δ93 mutants clustered but in a much wider space, likely reflecting their range of phenotypical variation (from ovaries to ovotestes). Remarkably, the Δ104 mutant samples were dispersed among the testis and the ovotestes/ovary clusters, consistent with their variable phenotypes. Collectively, these findings provide an explanation to the broad predictions retrieved from the in silico analysis of the *Fgf9* locus, suggesting that the regulatory potential is mostly distributed over the 306-kb region. As such, it is likely that *Fgf9* gonadal expression is sustained by redundant enhancer elements, associated with the ATAC-seq/H3K27Ac ChIP-seq peaks contained within the 93-kb and 104-kb subregions.

Lastly, we explored the tissue specificity of our in silico predictions. *Fgf9*-KO mutants undergo neonatal lethality, resulting from lung hypoplasia[44]. To investigate whether the Δ306 mutation causes lethality, we generated additional animals through aggregation methods. Those mutants developed normally and were alive after birth, thus overcoming the lethality associated with *Fgf9* complete loss of function. Phenotypic analyses on postnatal day 11 revealed that the sex-reversed phenotypes observed at embryonic stages were also maintained during the postnatal period. Phenotypes ranged from partially descended hypoplastic testes and ovotestes to full ovaries (Fig. 5). As such, the use of tissue-specific genome structure in our predictions allowed us to overcome the lethality of *Fgf9* loss of function, uncoupling gonadal and lung phenotypes. In summary, our transgenic experiments validated the in silico predictions at the *Fgf9* locus, revealing a critical noncoding region controlling mammalian sex determination.

**Gene regulatory networks analyses reveal a redundant function for *Meis* genes in the specification of sexual identity**
Lastly, we reconstructed gene regulatory networks associated with sex determination. We used public single-cell RNA sequencing (scRNA-seq) datasets[50,51] to extract single-cell transcriptional information from all cell types (XY and XX E10.5 *Nr5a1*-expressing cells, XY E13.5 *Sox9*-expressing cells and XX E13.5 *Runx1*-expressing cells). The data were used to infer gene expression correlations (positive and negative) using SCENIC, an approach to reconstruct gene regulatory networks[52,53]. We adapted the SCENIC pipeline to use the output derived from METALoci (schematics in Extended Data Fig. 8a), narrowing down the genomic location of CREs within the broader regions defined by the HH H3K27ac metaloci (Supplementary Data 2) with matching ATAC-seq data[27]. This allowed the identification of lineage-specific regulatory modules (named 'regulons'). Regulons correspond to TFs with high cell type specificity with binding sites in the METALoci regions of other genes with similar (positive regulon) or opposite expression patterns (negative regulon). Regulons can be ranked and compared according to the regulon specificity score (RSS), which reflects the degree of cell type specificity and can be used to discover genes involved in lineage specification[54].

We observed that RSSs tended to increase from E10.5 to E13.5, denoting that regulons gain cell type specificity as differentiation progresses (Fig. 6a and Extended Data Fig. 8b). Our analysis highlighted relevant regulons associated with known sex-determining TFs, such as RUNX1 or FOXL2 for granulosa cells or SOX9, SOX8 or SOX10 for Sertoli cells (Fig. 6a,b). Our approach also revealed factors that have not been studied in the context of sex determination, notably negative regulons that may act as repressors of lineage differentiation, an underexplored class compared to activators. Intriguingly, some factors were simultaneously listed as top regulons in the differentiation networks of both sexes. We focused on one of those factors, MEIS1, predicted as a top negative regulon for both granulosa and Sertoli cell differentiation and without previous involvement in sex determination. MEIS1 is a homeodomain protein with roles in hematopoiesis, angiogenesis and eye development and its complete inactivation is embryonically

lethal at E14.5 (refs. 55,56). We explored a potential role of *Meis1* in sex determination by generating XY mutant embryos through tetraploid aggregation and analyzing them before lethal stages. RNA-seq analyses at E13.5 gonads revealed a limited number of dysregulated genes in *Meis1*-KO mice. Among those, the female marker *Foxl2* displayed a significant upregulation (Extended Data Fig. 9a). Mutant gonads had similar size and structure as controls, yet immunofluorescence analyses revealed that the gonadal poles contain supporting cells expressing FOXL2 (Fig. 6c), suggesting that these cells differentiate as granulosa (females) instead of Sertoli (male) cells. These results confirmed that *Meis1* expression is relevant for testis lineage-specification.

MEIS2, also expressed during sex determination and a paralog of MEIS1, was previously shown to recognize and bind to the same DNA motif[57]. Importantly, our gene regulatory networks also featured MEIS2 among top regulons (Extended Data Fig. 8b). Interestingly, *Meis1* and *Meis2* have been shown to cooperate during limb formation and patterning[58,59]. Thus, we explored a potential functional redundancy between *Meis* genes during sex determination. Double-heterozygous *Meis* mutations are lethal, an aspect that precludes the generation of embryos through conventional breeding. To overcome these limitations, we conditionally deleted *Meis1*$^{flox/flox}$ and *Meis2*$^{flox/flox}$ alleles, through the expression of a maternal and paternal germline Cre recombinase (*Zp3*$^{Cre}$ and *Stra8*$^{Cre}$, respectively)[59]. This strategy allows a complete elimination of *Meis1* and *Meis2* zygotic expression. Embryos carrying a deletion of all four *Meis* alleles died around E9.0, before gonadal formation, yet embryos lacking three *Meis* alleles successfully progressed to midgestation and survived until E13.0, allowing the study of a combined *Meis* inactivation during sex determination. Immunofluorescence analyses in XY double-heterozygous embryos at E12.5 revealed analogous effects as for *Meis1* homozygous inactivation (Fig. 6c). Those effects suggest that the deletion of any given combination of two *Meis* alleles can induce female differentiation on a limited number of XY supporting cells, mainly at the gonadal poles. Importantly, male-to-female differentiation effects were intensified upon deletion of three *Meis* alleles. Independently of the combination of deleted alleles (*Meis1*$^{het}$;*Meis2*$^{hom}$ or *Meis1*$^{hom}$;*Meis2*$^{het}$), XY mutant gonads displayed FOXL2$^+$ cells along the entire gonad (Fig. 6c and Extended Data Fig. 9b). Interestingly, sex differentiation effects were also present in XX mutants; upon inactivation of three *Meis* alleles, male differentiation was observed on a limited number of XX supporting cells, denoted by SOX9 expression (Extended Data Fig. 9b). Of note, a triple inactivation of *Meis* alleles caused underdeveloped gonads compared to controls, suggesting that *Meis* genes are also important for gonadal growth. Thus, our findings identify a role for *Meis* genes during sex determination, acting in functional redundancy and being essential for the proper specification of female and male sexual identities.

## Discussion

Here, we investigated the dynamics of the 3D regulatory landscape during sex determination, transitioning from an initially bipotential system to one of two alternative fates. Using conventional Hi-C analyses, we observed limited variation in 3D chromatin organization, especially at the TAD level, between bipotential or differentiated stages. Although this may suggest the existence of a preformed TAD topology, as described for other biological systems[60], it contrasts with the extensive changes at the transcriptional and epigenetic levels during sex determination[27,31,61,62]. Such discrepancy prompted us to develop METALoci, a computational approach that integrates Hi-C and epigenetic data to provide an unbiased quantification of the regulatory environment around each gene. Using this approach, we identified 3D regulatory hubs in a genome-wide fashion. The formation of 3D regulatory hubs, often encompassing promoters, enhancers and/or structural elements, has been described in various cellular contexts[18]. Recent studies suggest that the interactions among the multiple components of 3D regulatory hubs often take place simultaneously[63]. Here, we demonstrate that 3D regulatory hubs are formed and pervasively rewired during sex determination, involving hundreds of genes. We also observe that 3D regulatory hubs can comprise interactions between enhancer and genes, as well as with other nearby promoters or TAD boundary regions (Fig. 4a). Changes in 3D regulatory hubs are minor at the bipotential stage but increase as sex is specified and differentiation progresses.

Transcriptomics analyses have shown that the early supporting lineage of the gonad is primed toward the ovarian fate[31,61,62]. Female priming is also reflected at the 3D regulatory hub level, denoted by moderate gene transitions during granulosa differentiation, in comparison to Sertoli cells. Our analyses also show that regulatory mechanisms leading to dimorphic gene expression are diverse and locus specific. During Sertoli differentiation, male-specific genes are commonly associated with increased regulatory activity, concomitant with decreased activity at many female-specific genes in XY cells. Yet, these mechanisms are not as prominent during granulosa differentiation. It is known that granulosa cells undergo a squamous-to-cuboidal transition upon ovarian follicle activation at postnatal stages[64]. Thus, it is plausible that active changes in regulation may become more obvious at later time points beyond E13.5. Nevertheless, we observed that genes involved in granulosa cell differentiation, such as *Foxl2*, acquire an active regulatory status and occupy prominent positions in gene regulatory networks. Similarly, although scRNA-seq data cannot resolve isoforms, WT1 ranked among the top regulons in female-specific networks, consistent with the ovarian-determining function of its KTS variant[9].

Our limited understanding of sex determination regulation is a challenge for diagnosing causal mutations in DSD[65]. Nearly all known mutations associated with DSD involve coding regions[65]. Although mutations in regulatory regions are expected to be causal in many cases, their identification has been very difficult. METALoci associated each gene with its time-specific and sex-specific 3D regulatory hub, thus providing a functional annotation of the noncoding genome during sex determination. Using METALoci, we captured regulatory interactions with validated enhancers at the *Bmp2* (ref. 27) and *Sox9* (refs. 5,42) loci. Previously, the identification of the Enh13 and TESCO enhancers involved an extensive characterization of the *Sox9* locus[5,42]. ATAC-seq

---

**Fig. 6 | Reconstruction of gene regulatory networks identifies *Meis* genes as sex-determining factors. a**, Top: the top 20 TF regulons for granulosa E13.5 XX (left) and Sertoli E13.5 XY cells (right). Regulons are ranked according to RSS, which is a metric for cell type specificity. Gray small points indicate regulons with mean < 0.3 normalized counts in scRNA-seq, which were discarded for further analysis. Bottom: the top ten positive and negative regulons are indicated. Underlined regulons indicate those factors previously associated with sex determination in the literature. *Meis1*, which was further investigated in this study, is indicated in blue. **b**, Gene regulatory network reconstruction for granulosa E13.5 XX (left) and Sertoli E13.5 XY cells (right). Large nodes (circles) represent TFs associated with top regulons (the name of the TF indicated within a white rectangle). The color of the circles represents the expression of the TF, in normalized RNA counts. Small nodes (diamonds) represent target genes (the name is indicated if the gene is a TF). Connecting lines represent TF–gene interactions, with the size of the line being proportional to the weight of the interaction (that is, correlation of expression in scRNA-seq). Red lines correspond to activation and blue lines correspond to repression. **c**, Immunofluorescence on XY E12.5 *Meis* mutants and controls (*n* ≥ 3). The ovarian marker FOXL2 (green) marks the presence of granulosa cells. Left: testicular marker SOX9 (red) indicates the presence of Sertoli cells. Note the similarities between XY mutants with two deleted *Meis* alleles (*Meis1* homozygous versus *Meis1*/*Meis2* double heterozygous), with FOXL2-positive cells located mainly in the gonadal poles (arrowheads). Note also the increase in the number of FOXL2-positive cells (arrowheads) extending across the entire gonads in mutants with three deleted *Meis* alleles (*Meis1* homozygous, *Meis2* heterozygous).

and DNAseI assays identified 33 putative gonadal enhancers, 16 of which were validated in transgenic mice[5] (Extended Data Fig. 3). Thus, the capability of METALoci analyses in pinpointing critical regulators highlights its potential to facilitate enhancer identification and to reduce the extensive workload of functional validations (Extended Data Fig. 3).

Enhancer identification based on chromatin interaction data generally relies on detecting chromatin loops, which is challenging as loop calling is influenced by the analytical method[66]. These approaches also operate under the assumption that enhancer–promoter interactions are relatively stable and highly frequent.

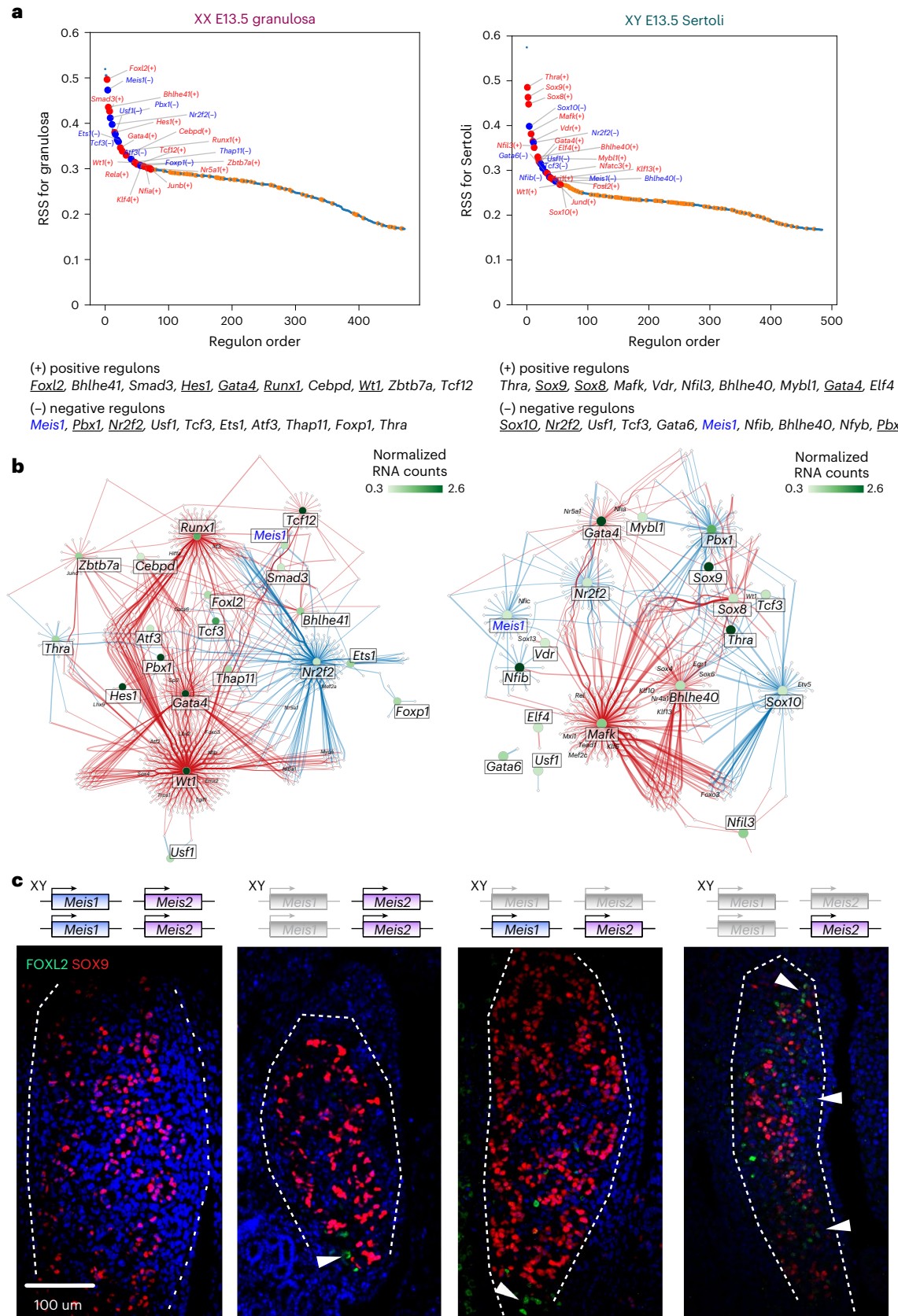

However, emerging models of gene regulation suggest that productive enhancer–promoter communication may rely on transient interactions or physical proximity, rather than direct contact[67,68]. When applied to our datasets, an enhancer detection strategy combining H3K27ac ChIP-seq and chromatin loop information proved largely uninformative for most sex-biased genes, including *Fgf9* (Extended Data Fig. 10a). In contrast, METALoci, which does not rely on predefined structures like chromatin loops, captured the regulatory dynamics underlying sex differentiation most effectively, supporting alternative models of enhancer–promoter communication. We validated these predictions by identifying a noncoding region controlling sex determination at the *Fgf9* locus. A differential H3K27Ac ChIP-seq analysis between Sertoli and granulosa revealed over 20 peaks distributed through the >1-Mb region downstream the *Fgf9* gene, which could be potential regulators. Here, we narrowed down the genomic space to search for critical regulators, demonstrating that the gonadal potential of the *Fgf9* locus is mostly sustained by the 306-kb region. Within this region, critical enhancers are likely located within the central 93-kb subregion, which contains two prominent Sertoli-specific H3K27ac ChIP-seq/ATAC-seq peaks (Extended Data Fig. 10b). Nevertheless, expression and phenotypical analyses demonstrate regulatory redundancy between the 93-kb and the 104-kb subregions. Redundancy has been reported for many developmental genes[69–71], including *Fgf8*, a paralogous gene to *Fgf9*, whose limb expression is controlled by several enhancers distributed over large genomic regions[72]. The gonadal phenotypes from XY Δ306 mutant mice range from ovotestes to ovaries, mimicking those reported in persons carrying coding mutations in *FGF9* (ref. 45) or its gonadal receptor *FGFR2* (ref. 73). Remarkably, this regulatory region contains human GWAS hits associated with fluctuations in testosterone levels[46–48], a phenotype consistent with altered *FGF9* regulation. In addition, METALoci predictions are tissue specific, highlighted by the uncoupling of gonadal and lung phenotypes in *Fgf9* mutants, which overcomes perinatal lethality. This aspect could be exploited for modulating transcription with cell type precision.

Most knowledge on sex-determining genes derives from human DSD-associated mutations. Yet, a molecular diagnosis is often not possible for almost half of DSD cases[65], suggesting that, beyond noncoding regions, relevant genes are yet to be discovered. Our regulatory networks analyses provides an alternative approach to identify such genes, particularly TFs, as they can be classified according to cell type specificity, a proxy for relevance. Using these approaches, we identified candidates such as PBX1, as negative regulator of sex-determining networks. Importantly, DSD-related data have associated *PBX1* mutations with male-to-female sex reversal[74,75]. Moreover, we discovered a role for *Meis* genes in sexual identity specification. Thus, *Meis* genes become part of a reduced list of factors, such as *Nr5a1* or *Wt1*, whose mutations are associated with defective sexual fate during both ovarian and testicular development. Both MEIS1 and MEIS2 can also act as regulators of *PBX1*, by controlling its nuclear translocation[76,77]. Furthermore, our results on *Meis* genes also highlight that sex-determining factors can act in functional redundancy. In that respect, we also observed that regulatory networks feature several HOX TFs among top regulons (Extended Data Fig. 8b), suggesting that these factors could also have a redundant role during sex determination. Compellingly, PBX, MEIS and HOX factors can act as non-DNA-binding partners in trimeric complexes, which may provide additional regulatory control through cofactor interactions[78]. Therefore, functional redundancy could be among the reasons that may have precluded the identification of certain sex-determining factors using traditional Mendelian disease approaches, underscoring the value of genomic approaches for candidate gene discovery. Overall, our study provides important insights into the process of sex determination, highlighting the power of integrative genomic approaches to uncover the molecular underpinnings of developmental processes.

## Online content

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

Irene Mota-Gómez[1,2,13], Juan Antonio Rodríguez [3,12,13], Shannon Dupont[4], Alicia Hurtado[1,2], Vanessa Cadenas[5,6], Leo Zuber[3], Iago Maceda[3], Oscar Lao[7], Johanna Jedamzick[1], Ralf Kühn [8], Scott Lacadie[9], Sara Alexandra García-Moreno[4], Miguel Torres[5,6], Francisca M. Real [2], Rafael D. Acemel [1,2], Blanche Capel [4 ✉], Marc A. Marti-Renom [3,10,11 ✉] & Darío G. Lupiáñez [1,2 ✉]

[1]Max-Delbrück Center for Molecular Medicine in the Helmholtz Association (MDC), Berlin Institute for Medical Systems Biology (BIMSB), Epigenetics and Sex Development Group, Berlin, Germany. [2]Centro Andaluz de Biología del Desarrollo (CABD), CSIC/UPO/JA, Seville, Spain. [3]Centre Nacional d'Anàlisi Genòmica (CNAG), Barcelona, Spain. [4]Department of Cell Biology, Duke University Medical Center, Durham, NC, USA. [5]Cardiovascular Regeneration Program, Centro Nacional de Investigaciones Cardiovasculares (CNIC), Madrid, Spain. [6]Centro de Investigación Biomédica en Red de Enfermedades Cardiovasculares (CIBERCV), Madrid, Spain. [7]Institut de Biologia Evolutiva (UPF-CSIC), Department of Medicine and Life Sciences, Universitat Pompeu Fabra, Parc de Recerca Biomèdica de Barcelona, Barcelona, Spain. [8]Max-Delbrück Center for Molecular Medicine in the Helmholtz Association (MDC), Berlin, Germany. [9]Max-Delbrück Center for Molecular Medicine in the Helmholtz Association (MDC), Berlin Institute for Medical Systems Biology (BIMSB), Computational Regulatory Genomics Group, Berlin, Germany. [10]Centre for Genomic Regulation (CRG), Barcelona Institute of Science and Technology (BIST), Barcelona, Spain. [11]ICREA, Barcelona, Spain. [12]Present address: Center for Evolutionary Hologenomics, University of Copenhagen, Copenhagen, Denmark. [13]These authors contributed equally: Irene Mota-Gómez, Juan Antonio Rodríguez. ✉e-mail: blanche.capel@duke.edu; martirenom@cnag.eu; dario.lupianez@csic.es

## Methods

Research carried out in this study complied with all relevant ethical regulations for animal experimentations. Mice used for tetraploid complementation assays were handled according to institutional guidelines under an experimentation license (G0111/17 and G0051/22) approved by the Landesamt fuer Gesunheit und Soziales or according to the Spanish law and EU Directive 2010/63/EU, with approvals from the University Pablo de Olavide Ethics Committee and the Junta de Andalucía (reference 09/02/2024/024). Reporter mice used for FACS of the cell populations of the gonad were handled in accordance with National Institutes of Health guidelines and with the approval of the Duke University Medical Center Institutional Animal Care and Use Committee (A089-20-04 9N). Mice used for the generation of the Cre/lox lines were handled in accordance with Centro Nacional de Investigaciones Cardiovasculares (CNIC) Ethics Committee, Spanish laws and the EU Directive 2010/63/EU for the use of animals in research. All mouse experiments were approved by the CNIC and Universidad Autónoma de Madrid Committees for 'Ética y Bienestar Animal' and the area of 'Protección Animal' of the Community of Madrid with reference PROEX 220/15. Newly generated materials are available upon request.

### Statistics and reproducibility

Statistical tests and sample sizes are indicated in figure legends. All experiments were performed in at least two biological replicates and good reproducibility was assessed by Pearson correlation. Sample sizes were two biological replicates for each sample in Hi-C experiment, with two to four technical replicates from each biological replicate. RNA-seq and immunofluorescence analysis of mutant and WT gonads were performed with more than three biological replicates. No statistical method was used to predetermine sample size. Samples were excluded only according to the genotype assessed in control experiments. The experiments were not randomized. The investigators were not blinded to allocation during experiments and outcome assessment.

### Transgenic mice

Progenitor supporting cells were isolated from both sexes at E10.5, using an *Sf1–eGFP* line[31]. At this stage, before the expression of *Sry*, *Sf1–eGFP* cells are bipotential because of their capacity to differentiate toward either the female or the male lineage. At E13.5, a *Sox9–eGFP* line was used to isolate Sertoli cells from developing testes[31], while a *Runx1–eGFP* line was used to obtain their counterparts in ovaries, the granulosa cells[11]. The *Sf1–eGFP* (*Nr5a1–eGFP*) and *Sox9–eCFP* reporter mouse lines previously generated were maintained on a C57BL/6 (B6) background[79,80]. The *Runx1–GFP* reporter mouse was generously gifted by H. Yao at the National Institute of Environmental Health Sciences[11] (MMRRC_010771-UCD). Timed matings were generated with reporter males and WT CD-1 females. The morning of a vaginal plug was considered E0.5. Embryos were collected at E10.5 and E13.5, with genetic sex determined using PCR for the presence or absence of the Y-linked gene *Uty* (pF1: TCATGTCCATCAGGTGATGG, pF2: CAATGTGGACCAT-GACATTG, pR: ATGGACACAGACATTGATGG). XY was indicated by two bands and XX was indicated by one band.

### Cell collection with FACS

Gonads were dissected from E10.5 or E13.5 embryos and the mesonephros was removed using syringe tips. The gonads were incubated in 500 µl of 0.05% trypsin for 6–10 min at 37 °C and then mechanically disrupted in 1× PBS with 10% fetal calf serum (FCS). The cell suspension was pipetted through a 40-µm filter top and the supporting cells were collected with FACS. Cell sorting was performed using the Duke Cancer Institute Flow Cytometry Shared Resource. GFP-positive cells were sorted using a Becton Dickinson (BD) DiVa, controlled using BD FACSDIVA software (version 7), and analyzed with the BD Fortessa X-20 using the BD FACSDiVa software. CFP-positive cells were sorted using a Beckman-Coulter Astrios and analyzed with the BD Fortessa X-20 using

the BD FACSDiVa software. Postsort purity was determined to be greater than 97% by reanalyzing the postsort fraction by FACS. Cell population abundance was on average as follows: 2.8% for E10.5 XX, 2.5% for E10.5 XY, 30% for E13.5 XX and 20% for E13.5 XY. Representative FACS plots, including the gating strategy, are shown in Supplementary Figs. 1 and 2.

### Cell preparation for Hi-C

Gonads were dissected from E10.5 or E13.5 embryos and the mesonephros was removed using syringe tips. The gonads were incubated in 500 µl of 0.05% trypsin for 6–10 min at 37 °C and then mechanically disrupted in 1× PBS with 10% FCS. The cell suspension was pipetted through a 40-µm filter top and the supporting cells were collected with FACS. After FACS, cells were prepared for Hi-C analysis as follows. Cells were spun down at 300$g$ for 5 min at 4 °C and resuspended in 250 µl of 1× PBS with 10% FCS. The cells were then fixed in a final concentration of 2% PFA in PBS with 10% FCS for 10 min at room temperature. The crosslinking reaction was quenched with the addition of 50 µl of 1.425 M glycine and the cells were put on ice. Next, the cells were spun for 8 min (750$g$ at 4 °C) and the supernatant was removed. The cell pellet was resuspended in cold lysis buffer (50 mM Tris, 150 mM NaCl, 5 mM EDTA, 0.5% NP-40, 1.15 Triton X-100 and 6.25× protease inhibitor cocktail). Cells were centrifuged for 3 min (1,000$g$ at 4 °C) and the supernatant was removed. Cells were then snap-frozen in liquid $N_2$ and stored at −80 °C until use.

### Hi-C library preparation

The low-input Hi-C protocol was performed from fixed, lysed and snap-frozen cells as previously described[32], with some modifications. Pelleted aliquots were thawed on ice and resuspended in 25 µl of 0.5% SDS to permeabilize nuclei and incubated at 62 °C for 10 min. SDS was quenched by adding 12.5 µl of 10% Triton X-100 and 72.5 µl of $H_2O$ and incubated for 45 min at 37 °C with rotation. Chromatin was then digested by adding MboI (5 U per µl) in two installments in NEB2.1 digestion buffer for a total of 90 min, adding the second installment after 45 min. Digestion was heat-inactivated for 20 min at 65 °C. DNA overhangs were filled with biotin-14-dATP (0.4 mM), dTTP, dGTP, dCTP (10 mM) and DNA Pol I Klenow (5 U per µl) and incubated for 90 min at 37 °C with gentle rotation. Filled-in chromatin was then ligated by adding ligation master mix (60 µl of 10x R4 DNA ligase buffer, 50 µl of 10% Triton X-100, 6 µl of BSA (20 mg ml⁻¹) and 2.5 µl of T4 ligase in two installments) for a total of 4 h at room temperature with gentle rotation, with the second installment after 2 h. Ligated chromatin was then spun down for 5 min (2,500$g$ at room temperature) and reverse-crosslinked by resuspension in 250 µl of extraction buffer (10 mM Tris pH 8.0, 0.5 M NaCl, 1% SDS and 20 mg ml⁻¹ proteinase K) and incubated for 30 min at 55 °C while shaking (1,000 rpm). Next, 56 µl of 5 M NaCl was added and incubated overnight at 65 °C while shaking (1,000 rpm). Chromatin was then purified using phenol, chloroform and isoamyl (25:24:1), precipitated with ethanol and resuspended in 15 µl of Tris pH 8.0. DNA was quantified at this step by Qubit and RNA was digested by adding 1 µl of RNase and incubating for 15 min at 37 °C. Next, biotin was removed from unligated fragments by adding 5 µl of 10x NEB2 buffer, 1.25 µl of 1 mM dNTP mix, 0.25 µl of BSA (20 mg ml⁻¹), 25.5 µl of $H_2O$ and 3 µl of T4 DNA polymerase (3 U per µl) and incubated for 4 h at room temperature. Sample volume was then brought up to 120 µl and DNA shearing was performed with a Covaris S220 (two cycles, each 50 s: duty, 10%; intensity, 4; 200 cycles per burst). Biotin pulldown was performed by adding an equal volume of Hi-C sample with Dynabeads MyOne Streptavidin T1 beads (Invitrogen, 65602) and incubated by 15 min with rotation. Beads were washed two times with bind and wash buffer (10 mM Tris-HCl pH 7.5, 1 mM EDTA and 2 M NaCl) and a final wash with 10 mM Tris-HCl pH 8.0 was performed. Samples were resuspended in 50 µl of Tris-HCl pH 8.0. Library preparation was performed using the NEBNext Ultra DNA library prep kit for Illumina (E7645L). Briefly, end repair of the libraries was performed by adding 6.5 µl of 10x end repair reaction buffer and 3 µl of end prep enzyme mix and incubated at room

temperature for 30 min followed by 65 °C for 30 min. Next, adaptor ligation was performed by adding 15 µl of blunt/TA ligase master mix, 2.5 µl of NEBNext adaptor for Illumina and 1 µl of ligation enhancer. The mixture was incubated at room temperature for 15 min, followed by the addition of 3 µl of USER enzyme and incubation for 15 min at 37 °C. The beads were separated on a magnetic stand and washed two times with 1× bind and wash buffer + 0.1% Triton X-100. Sample was transferred to a new tube and a final wash was performed with 10 mM Tris pH 8.0 before resuspending the beads in 50 µl of 10 mM Tris pH 8.0.

For PCR library amplification, the sample was divided into four reactions of 12.5 µl to optimize the number of cycles. The PCR was performed using 12.5 µl of library bound to beads, 25 µl of 2× NEBNext Ultra II Q5 master mix, 5 µl of 10 µM universal primer, 5 µl of 10 µM indexed PCR primer and 7.5 µl of nuclease-free $H_2O$, with the following PCR program: 1 min at 98 °C, followed by 12–20 cycles of 10 s at 98 °C and 75 s at 65 °C, ramping at 1.5 °C s$^{-1}$, with a final elongation at 65 °C for 5 min. Double size selection was performed with AmpureXP beads. Library quantification was assessed with Qubit dsDNA high-sensitivity kit and the size and quality of the libraries were checked using TapeStation.

## Hi-C data processing

Raw data were processed and filtered with Juicer[81] using default parameters. For downstream analysis, Knight–Ruiz (KR)-normalized Hi-C matrices in hic format were converted to FAN-C[82] format at 10-kb and 100-kb resolutions including a low-coverage filtering step to exclude bins with less than 20% relative coverage. Renormalization of the filtered matrices was performed using the KR normalization method.

## A/B compartment analysis

A/B compartment analysis was performed using FAN-C[82] in each replicate and chromosome individually in the normalized 10-kb KR matrices. After a high Pearson correlation between replicates was confirmed, the matrices of both replicates were merged. The first eigenvector was calculated again in the merged matrices and the sign of the eigenvector was corrected if needed depending on the G+C percentage and amount of ATAC-seq signal in each chromosome independently.

Chromosomes X and Y were excluded from this analysis, as they are not comparable between XX and XY samples. For differential compartment analysis, pairwise comparisons were performed using BEDtools[83] by counting the number of bins that corresponded to the A or B compartment in each sample. Genes and gene expression belonging to each compartment type were included in the analysis using the BEDtools intersect function. To test significance in differential gene expression between compartments, Benjamini–Hochberg-corrected P values were reported after a pairwise Mann–Whitney U-test and chi-squared test.

## Insulation analysis

Insulation scores and boundary scores[84] were calculated in the 10-kb KR-normalized merged matrices using FAN-C[82] (parameters: window size, 500 kb; impute_missing = TRUE). To consider that a certain boundary was a TAD boundary, a threshold of 0.25 in the boundary score was used based on visual inspection as recommended by the FAN-C developers. Chromosomes X and Y were excluded for the downstream analysis as they are not comparable between XX and XY samples. A total set of boundaries was obtained using the BEDtools 'cat' (parameters: postmerge = false) and 'merge' functions[83] (parameters: --d = 2001). Subsequently, the BEDtools 'intersect' function was used to assess which boundaries were present or absent in each sample.

To generate a quantitative analysis on insulation, a pairwise set of boundaries were generated between the samples that needed to be compared (early sex-specific, late sex-specific, XX temporal and XY temporal). Next, the insulation scores of both datasets were mapped to the common set of boundaries and an absolute difference in insulation score was calculated. Aggregate profile plots of insulation were generated using deepTools[85].

## METALoci, genome spatial autocorrelation analysis

The main goal of METALoci is to identify spatially autocorrelated signals in the structure of the genome. In contrast to methods such as HiCorr[86] or PSICHIC[87] that focus on identifying promoter–enhancer interactions or chromatin assortativity[40] that aims to identify proteins or chromatin marks mediating genomic contacts, METALoci relies for the first time on geostatistics approaches to identify spatially autocorrelated signals in the genome in an unbiased way (that is, independent of precalculated structural features from Hi-C). All METALoci analysis was performed using the METALoci Python 3 library publicly available (https://github.com/3DGenomes/METALoci). The code relies on a series of standard libraries such as SciPy, NumPy (1.21.6), Pandas (1.3.5), Matplotlib (3.5.2) and seaborn (0.11.2), as well as other specialized libraries such as GeoPandas (https://geopandas.org; 0.10.2), NetworkX (https://networkx.org; 2.6.3), libpysal (https://pysal.org; 4.6.2), ESDA (https://pysal.org/esda/; 2.4.1) and pyBigWig library from deepTools (https://deeptools.readthedocs.io; 0.3.18).

**Genome parsing.** The first step in METALoci is to define the set of genomic regions of interest to analyze. This can be a single gene or a series of ad hoc selected regions. Specifically for this work, the mouse reference genome (mm10, December 2011) was parsed taking each of the bins containing a transcription start site (TSS) for any of the 24,027 annotated genes as a center point for METALoci. Each region of interest was then centered in its gene TSS and a total of 2 Mb of DNA upstream and downstream was included. This resulted in a list of 24,027 regions of interest each of 4 Mb of DNA that were run for the METALoci analysis (Supplementary Table 3).

**Hi-C interaction data parsing.** METALoci uses as input normalized Hi-C interactions at 10-kb resolution, produced as described above. Normalized data were first $\log_{10}$-transformed and subset to remove any interaction that was below a score of 1.0. This cutoff for interaction selection can be defined by the user and balances the consistency of the resulting Kamada–Kawai layout described below and the computational burden. It should be noted that this cutoff (--cutoff parameter in METALoci layout) will determine the amount of interaction data that will be used from the Hi-C input matrices. Currently, METALoci has an automatic setting to select the best cutoff given the input Hi-C dataset, which relies on the signal-to-noise ratio in the input matrix. Users are invited to assess different cutoffs beyond the automatically selected one. Additionally to the cutoff parameter, METALoci assigns a strength to consecutive bins in the layout on the basis of what we call 'persistence length' of the polymer. This parameter (--pl in METALoci layout) contrasts the imposed restraints selected by the above cutoff by increasing or decreasing the bendability of the polymer. A very high persistence length value would result in straight layout, whereas a very low persistence length value would result in a 'zig-zag' layout. Currently, METALoci has an automatic setting to select the best persistence length given the input Hi-C dataset and the cutoff parameter. Users are invited to assess different persistence length values beyond the automatically selected one. In our application to the mouse genes, several cutoffs were assayed for the list of genes; a cutoff of 1.0 resulted in layouts consistent to others produced with different cutoffs with a reasonable computational time. The persistence length parameter was set to 10.0. The subset matrix was then transformed from interaction frequencies (that is, a 'similarity' matrix) to the inverse of the interactions (that is, a 'distance' matrix). Finally, the resulting pairwise distances between any pair of bins in the region of interest was saved as a sparse matrix to input to the Kamada–Kawai graph layout algorithm.

**Kamada–Kawai layout.** Next, the sparse distance matrix obtained from Hi-C was used as the source to generate a graph layout that best represents the observed genomic interactions. This was accomplished by using the Kamada–Kawai graph layout[41], which attempts to position

nodes (that is, genomic bins) on a space of 1 × 1 arbitrary units so that the geometric distance between them is as close as possible to the input distance matrix. It should be noted that the size of the arbitrary space has no effect on the final layout apart from changing its scale, which is irrelevant to the next steps of METALoci. The 'kamada_kawai_layout' function of the NetworkX python library was used with default parameters to generate the final layouts and obtain the Cartesian 2D coordinates for each of the genomic bins of 10 kb. Next, the closed Voronoi polygons for each of the bins were calculated using the 'Voronoi' function of the SciPy spatial library. The bins at the edge of the layout were closed by placing eight dummy nodes closing the entire space occupied by the layout. This ensured that every single genomic bin had a finite polygon. Next, a buffer distance around each bin was placed corresponding to 1.5 times the mean spatial distance between consecutive genomic bins. Finally, the spatial occupancy of each of the genomic bins was calculated as the intersection of their Voronoi polygon and the buffer space around them. This resulted in a 'worm-like' 2D representation of each Kamada–Kawai layout that we named a 'Gaudí plot' as it resembles the famous broken tile mosaics or 'trencadís' by the Catalan architect Antoni Gaudí (Fig. 2b).

**H3K27ac signal mapping into the graph layout.** Next, METALoci was input the normalized H3K27ac ChIP-seq signal, produced as described above. H3K27ac coverage per each of the 10-kb bins was obtained using the pyBigWig library, which resulted in a read coverage for each of the bins into the Kamada–Kawai layout. It is important to note that the H3K27ac signal is input in METALoci as the coverage per bin; as such, there is no step for peak detection or normalization of the data by peak length. Next, the H3K27ac signal was $\log_{10}$-transformed and mapped into each of the polygons of the Gaudí plots. The final result is, thus, a graph layout representing the input Hi-C interactions and the mapped H3K27ac signal onto the space occupied by each genomic bin. This is then used as input to assess the spatial autocorrelation of H3K27ac using LMI analysis[37].

**LMI autocorrelation analysis.** LMI is a measure describing the overall dependence of a given signal over nearby locations in space. LMI is computed as the weighted average of the values of autocorrelation at each $i$ sampled point[36,37]:

$$\mathrm{Moran's\ I} = \frac{\sum_{i=1}^{n} \mathrm{LMI}_i}{n}$$

$$\mathrm{LMI}_i = z_i \sum_{j}^{n} \frac{w_{ij} z_j}{\sum_{j}^{n} w_{ij}}$$

where $z_i$ is the normalized signal at point $i$, $n$ is the total number of points (genomic bins) in the layout and $w_{ij}$ is the assigned weight between point $i$ and $j$. Positive LMI values are obtained when a point $|z_i| > 0$ is surrounded by points with similar values (that is, HH or LL values) and it is indicative of a hub of points with similar behavior around location $i$. Negative LMI values are obtained when a point $|z_i| > 0$ is surrounded by points with the reverse pattern (that is, HL or LH values) and it is suggestive of negative autocorrelation at location $j$. LMI values close to cero indicate poor spatial dependence between contiguous points for the considered signal.

Weights between bins in the Kamada–Kawai graph were calculated on the basis of their spatial distance informed by the Hi-C contacts. A distance band was assessed by the 'weights.DistanceBand' function of the libpysal python library with a distance cutoff corresponding to three times the mean distance between consecutive genomic bins. This ensured that the weights calculated would be based on at least two upstream and two downstream bins as the buffer space for a bin was calculated as 1.5 times the mean distance between consecutive

genomic bins (above). Next, the weights were input to the 'Moran_Local' function of the ESDA python library with default parameters and for a total of 50,000 permutations to assess the statistical significance of the LMI scores for each bin. For each permutation, the LMI statistic was recalculated and a pseudo $P$ value was computed as the number of times a permuted statistic was equal to or more extreme than the observed statistic. During the perturbation assay in LMI, it is important to consider the spatial distribution of particles while randomizing as some observations might have more neighbors than others. To avoid this problem, ESDA implements what is called 'maximal cardinality', which first assesses the maximum number of neighbors across all observations and then, for each observation, its row sum is divided by this maximal cardinality instead of its actual number of neighbors. This approach ensures that observations with fewer neighbors are not given disproportionate weight in the permutation analysis. Another aspect that the user may want to consider during the permutation test is the linearity of the genome. Two linearly adjacent genomic loci are more likely to have similar signals than two loci separated in sequence. We included in METALoci the option to account for such an effect. However, we did not use this option in the current work as linearity in enhancer signatures have been previously and extensively described (for example, super-enhancers[88]) and accounting for linearity as a confounding factor would result in miss detection of collections of regulatory enhancers that are proximal in the sequence of the genome.

The results of the LMI calculations are the Moran I score, the Moran I quadrant and its significance for each of the bins in the Gaudí plots. Thus, the LMI analysis results in all bins placed into any of the four quadrants of the Moran scatter plot: the HH (red) quadrant for bins with high signal with a neighborhood of high signal, the LH quadrant (cyan) for bins with low signal with a neighborhood of high signal, the LL quadrant (blue) for bins with low signal with a neighborhood of low signal and the HL quadrant (orange) for bins with high signal with neighborhood of low signal. Moreover, after randomizing the signal values over the layout a user-defined number of times, the algorithm also produces a probability value for each assignment being random. We selected significant HH, LH, LL and HL bins according to a $P$ value < 0.05. Contiguous bins with significant LMI of the same quadrant and their immediate neighbors correspond to what we call 'metaloci' of the signal. Here, we were interested in detecting genes whose TSS (that is, the bin in the genomic middle of the layout) was considered a metalocus for enrichment of H3K27ac mark in the HH quadrant (that is, the TSS and its spatial neighborhood are enriched in H3K27ac).

**LMI volcano plots.** LMI inverted volcano graphs (Fig. 2c) were plotted by changing the signal of the LMI score for each bin in quadrants LH and LL and changing the signal of the Moran's $\log_{10}(P$ value) for bins in quadrants HL and LL. We selected bins containing the TSS gene as significant in each quadrant if the absolute value of LMI was larger than 1.0 and the absolute $\log_{10}(P$ value) was larger than 1.3 ($P < 0.05$).

**Gene transitions.** A gene transition was calculated as the distance (in arbitrary units) that the gene makes in the LMI inverted volcano between two or more sample points. Specifically, we calculated gene transitions for XX and XY cells between time points E10.5 and E13.5. A gene transition is positive if the vector connecting the two analyzed time points for the gene of interest points toward the top right corner of the LMI inverted volcano (that is, the HH quadrant). A gene transition is negative if the vector connecting the two analyzed time points for the gene of interest points toward the bottom left corner of the LMI inverted volcano (that is, the LL quadrant).

## Gene Ontology term enrichment
Lists of selected genes were used to analyze gene enrichment of biological process Gene Ontology (GO) terms using the website for WebGestalt (http://webgestalt.org, accessed September 2022)[89] with coding genes

in the mouse genome as a background list. Only GO terms that were deemed significant (FDR < 0.01) were kept.

**SCENIC TF network analysis.** PySCENIC (https://github.com/aertslab/pySCENIC; 0.12.1), a Python implementation of the SCENIC package, was used to uncover gene regulatory networks by integrating single-cell gene coexpression analysis with motif enrichment[90]. The basic steps of PySCENIC are to firstly obtain a normalized cell type expression matrix from scRNA-seq, which in this work was obtained from a previous study (https://github.com/IStevant/XX-XY-mouse-gonad-scRNA-seq/tree/master/data/all_count.Robj)[50,51]. Next, gene coexpression modules are generated to identify groups of genes that are coexpressed across the scRNA-seq data. Then, motif enrichment is performed on the gene modules using a TF motif database (V10: 2022 SCENIC+; file v10nr_clust_public subset to contain only JASPAR vertebrate motifs[91]) to predict which TFs are likely regulating the gene modules. This is accomplished in SCENIC by searching motifs in a FASTA file composed of sequences around the promoter of known genes (that is, 10 kb upstream or downstream from TSS). Lastly, AUCell analysis is performed in SCENIC to score the activity of the identified TFs across cell types. All four steps were executed with default parameters as previously described[52] with the exception of the use of a METALoci-derived set of sequences for the enrichment analysis in the third step. Specifically, we created our own FASTA file with regulatory regions to input to 'create_cisTarget_database' software (https://github.com/aertslab/create_cisTarget_databases) used to generate a database of sequences and their motifs for SCENIC. Our customized FASTA file contained only sequences of open chromatin (that is, overlapping an ATAC-seq peak) that were part of an HH METALoci bin for the H3K27ac mark, which were specific for each of the four cell types analyzed. In summary, SCENIC was used with default parameters but forced to search for motifs within an open chromatin with an enhancer environment spatially closed to a promoter, regardless of their sequence proximity (Extended Data Fig. 8a). Finally, the resulting networks were visualized using CytoScape[92].

**Simulation of genomic perturbations**
To computationally predict the effect of CRISPRing out regions of the genome, we devised a strategy where five consecutive bins of 10 kb would be removed using a running window from the beginning to the end of the region of interest in one-bin steps. Once a set of five bins was removed, all interactions from those bins and the H3K27ac signal were removed and a new METALoci analysis was performed on the resulting Hi-C map and H3K27ac signal. Next, we assessed whether a particular deletion of 50 kb (five bins) could affect the metaloci status for the bin containing the TSS of the gene of interest. Bin removals that decreased the LMI for the TSS by more than 1 s.d. of all analyzed deletions were annotated as predicted perturbation affecting the gene of interest (Fig. 4a, blue lines in predictions for the *Fgf9* gene).

**Generation of mouse mutants through tetraploid aggregation**
Deletions at the *Fgf9* locus and for the *Meis1* gene were generated on G4 mES cells using CRISPR–Cas9 as previously described[93]. G4 cells were obtained from A. Nagy's lab (Lunenfeld-Tanenbaum Research Institute). For each experiment, two single guide RNAs (sgRNAs) were designed in the regions of interest using Benchling (https://www.benchling.com). The sequences of the sgRNAs are listed in Supplementary Table 4. For the *Fgf9* METALoci perturbation analysis experiments, the bin numbers in the primer names correspond to the number of bins away from *Fgf9*. To generate *Δ306* mutants, sgRNAs within bins 230 and 259 were used. To generate *Δ93* mutants, sgRNAs within bins 240 and 250 were used. For *Δ104* mutants, sgRNAs within bins 250 and 259 were used. The absence of the deleted region was assessed by genotyping the flanking regions of the deletion and by genomic qPCR using three different pairs of primers located in different areas inside the

deletion (Supplementary Table 4). Edited cells were then used to generate embryos using tetraploid complementation assay as previously described[93,94]. CD-1 female and male mice of various ages were used as donors and fosters for embryo retransferring by tetraploid aggregation. The specimens isolated to perform experimental analysis were Bl6/129Sv5 male mice, E13.5 and E14.5 in age. All mice were housed in standard cages at the Animal Facilities of the Max-Delbrück Center for Molecular Medicine in Berlin or the Centro Andaluz de Biología del Desarrollo in Seville in a pathogen-free environment.

**Generation of mouse mutants through Cre/lox recombination**
*Meis*-conditional-KO embryos were obtained following the strategy described previously[59] by mating *Meis1^flox* and *Meis2^flox* with the Cre lines *Stra8^Cre* and *Zp3^Cre*.

To obtain embryos at different gestational stages, mice were mated in the afternoon and females were checked every morning for the presence of a vaginal plug; noon on the day the plug was observed was considered as gestational day 0.5 (E0.5). Embryos at somitogenic stages were staged according to age and somite number.

**Immunofluorescence**
Gonads were dissected out at E14.5, fixed in Serra fixative solution (60% ethanol, 30% formic acid and 10% acetic acid), prepared for standard histological methods with paraffin embedding and sectioned in 5-μm slides. Immunofluorescence was performed as previously described[22]. The primary antibodies and working dilutions used in this study were SOX9 (Merck Millipore, AB5535; 1:600), FOXL2 (abcam, ab5096; 1:150) and SCYP3 (abcam, ab15093; 1:200). The secondary antibodies and working dilutions were Alexa Fluor 488 donkey anti goat IgG (Life Technologies, A11055; 1:200), Alexa Fluor 555 donkey anti rabbit IgG (Life Technologies, A31572; 1:200). All microscopy-related information can be found in Supplementary Table 5.

**RNA-seq**
Mutant and WT gonads were dissected at the stage of E13.5 in 1× PBS and snap-frozen in liquid nitrogen. RNA was then extracted from individual gonads using an RNeasy micro kit (Qiagen, 74004), following the manufacturer's specifications. Quality of RNA was assessed using TapeStation and samples were stored for a maximum of 1 week at −80 °C. Libraries were prepared using the NEBNext Ultra II directional RNA library prep kit for Illumina (E7760), using the protocol that included the poly(A) magnetic isolation module (E7490) following the specifications of the manufacturer. Library quality was checked in a TapeStation. Sequencing was performed using 200-bp paired-end reads in a NovaSeq 6000 sequencer.

**ChIP-seq processing**
H3K27ac ChIP-seq reads were obtained from the Gene Expression Omnibus (GEO; GSE118755)[27]. In this publication, Garcia-Moreno et al. obtained progenitor supporting cells from both sexes at E10.5 using the same *Sf1–eGFP* transgenic mouse line as used here. In the case of XY Sertoli cells, H3K27Ac data from Garcia-Moreno et al. were obtained from sorted cell populations using a TESCO–CFP mouse line, whereas, in this study, we used an *Sox9–eGFP* line. Both lines mark Sertoli cells at E13.5. In the case of XX granulosa cells, H3K27Ac data from Garcia-Moreno et al. were obtained from sorted cell populations using the TESMS–CFP line, whereas, in this study, we used an *Runx1–eGFP* line. Both lines mark granulosa cells at E13.5.

H3K27ac ChIP-seq reads were mapped to the mm10 genome assembly with bowtie2 with default parameters[95]. Mapped reads were filtered for mapping quality and PCR duplicates using SAMtools 'view' and 'markdup' (parameters: -q 30)[96]. Mapped reads from replicates were combined with SAMtools 'merge', extended according to sample and control average fragment estimates ('x') from MACS2 (ref. 97) and converted to bigWig signal tracks using deepTools 'bamCompare'

where control background signals (for example, input) were subtracted from the foreground (parameters: --operation subtract --binSize 50 --scaleFactorsMethod None --normalizeUsing CPM --smoothLength 250 --extendReads 'x').

## ATAC-seq processing

ATAC-seq reads were obtained from the GEO (GSE871155)[27] and trimmed for adaptors using flexbar (parameters: -u 10)[98] followed by mapping to the mm10 genome assembly with bowtie2 with default parameters[95]. Mapped reads were filtered for mapping quality and PCR duplicates using SAMtools 'view' and 'markdup' (parameters: -q 30)[96]. The resulting BAM files were converted to BED files using BEDtools[83] and the 5′ end of mapped coordinates was extended 15 bp upstream and 22 bp downstream according to strand using BEDtools 'slop' (parameters: -l 15 -r 22 -s) to account for sterics during Tn5 transposition[99]. Replicates of extended coordinate BED files were concatenated and then converted back to BAM format with BEDtools and finally to bigWig format using deepTools 'bamCoverage' (parameters: --binSize 10 --normalizeUsing CPM --smoothLength 50 --extendReads 38)[85]. ATAC-seq peaks were called using MACS2 'callpeak' (parameters: -f BAM, --keep-dup all --q 0.01)[97].

## Enhancer identification through chromatin loop analysis

Loop calling was performed with Mustache at a 5-kb resolution and using an FDR threshold of 0.05 (ref. 100). Chromatin loops in which one anchor overlapped with a differential H3K27Ac ChIP-seq peaks and the other with a gene promoter were retained. The analysis was performed for granulosa XX and for Sertoli XY cell differentiation and for genes displaying sex-biased expression.

## scRNA-seq analysis

The count data matrix was obtained from a previous study (https://github.com/IStevant/XX-XY-mouse-gonad-scRNA-seq/tree/master/data/all_count.Robj)[50]. Cells of the clusters of interest were then extracted and divided randomly into two pseudoreplicates. Counts from these two pseudoreplicates were summed. Scaling and gene expression was then performed using the R package DEseq2 (ref. 101) and treated as bulk RNA-seq with two replicates.

## RNA-seq bulk data processing

Alignment and gene expression quantification were performed using the PiGx RNA-seq pipeline with default parameters[102] using the Mus_musculus.GRCm39 reference genome and annotation provided by Ensembl. Briefly, reads were trimmed with Trim Galore (https://github.com/FelixKrueger/TrimGalore) and aligned with STAR[103]. Per-gene quantification was performed using the GenomicAlignments::summarizeOverlaps R function[104]. Differential analysis was then performed using DESeq2 (ref. 101). Specifically, the FDR-corrected P values of all pairwise comparisons between genotypes were computed using DESeq2::contrast function (using Wald tests). The PCA plot was obtained with the DESeq2::vst and DESeq2::plotPCA functions with default parameters.

## Reporting summary

Further information on research design is available in the Nature Portfolio Reporting Summary linked to this article.

## Data availability

The Hi-C and bulk RNA-seq datasets generated in this study can be obtained from the GEO under accession code GSE217618. The scRNA-seq count matrix from supporting used populations was obtained from GitHub (https://github.com/IStevant/XX-XY-mouse-gonad-scRNA-seq/tree/master/data/all_count.Robj). ChIP-seq and ATAC-seq raw fastq files were obtained from the GEO under accession code GSE118755. Source data are provided with this paper.

## Code availability

METALoci is available as a Python 3 package from GitHub (https://github.com/3DGenomes/METALoci), which can be used through command line tools or an API. The remaining analysis was performed with previously published software packages or scripts, which are maintained and available in their respective repositories.

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

## Acknowledgements

We thank the sequencing core, transgenic unit and animal facilities of the Max-Delbrück Center for Molecular Medicine for technical assistance. We thank C. Scholl, M. Altmann, G. Kussagk, S. Bomberg, S. Reissert-Oppermann, C. Westphal, A.J. Franco, C. Mateos, P. López, L. Pérez, A. López, A. Benitez, I. Guerrero and J. López-Rios for their support with the transgenic work. We thank F. Martínez Real and members of the D.G.L. laboratory for their valuable input and comments on the paper. Research in the D.G.L. lab was funded by the Deutsche Forschungsgemeinschaft (International Research Training Group 2403, including PhD fellowship to I.M.-G.), by the Spanish Agencia Estatal de Investigación (grant no. PID2022-143253NB-I00/AEI/10.13039/501100011033/FEDER), and by the European Research Council (grant no. 101045439, 3D-REVOLUTION). Funded by the European Union. Views and opinions expressed are, however, those of the authors only and do not necessarily reflect those of the European Union or the European Research Council Executive Agency. Neither the European Union nor the granting authority can be held responsible for them. R.D.A. was supported by a European Molecular Biology Organization postdoctoral fellowship (grant no. EMBO ALTF 537-2020). M.A.M-R. acknowledges support by the Spanish Ministerio de Ciencia e Innovación (grant numbers PID2020-115696RB-I00 and PID2023-151484NB-I00) and Generalitat de Catalunya (2021-SGR-01127). M.T. and V.C. acknowledge support by the Pro CNIC Foundation and the Spanish Ministerio de Ciencia e Innovación (PID2022-140058NB-C31 and CEX2020-001041-S). B.C. and S.D. were supported by grants from the National Institutes of Health (R01-HD039963 and R01-HD103064).

The content is solely the responsibility of the authors and does not necessarily represent the official views of the National Institutes of Health. The funders had no role in study design, data collection and analysis, decision to publish or preparation of the manuscript.

## Author contributions

I.M.-G., B.C., M.A.M.-R. and D.G.L. conceptualized the study and designed the experiments. I.M.-G. performed most of the experiments, with the support of A.H. S.D. and S.A.G.-M. performed the gonadal cell collections. V.C. and M.T. obtained and processed the embryonic material from *Meis*-KO mutants. J.A.R. and M.A.M-R conceptualized METALoci with the help of O.L. J.A.R., L.Z., I.M and M.A.M.-R. developed and applied METALoci. I.M.-G. and M.A.M.-R. performed most of the computational analyses, with the support of R.D.A. and S.L. I.M.-G., M.A.M.-R. and D.G.L. analyzed the data. A.H., J.J., R.K. F.M.R. and D.G.L. performed the tetraploid aggregations. I.M.-G., M.A.M.-R. and D.G.L. wrote the paper with input from all authors.

## Funding

## Competing interests

The authors declare no competing interests.

## Additional information

**Extended data** is available for this paper at https://doi.org/10.1038/s41594-026-01749-z.

**Correspondence and requests for materials** should be addressed to Blanche Capel, Marc A. Marti-Renom or Darío G. Lupiáñez.

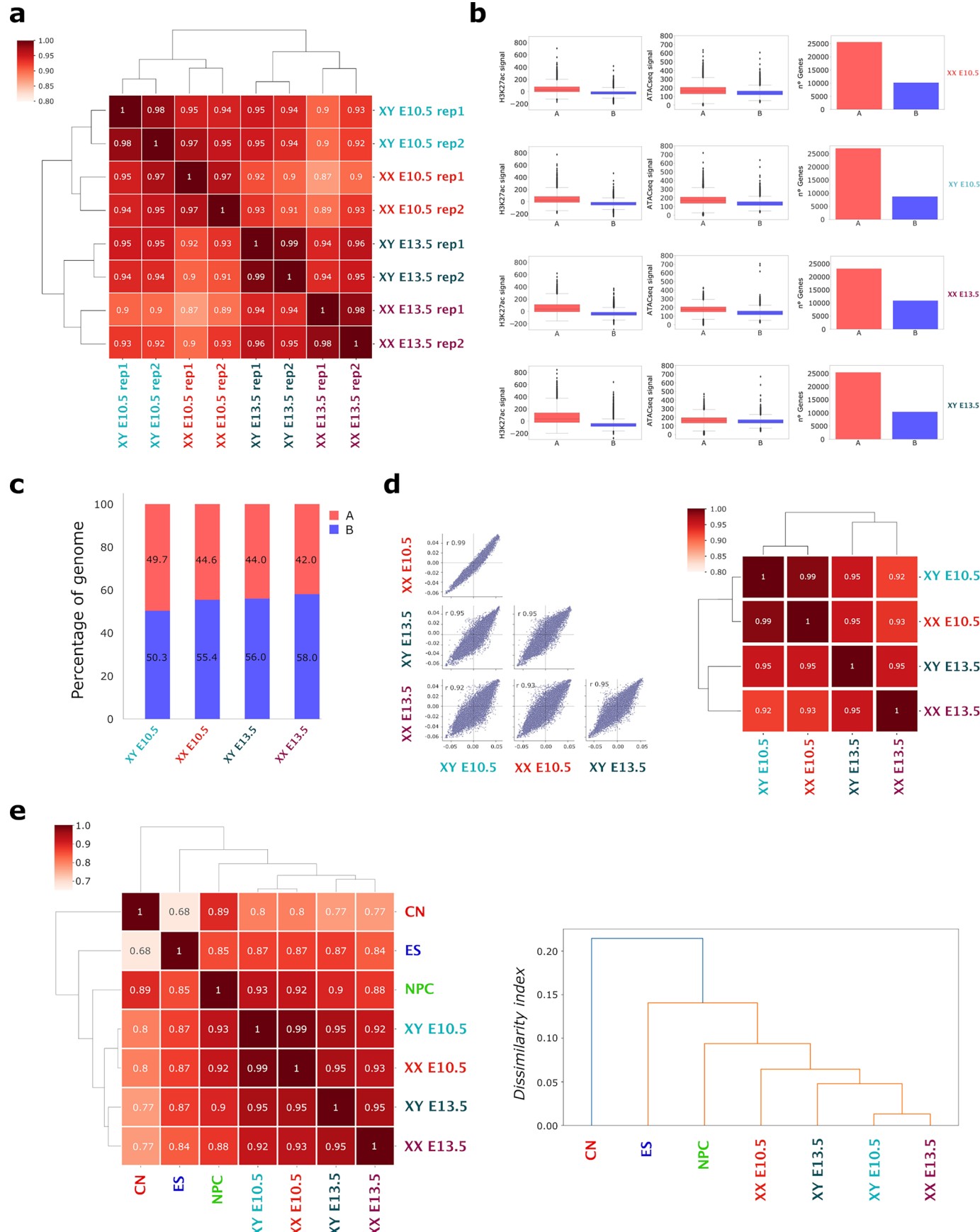

**Extended Data Fig. 1 | See next page for caption.**

**Extended Data Fig. 1 | A/B compartment analyses. a.** Correlation of Hi-C replicates based in A/B compartment signal. Pearson correlation of the first eigenvector between samples and individual replicates. Note that replicates from each sample have increased correlation compared to other samples. **b.** Correlation of A/B compartments with active chromatin, open chromatin and with transcription. H3K27ac ChIP-seq and ATAC-seq data obtained from[16]. Left: H3K27ac normalized signal (CPM) in A and B compartments. Middle: normalized ATAC-seq signal (CPM) in A and B compartments. Right: number of genes in A and B compartments. Boxplots show the median, IQR (25th–75th percentiles), and whiskers (± 1.5×IQR). Compartment bin counts: XX E10.5 A/B = 11,582/13,057; XY E10.5 = 12,788/11,851; XX E13.5 = 10,947/13,692; XY E13.5 = 11,485/13,154. **c.** Percentage of the genome located in A/B compartments. Stacked barplots representing which percentage of the genome corresponds to A compartment (red) or B compartment (blue) in each of the samples. **d.** Correlation of A/B compartments between different samples. Left panel. Scatterplot of pairwise correlation analysis of different samples. Right panel. Heatmap of pairwise correlation analysis of different samples. For both graphs, note the increased correlation between XX E10.5 and XY E10.5, compared to other samples. **e.** Correlation of Hi-C samples from different cell types, based in A/B compartment signal. Left panel. Pearson correlation of the first eigenvector between samples from this study and from[33]. Right panel. Dissimilarity index in A/B compartments between different samples. CN: Cortical Neurons; NPC: Neuronal Progenitors; ES: mouse embryonic stem cells.

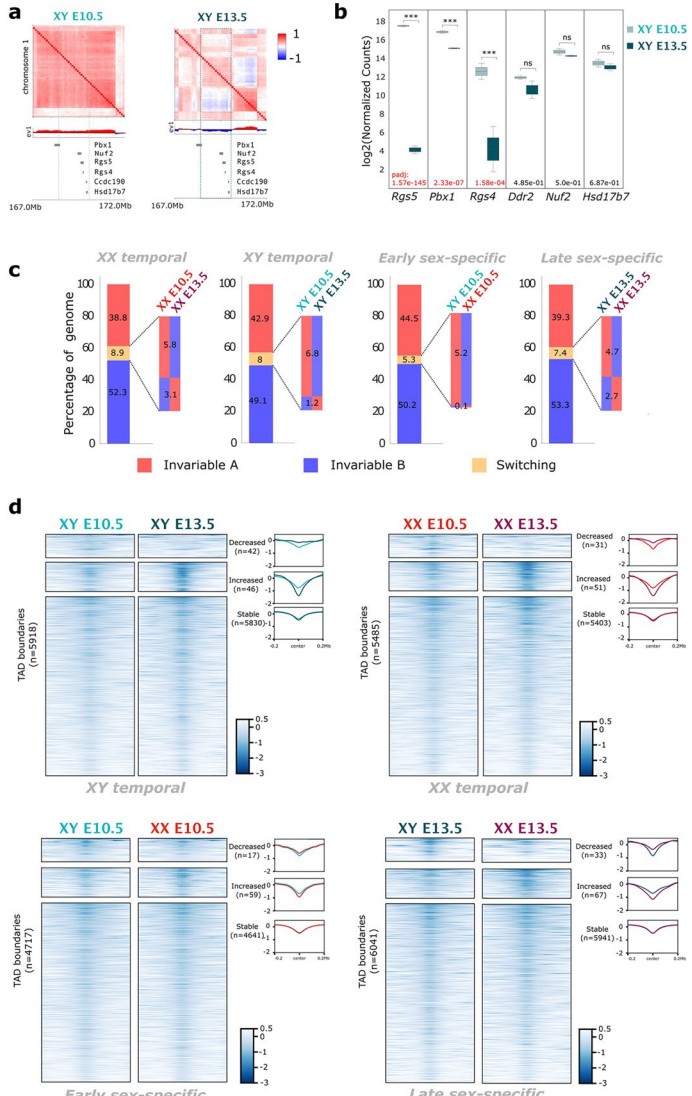

**Extended Data Fig. 2 | Dynamics of 3D chromatin organization during sex determination. a**. Example of a genomic regions that switch compartments during Sertoli cell differentiation. **b**. Expression levels of genes contained withing the A to B switched region in panel **a**. Note the decrease in expression for genes contained within the region. Differential expression was assessed using a two-sided Wald test with Benjamini–Hochberg correction. Adjusted p-values (p-adj) are shown below the boxplots (ns, not significant; * p-adj < 0.05; ** p-adj < 0.01; *** p-adj < 0.001). Boxplots show the median, IQR (25th–75th percentiles), and whiskers (±1.5×IQR). **c**. Percentage of genome that switches compartments or remain invariable. **d**. Insulation heatmaps for TAD boundaries identified in pairwise comparisons. Note that the "decreased" and "increased" heatmaps are enlarged with respect to the "stable" category, to facilitate visualization.

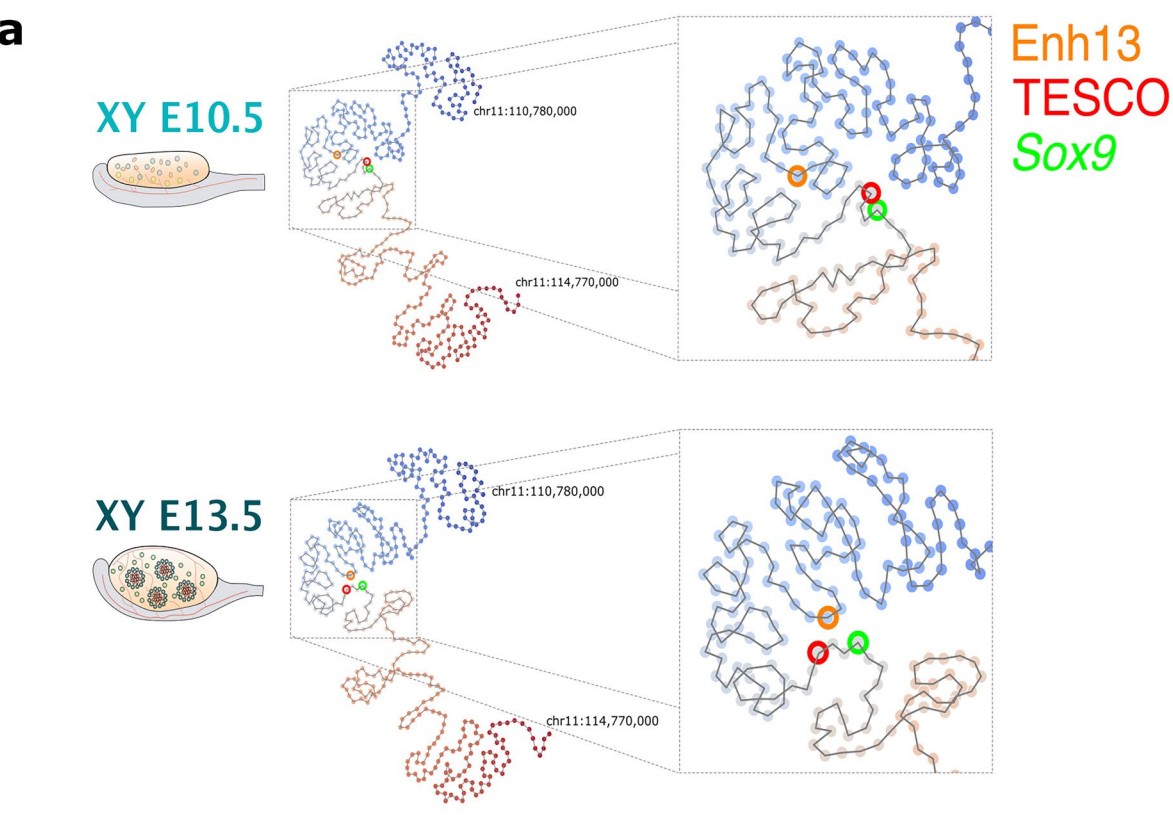

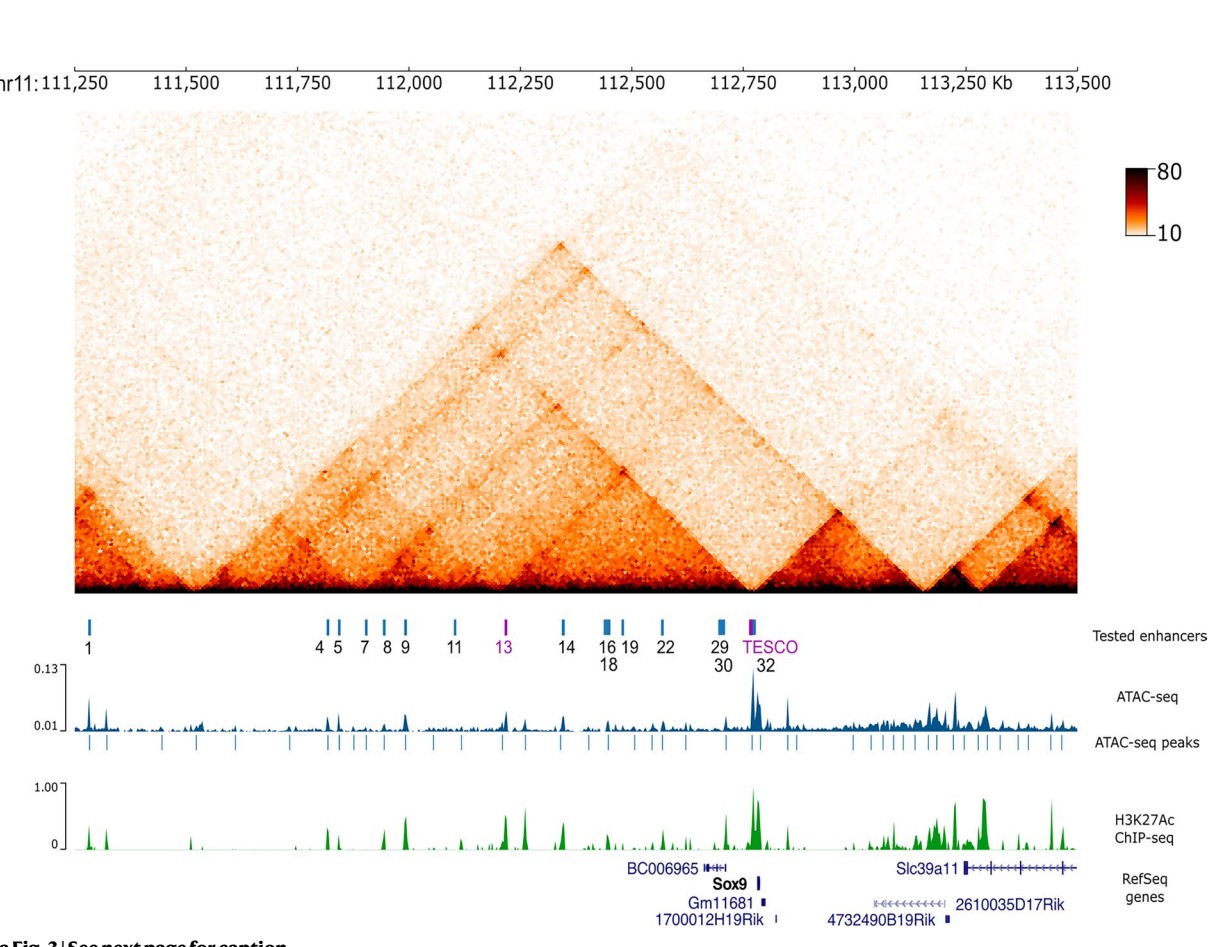

**Extended Data Fig. 3 | See next page for caption.**

**Extended Data Fig. 3 | *METALoci* analysis at the *Sox9* locus during Sertoli cell differentiation. a**. Spatial dynamics of the *Sox9* locus. Left. Kamada-Kawai 2D layout of the Sox9 locus at XY E10.5 and E13.5 cells. Right panel. Zoom in of the squared region of the 2D layout. Note the increased proximity between Enh13 and the *Sox9* promoter during Sertoli cell differentiation. The distance between the TESCO enhancer and the *Sox9* promoter remain similar. **b**. *METALoci* identifies critical regulators at the *Sox9* locus. Hi-C maps, as well as ATAC-seq and H3K27 ChIP-seq tracks are shown for Sertoli E13.5 XY cells (Sertoli). *METALoci* perturbations are shown in the lower track, in which dots that are below the defined threshold (dots with bars) represent 10Kb regions which deletion is predicted to cause a disruption of *Sox9* regulatory activity. The tested enhancer candidates from[5] are indicated. From a total of 33 putative enhancers, 16 were tested in mouse *LacZ* reporter assays (blue squares) and only two were shown to reduce *Sox9* expression upon deletion (purple squares). Note that *METALoci* predictions identifies these two enhancers as the functionally relevant among the upstream gene desert to *Sox9*. METALoci also identifies other potential regulators on the downstream TAD to *Sox9*.

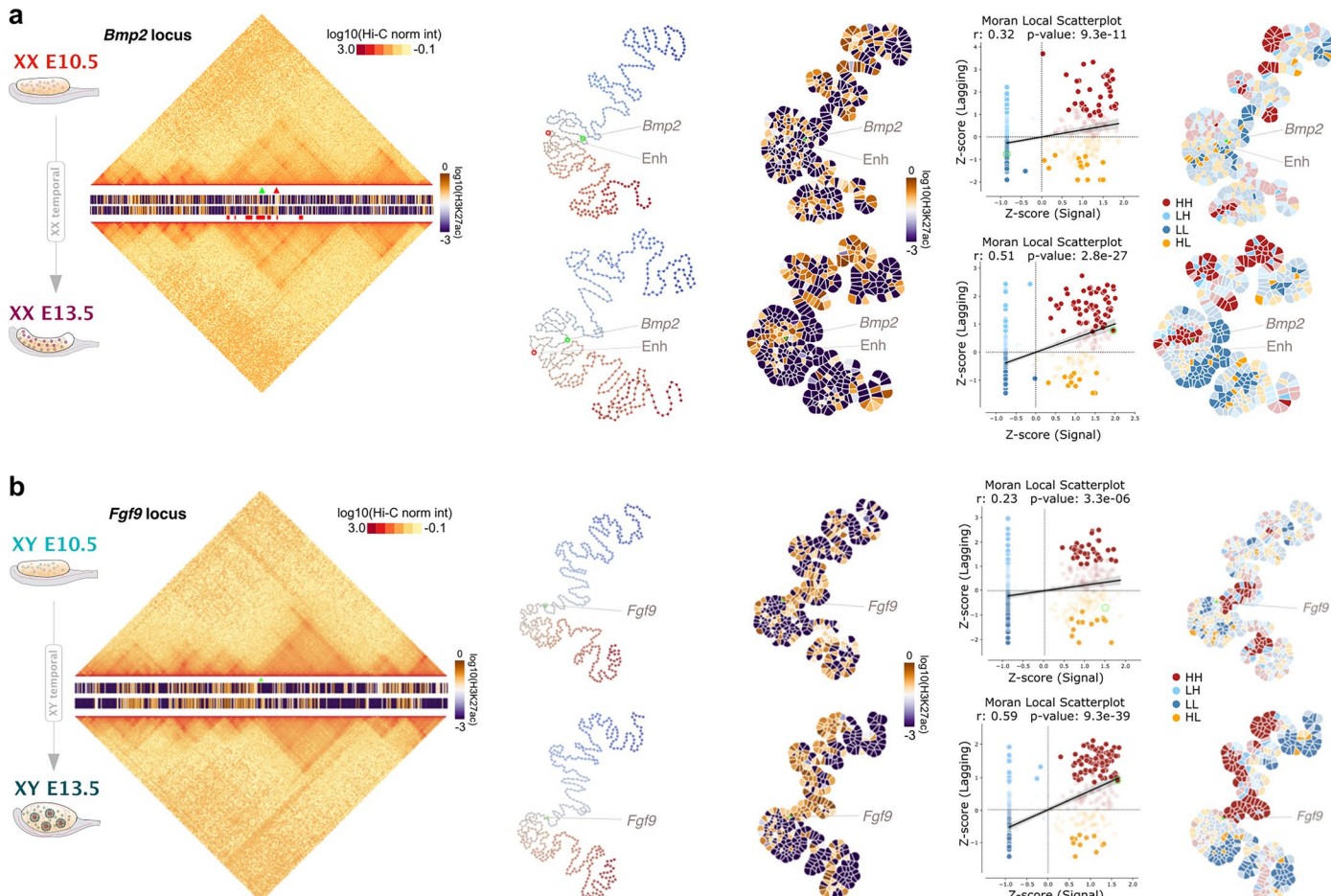

**Extended Data Fig. 4 | *METALoci* analysis for at selected loci during sex determination. a**. *METALoci* analysis for the *Bmp2* locus during sex determination. From left to right. First, Hi-C data and H3K27ac signal for *Bmp2* locus centered at chr2:131,550,000-135,540,000 coordinates. Hi-C and H3K27ac ChIP-seq for granulosa cells development (XX E10.5 to XX E13.5). The position of the *Bmp2* promoter (green arrowhead), as well as the recently identified enhancer[27] (red arrowhead), are highlighted. Squared red marks under H3K27ac track indicate the non-continous metaloci detected for *Bmp2* locus at XX E13.5. Second, Kamada-Kawai 2D layout landmarking the position in the map for the *Bmp2* promoter (green circle) as well as the annotated enhancer (red circle). Third, H3K27ac signal mapped into the graph layout and represented as a Gaudí

plot. Fourth, LMI scatter plot. Fifth, Gaudí plot highlighting the LMI quadrants with solid color indicating statistical significance (p < 0.05). For the regression plot, significance assessed using a parametric regression test (t-test on the regression coefficient). For the LMI in the Gaudi Plot significance assessed using a Monte Carlo permutation test of spatial randomness. **b**. *METALoci* analysis for the *Fgf9* locus during testicular determination. From left to right. First, Hi-C data and H3K27ac signal for *Fgf9* locus centered at chr14:56,070,000-60,060,000 coordinates. Hi-C and H3K27ac ChIP-seq for XY E10.5 and XY E13.5 cells are displayed. Second, Kamada-Kawai layout highlighting the *Fgf9* locus (green circle). Third, H3K27ac Gaudí plot. Fourth, LMI scatter plot. Fifth, Gaudí plot of LMI quadrants. Significance tests are the same as indicated above.

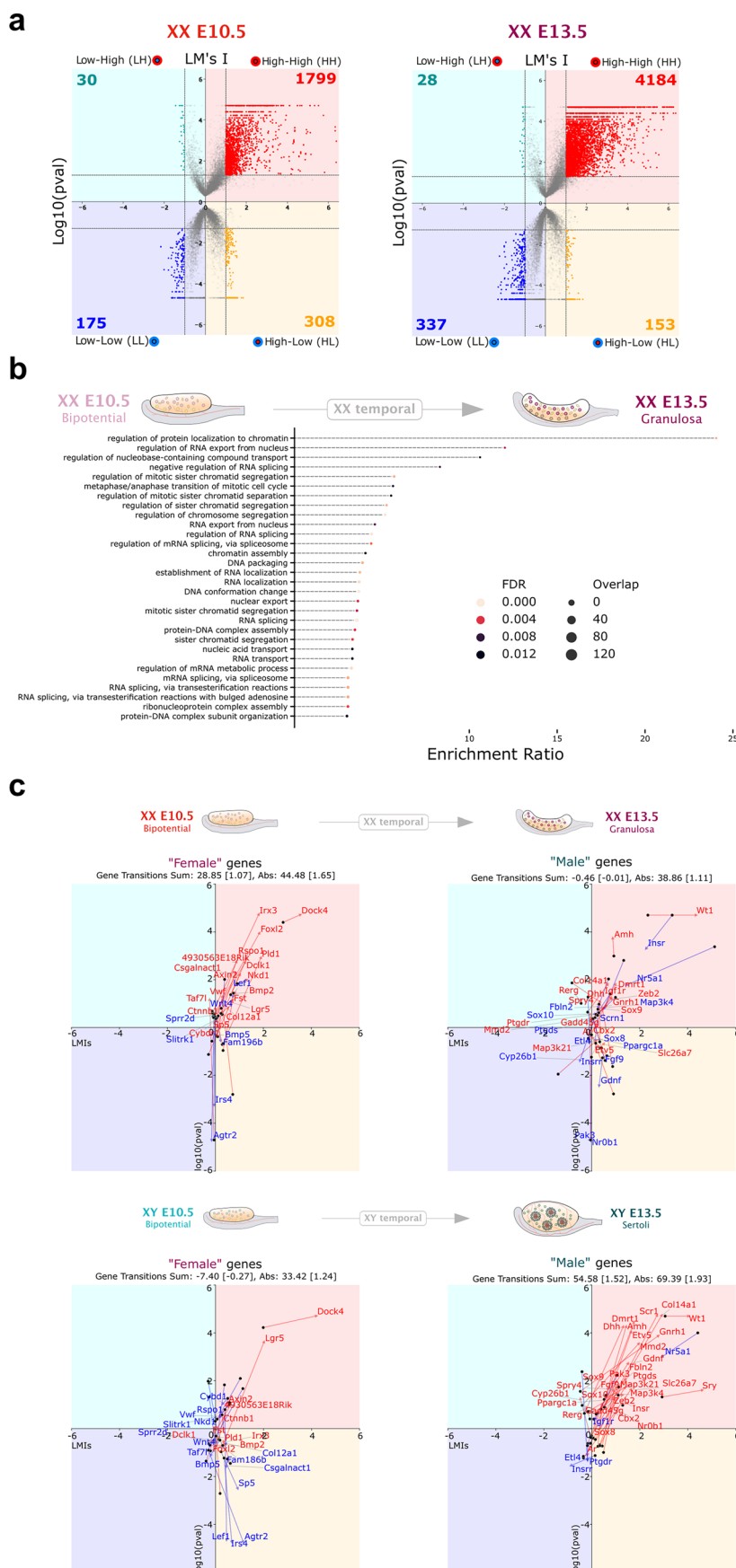

**Extended Data Fig. 5 | See next page for caption.**

**Extended Data Fig. 5 | Rewiring of 3D regulatory hubs during sex differentiation. a**. Changes in 3D regulatory hubs during granulosa cell differentiation. LMI quadrants for XX E10.5 (left) and XX E13.5 cells (right) for 24,027 annotated gene promoters in the mm10 reference genome. Quadrants include the total number of statistically significant genes in each quadrant. Note the increased numbers after differentiation in the HH and LL, denoting simultaneous activation and repression of genes during differentiation. P-values for LM's I were assessed using a Monte Carlo permutation test of spatial randomness. **b**. List of GO enrichment terms during sex determination in during granulosa cell differentiation. GO Biological Process enrichment analysis for genes that transition from LL or HL to HH during granulosa cell differentiation (XX E10.5 to XX E13.5). The list displays the 30 top terms retrieved from the analysis. GO-term enrichment significance assessed by a hypergeometric test,

with multiple testing corrections using Benjamini–Hochberg FDR. **c**. Gene transitions for male- and female-specific genes during sex determination in both sexes. LMI transition (or "gene transitions") for genes acquiring female and male-specific expression during sex differentiation for the differentiation of granulosa cells (XX E10.5 to XX E13.5) and Sertoli cells (XY E10.5 to XY E13.5). The plots show the four LMI quadrans (that is, HH, LH, LL, and HL in the LMI p-value and LM's I axis). Arrows indicate individual transition for each gene, which names are indicated at the arrowhead. Red arrows indicate tendency to transition towards H3K27ac environment while blue arrows indicate otherwise. Gene transition sum as well as the sum of absolute gene transitions are shown at the top of the graph together with mean values per gene shown between brackets. P-values for LM's I were assessed using a Monte Carlo permutation test of spatial randomness.

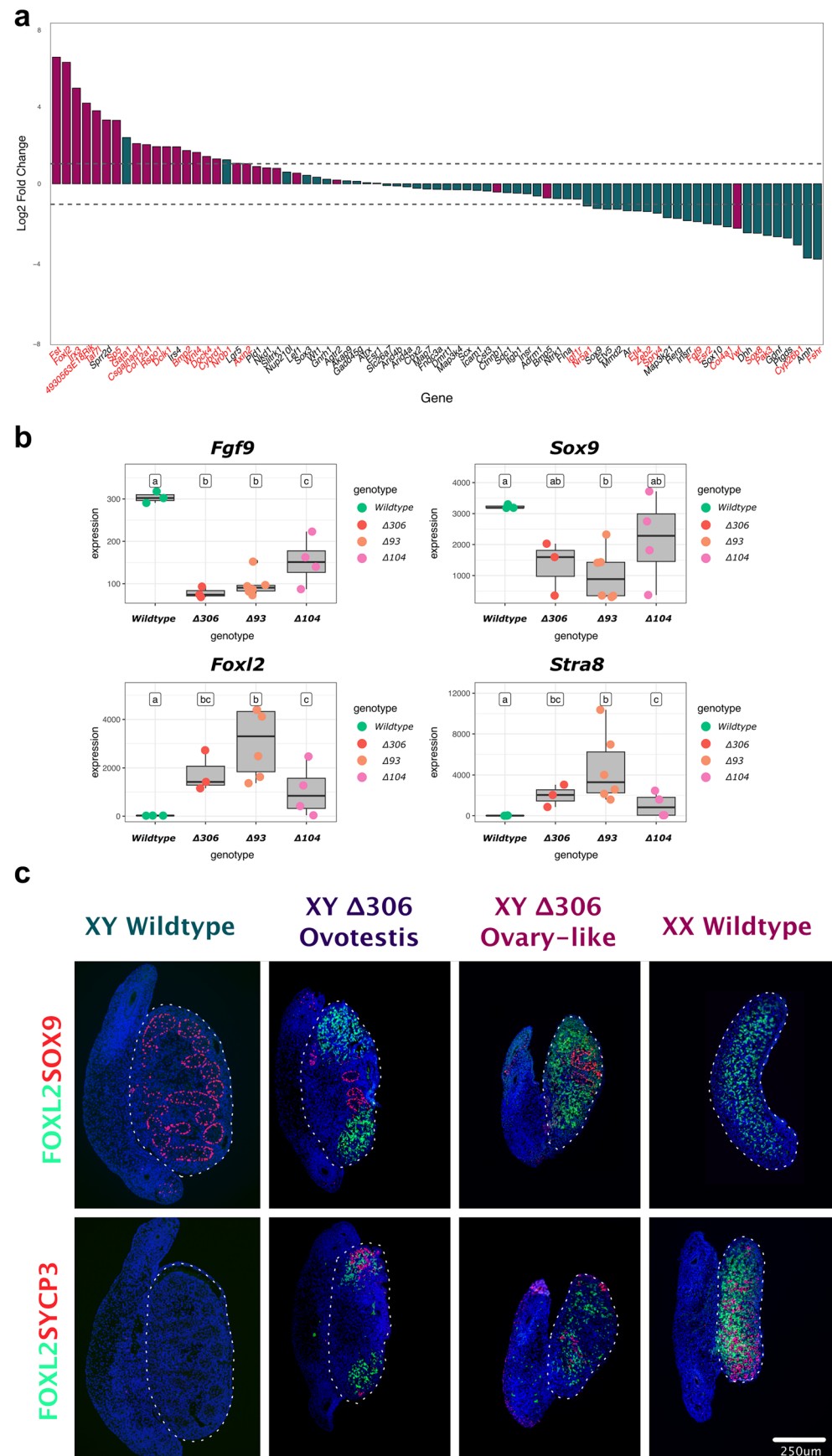

**Extended Data Fig. 6 | See next page for caption.**

**Extended Data Fig. 6 | Gene expression analyses on *Fgf9* mutants. a**. RNA-seq analysis on *Fgf9 Δ306* mutants. Bar plot for genes associated to female and male development[43]. Discontinuous line represents the threshold for genes to be considered as differentially expressed. Genes in red are those displaying significant p-values. **b**. Expression levels for selected genes in *Fgf9* mutants. Differential expression of *Fgf9*, as well as male (*Sox9*) and female (*Foxl2, Stra8*) markers between genotypes. Boxplots showing normalized DESeq2 expression between different genotypes. Expression values of individual samples are depicted with points colored according to the genotype. (WT n = 3, *Δ104* n = 4, *Δ93* n = 5 and *Δ306* n = 3) The results of pairwise Wald tests are shown using compact letter display. Briefly, the differences in expression of a particular gene across two genotypes are statistically indistinguishable if one letter is shared (fdr corrected p-values < 0.05). Boxplots display the median (center line), the interquartile range (IQR; 25th–75th percentiles), and whiskers extending to the datapoints within 1.5×IQR. Pairwise differential expression (log2FoldChange, padj). *Fgf9*: WT−*Δ104* (1.00, $1.75 \times 10^{-2}$), WT−*Δ306* (1.95, $4.96 \times 10^{-7}$), WT−*Δ93* (1.64, $2.84 \times 10^{-7}$), *Δ306 −Δ104* (− 0.95, 0.0618), *Δ306–Δ93* (− 0.31, 0.689), *Δ93–Δ104* (− 0.64, 0.103). *Foxl2*: WT−*Δ104* (− 5.46, $4.50 \times 10^{-7}$), WT−*Δ306* (− 6.22, $1.21 \times 10^{-8}$), WT−*Δ93* (− 7.01, $7.22 \times 10^{-15}$), *Δ306–Δ104* (0.75, 0.705), *Δ306–Δ93* (− 0.80, 0.717), *Δ93–Δ104* (1.55, 0.158). *Sox9*: WT−*Δ104* (0.58, 0.715), WT−*Δ306* (1.28, 0.357), WT−*Δ93* (1.65, 0.110), *Δ306–Δ104* (− 0.71, 0.716), *Δ306–Δ93* (0.37, 0.887), *Δ93–Δ104* (− 1.07, 0.325). *Stra8*: WT−*Δ104* (− 6.75, $8.39 \times 10^{-5}$), WT−*Δ306* (− 7.68, $4.75 \times 10^{-6}$), WT−*Δ93* (− 8.90, $1.26 \times 10^{-10}$), *Δ306–Δ104* (0.94, 0.767), *Δ306–Δ93* (− 1.22, 0.708), *Δ93–Δ104* (2.16, 0.190). **c**. Immunofluorescence analysis for sex-specific and meiotic markers for *Fgf9 Δ306* mutants (n ≥ 3). Upper panel shows immunofluorescence for the testis marker SOX9 (red) and the ovarian marker FOXL2 (green). Note the presence of both markers in the *Δ306* mutant gonads, which display two phenotypes: ovotestis and ovary-like. Lower panel show immunofluorescence of female marker FOXL2 (green) and meiosis marker SYCP3 (red), note of the presence of both markers in XY *Δ306* and XX wild type gonads (middle and right) and the absence of them in the XY wild type gonad (left). Gonads are delineated by a discontinuous line.

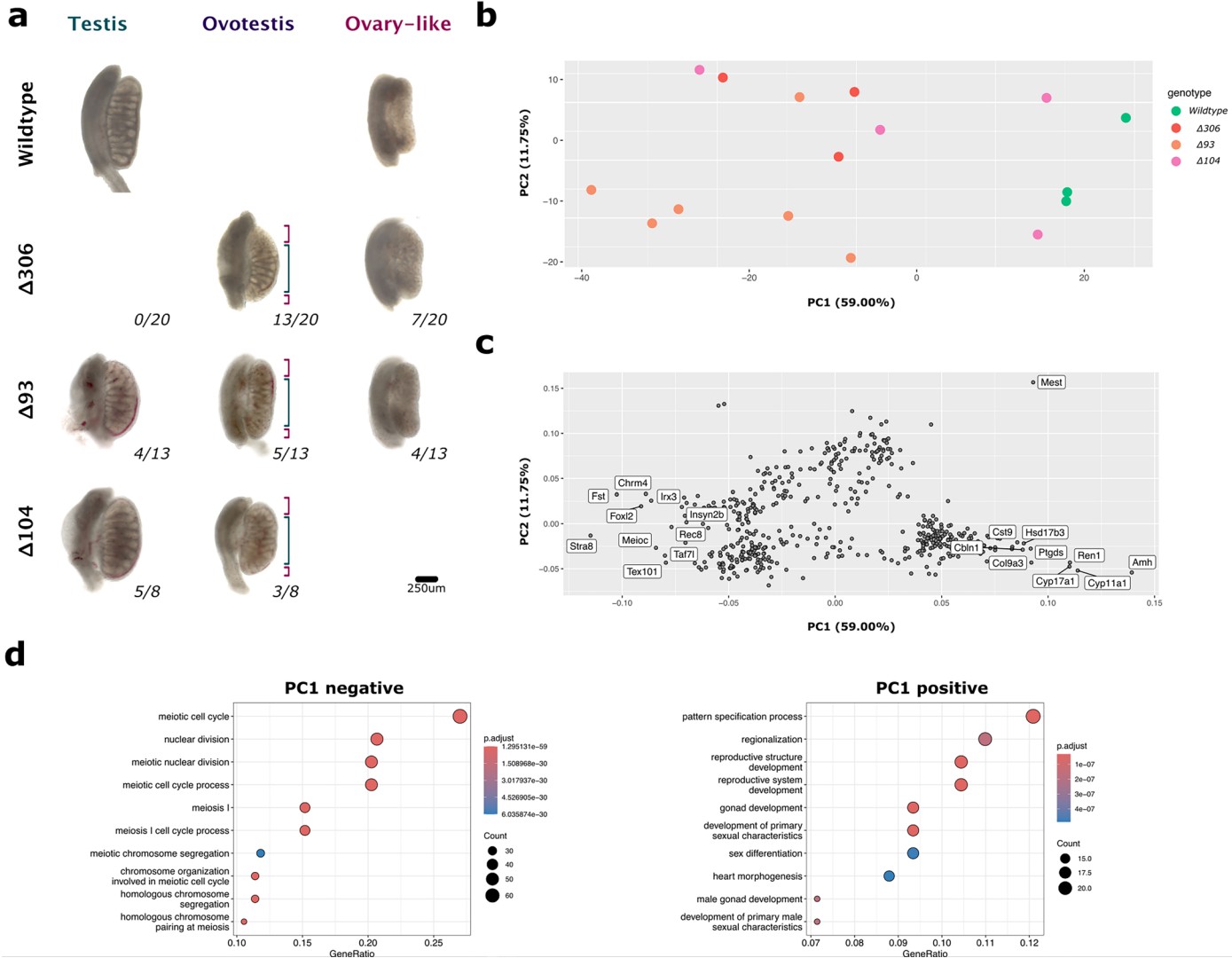

**Extended Data Fig. 7 | Phenotypical and transcriptomic variation of *Fgf9* mutants. a.** Phenotypical analysis of *Fgf9* mutants. First row displays control gonads of wildtype animals at E14.5 (testis are from XY and ovaries from XX individuals). Second to fourth rows show gonadal phenotypes for the different *Fgf9* mutants. Note the presence of three distinct phenotypes. The ovotestes phenotypes is characterized by the formation of testicular tissue in the center of the gonad (green bracket) and ovarian tissue at the poles (purple brackets). Phenotypic analyses suggest that sex-reversal phenotypes are more severe for *Δ306* (only ovotestes or ovary-like gonads), followed by *Δ93* (all three phenotypes) and then *Δ104* (ovotestes or testes) mutants. All mutant phenotypes correspond to XY individuals. The numbers below each gonadal picture represent the number of observed phenotypes out of the total number of analyzed gonads. **b.** Principal Component Analysis (PCA) for *Fgf9* mutants. DESeq2 PCA analysis of the different RNA-seq samples (default parameters with top 5000 variable genes used in PCA calculation). Different genotypes are depicted with different colors. **c.** Representation of genes that drive the observed variation. Relevant genes are indicated. **d.** Gene ontology analysis for genes that drive the variation. Note the enrichment in terms related to sex differentiation, such as meiosis or hormone biosynthesis. GO enrichment analysis was performed with the R package clusterProfiler (function enrichG) that employs a one-sided hypergeometric Fisher's exact test with Benjamini–Hochberg correction.

## a

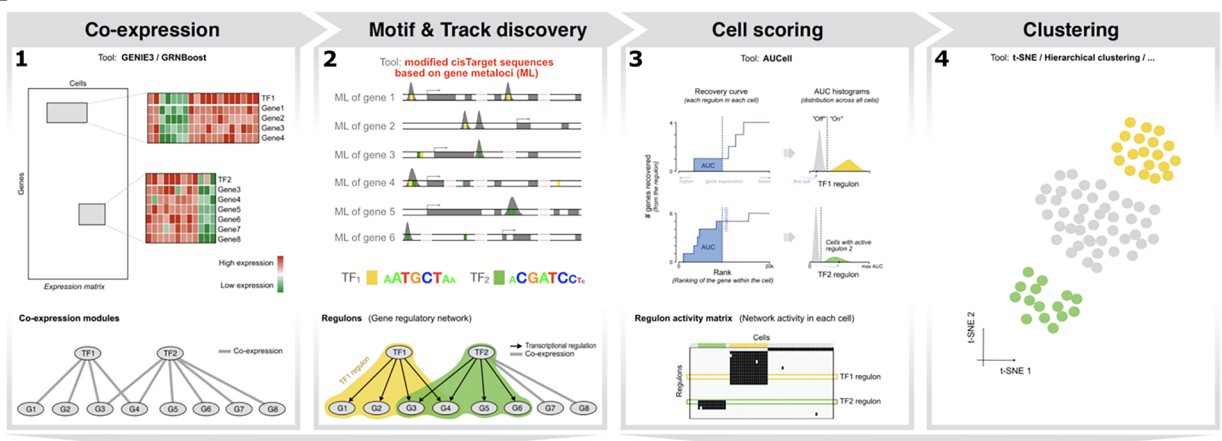

## b

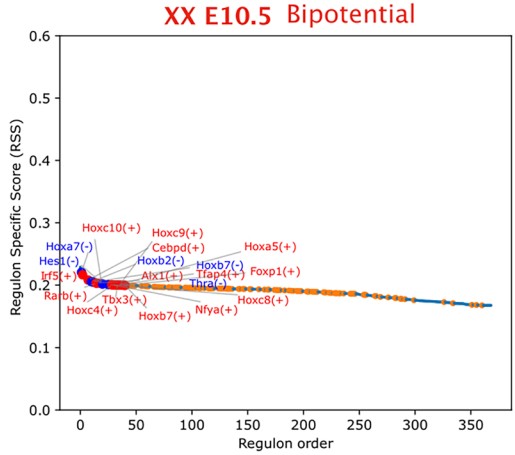

**(+) POSITIVE REGULONS**
Irf5, Cebpd, Alx1, Rarb, Hoxc10, Nfya, Hoxc4, Tbx3, Hoxa5, Tfap4

**(-) NEGATIVE REGULONS**
Hoxa7, Hoxb2, Hoxb7, Hes1, Thra, Maf, Nfia, Hoxd9, Junb, Emx2

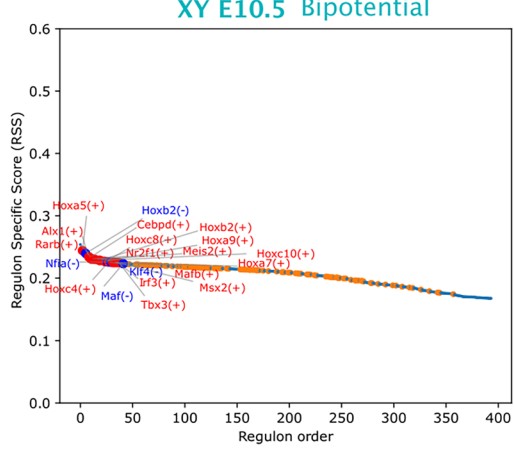

**(+) POSITIVE REGULONS**
Hoxa5, Alx1, Cebpd, Nr2f1, Rarb, **Meis2**, Hoxb2, Hoxc10, Hoxa9, Hoxc8

**(-) NEGATIVE REGULONS**
Hoxb2, Klf4, Nfia, Maf, Hoxd9, Gata4, Jund, Thra, Junb, Fosb

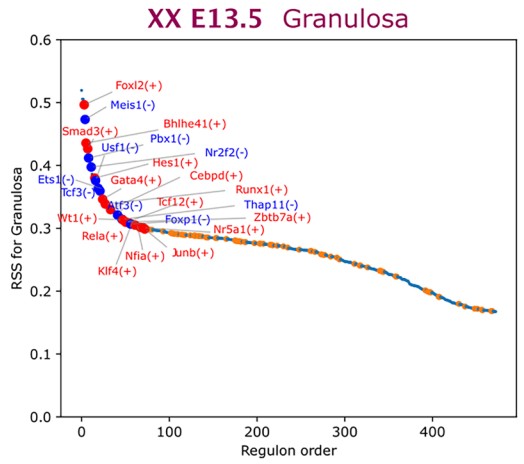

**(+) POSITIVE REGULONS**
Foxl2, Bhlhe41, Smad3, Hes1, Gata4, Runx1, Cebpd, Wt1, Zbtb7a, Tcf12

**(-) NEGATIVE REGULONS**
**Meis1**, Pbx1, Nr2f2, Usf1, Tcf3, Ets1, Atf3, Thap11, Foxp1, Thra

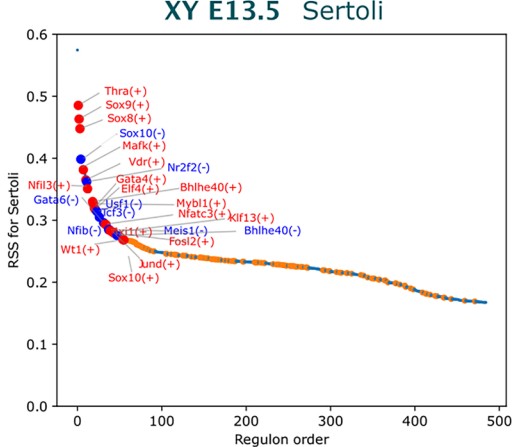

**(+) POSITIVE REGULONS**
Thra, Sox9, Sox8, Mafk, Vdr, Nfil3, Bhlhe40, Mybl1, Gata4, Elf4

**(-) NEGATIVE REGULONS**
Sox10, Nr2f2, Usf1, Tcf3, Gata6, **Meis1**, Nfib, Bhlhe40, Nfyb, Pbx1

**Extended Data Fig. 8** | See next page for caption.

**Extended Data Fig. 8 | Reconstruction of Gene Regulatory Networks using *METALoci* and SCENIC. a**. Figure adapted from the original Supplementary Fig. 1 in SCENIC paper[90]. Once Co-expression networks are built based on sc-RNA-seq data (1), the cisTarget program of the SCENIC package (2) was modified to accept as input gene targets that are not continuous in sequence. Those cisTarget regions per gene contained the HH metaloci identified by *METALoci*, which do not need to be continuous in sequence and very often include several sequences separated by hundreds of kilobases in the genome. Thae rest of the

SCENIC pipeline was not modified (3 and 4). **b**. Reconstruction of gene regulatory networks associated with sex determination. TOP 20 TF regulons for bipotential E10.5 XX, bipotential XY E10.5, granulosa E13.5 XX and Sertoli E13.5 XY cells. Regulons are ranked according to Regulon Specific Score (RSS), which is a metric for cell type specificity. Grey small points indicate regulons with mean < 0.3 normalized counts in scRNA-seq, which were discarded for further analysis. Below, the TOP 10 positive and negative regulons are indicated. Meis1 and Meis2, which are further investigated in this study, are indicated in blue and purple.

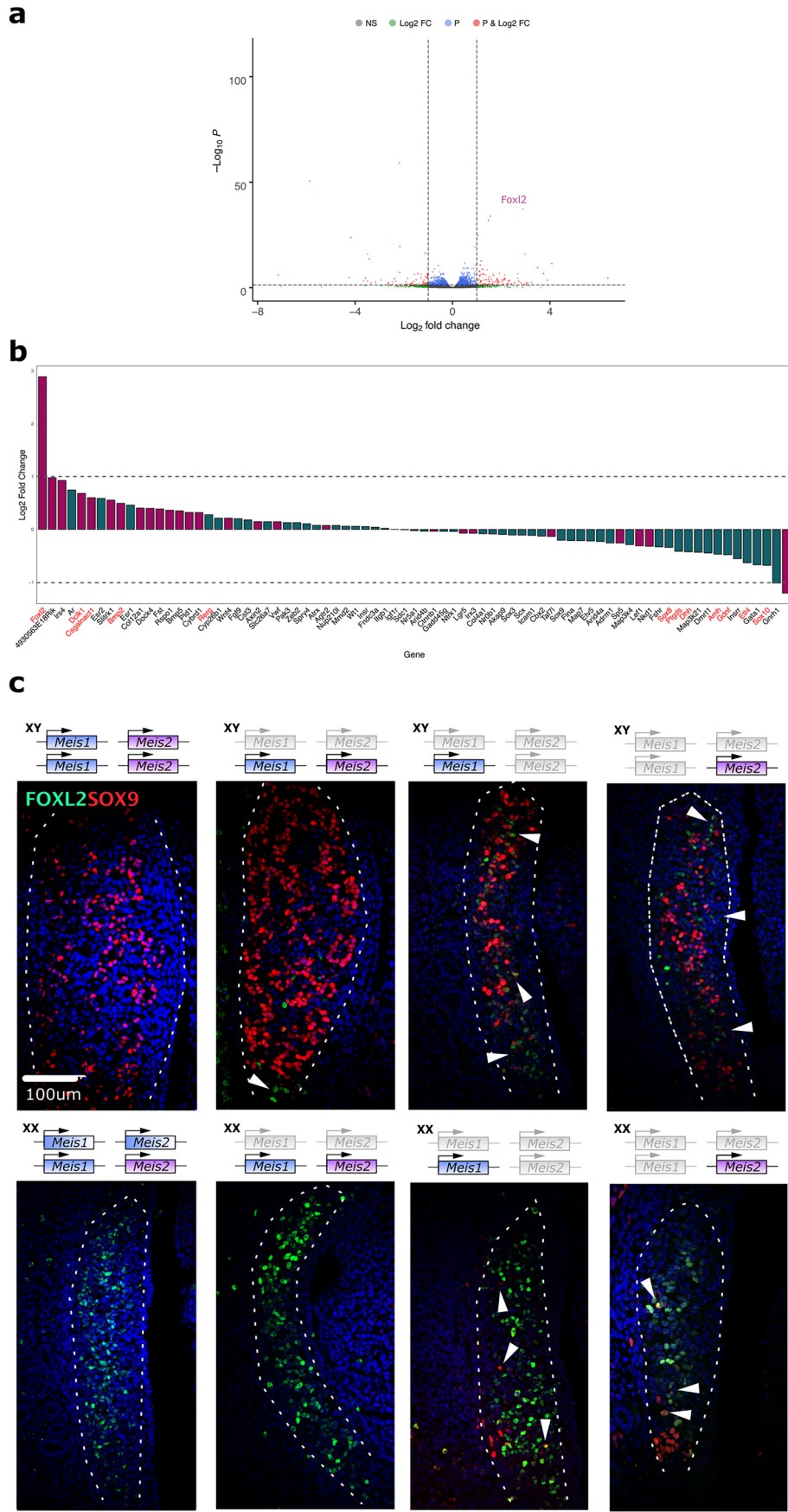

**Extended Data Fig. 9 | See next page for caption.**

**Extended Data Fig. 9 | Gene expression analysis on *Meis* mutants. a**. RNA-seq analysis on *Meis1* mutants. Volcano plot of RNA-seq from XY E13.5 *Meis1* mutant and control gonads. From a list of associated to female and male development[43], only *Foxl2* appears as upregulated. Differential expression was assessed using DESeq2 (two-sided Wald test with Benjamini–Hochberg adjusted p-values. **b**. Bar plot for genes associated with associated to female and male development[43]. Discontinuous line represents the threshold for genes to be considered as differentially expressed. Genes in red are those displaying significant p-values. **c**. Immunofluorescence analysis on Meis knockout mutants. Immunofluorescence on XY E12.5 *Meis* mutants and controls (n ≥ 3). The ovarian marker FOXL2 (green) marks the presence of granulosa cells. Left. Testicular marker SOX9 (red) indicates the presence of Sertoli cells. Note the presence of FOXL2 positive cells at the gonadal poles (arrowheads) in XY mutants with two deleted Meis alleles. Note also the increase in number of FOXL2 positive cells and its distribution across the entire gonads in mutants with three deleted *Meis* alleles (*Meis1* homozygous; *Meis2* heterozygous or *Meis1* heterozygous; *Meis2* homozygous). XX mutants also display sex reversal upon deletion of three *Meis* alleles, as indicated by the presence of SOX9 positive cells (arrows).

# a

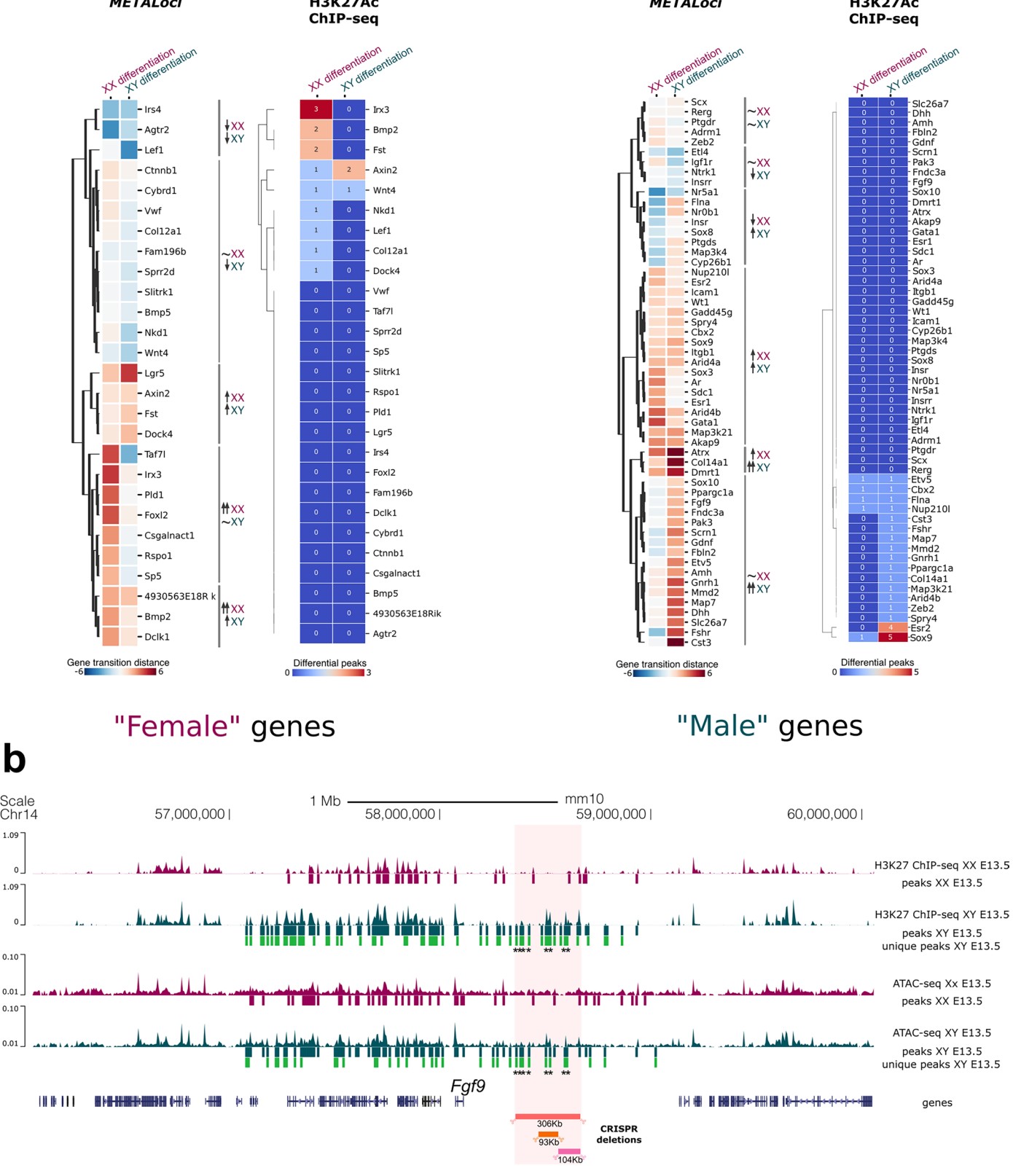

**Extended Data Fig. 10 | See next page for caption.**

**Extended Data Fig. 10 | Comparison between *METALoci* and other approaches for enhancer identification. a**. Comparison between *METALoci* and a chromatin loop-based approach for genes displaying female-biased or male-biased expression[43] during XX granulosa differentiation and XY Sertoli differentiation. *METALoci* analysis shows individual gene transitions for each female and male-specific genes during sex-determination. Genes are grouped by unsupervised clustering based on gene transitions from E10.5 to E13.5 in female and male differentiation (see also Fig. 3). H3K27Ac ChIP-seq analysis shows the number for differential peaks that overlap with a chromatin loop that has a gene promoter in the other anchor. Note that the loop approach does not detect any differentially "looped" H3K27Ac ChIP-seq peak for most of the studied genes. **b**. H3K27ac ChIP-seq and ATAC-seq differential peak analysis between Sertoli and granulosa cells at embryonic day 13.5 (E13.5). The figure displays the entire 4 Mb region surrounding the *Fgf9* gene (chr14:56,070,000–60,060,000), with the gene promoter positioned at the center of the visualized region. This genomic window corresponds to the one used in the *METALoci* analysis. However, the differential peak analysis focuses specifically on regions that locate ±1 Mb from the *Fgf9* transcription start site. Unique peaks in Sertoli cells are marked with green rectangles. Asterisks indicate 8 unique peaks overlapping between H3K27Ac ChIP-seq and ATAC-seq data signal, within the 306 Kb region that was deleted in transgenic mice.

| | |
|---|---|

# Reporting Summary

## Statistics

For all statistical analyses, confirm that the following items are present in the figure legend, table legend, main text, or Methods section.

| n/a | Confirmed | |
|---|---|---|
| ☐ | ☒ | The exact sample size (*n*) for each experimental group/condition, given as a discrete number and unit of measurement |
| ☒ | ☐ | A statement on whether measurements were taken from distinct samples or whether the same sample was measured repeatedly |
| ☐ | ☒ | The statistical test(s) used AND whether they are one- or two-sided<br>*Only common tests should be described solely by name; describe more complex techniques in the Methods section.* |
| ☒ | ☐ | A description of all covariates tested |
| ☒ | ☐ | A description of any assumptions or corrections, such as tests of normality and adjustment for multiple comparisons |
| ☐ | ☒ | A full description of the statistical parameters including central tendency (e.g. means) or other basic estimates (e.g. regression coefficient) AND variation (e.g. standard deviation) or associated estimates of uncertainty (e.g. confidence intervals) |
| ☐ | ☒ | For null hypothesis testing, the test statistic (e.g. *F*, *t*, *r*) with confidence intervals, effect sizes, degrees of freedom and *P* value noted<br>*Give P values as exact values whenever suitable.* |
| ☒ | ☐ | For Bayesian analysis, information on the choice of priors and Markov chain Monte Carlo settings |
| ☒ | ☐ | For hierarchical and complex designs, identification of the appropriate level for tests and full reporting of outcomes |
| ☒ | ☐ | Estimates of effect sizes (e.g. Cohen's *d*, Pearson's *r*), indicating how they were calculated |

*Our web collection on statistics for biologists contains articles on many of the points above.*

## Software and code

Policy information about availability of computer code

| Data collection | No software was used for data collection |
|---|---|
| Data analysis | Most analysis were performed with standard packages for NGS analysis and/or with custom python scripts.<br><br>New code:<br>- METALoci in house developed Python 3 code available as a Jupyter notebook in GitHub (https://github.com/3DGenomes/METALoci).<br><br>The following public software was used for data analysis:<br><br>Bowtie2 version= 2.3.4.3<br>samtools versions= 1.9<br>STAR version= 2.7.10a<br>MACS2 version=2.2.7.1<br>GeoPandas (https://geopandas.org, 0.10.2)<br>NetworkX (https://networkx.org, 2.6.3)<br>libpysal (https://pysal.org, 4.6.2)<br>ESDA (https://pysal.org/esda/, 2.4.1)<br>pyBigWig library from deepTools (https://deeptools.readthedocs.io, 0.3.18).<br>flexbar version=3.4<br><br>R version=4.0.2 |

```
featureCounts version= v2.0.1
DEseq2 version= 1.28.1
ggplot2 version=3.3.6
reshape version=0.8.8
biomaRt version=2.56.3
PiGX version=0.1.1
Trim Galore! version= 0.6.5

python version= 3.7.9
fanc version=0.9.8
numpy version=1.21.6
seaborn version=0.11.2
pandas version=  1.3.5
pybedtools version=0.8.1
matplotlib version= 3.5.2
juicer version=1.5.6
deeptools version =  3.4.3
bedtools version= 2.3
SciPy  version= 1.9.3

pySCENIC version= 0.12.1
```

For manuscripts utilizing custom algorithms or software that are central to the research but not yet described in published literature, software must be made available to editors and reviewers. We strongly encourage code deposition in a community repository (e.g. GitHub). See the Nature Portfolio guidelines for submitting code & software for further information.

# Data

Policy information about availability of data

All manuscripts must include a data availability statement. This statement should provide the following information, where applicable:
- Accession codes, unique identifiers, or web links for publicly available datasets
- A description of any restrictions on data availability
- For clinical datasets or third party data, please ensure that the statement adheres to our policy

Raw data were submitted to GEO under the accession number: GSE217618 and they are available for the public

# Research involving human participants, their data, or biological material

Policy information about studies with human participants or human data. See also policy information about sex, gender (identity/presentation), and sexual orientation and race, ethnicity and racism.

| Reporting on sex and gender | Use the terms sex (biological attribute) and gender (shaped by social and cultural circumstances) carefully in order to avoid confusing both terms. Indicate if findings apply to only one sex or gender; describe whether sex and gender were considered in study design; whether sex and/or gender was determined based on self-reporting or assigned and methods used. Provide in the source data disaggregated sex and gender data, where this information has been collected, and if consent has been obtained for sharing of individual-level data; provide overall numbers in this Reporting Summary. Please state if this information has not been collected. Report sex- and gender-based analyses where performed, justify reasons for lack of sex- and gender-based analysis. |
|---|---|
| Reporting on race, ethnicity, or other socially relevant groupings | Please specify the socially constructed or socially relevant categorization variable(s) used in your manuscript and explain why they were used. Please note that such variables should not be used as proxies for other socially constructed/relevant variables (for example, race or ethnicity should not be used as a proxy for socioeconomic status). Provide clear definitions of the relevant terms used, how they were provided (by the participants/respondents, the researchers, or third parties), and the method(s) used to classify people into the different categories (e.g. self-report, census or administrative data, social media data, etc.) Please provide details about how you controlled for confounding variables in your analyses. |
| Population characteristics | Describe the covariate-relevant population characteristics of the human research participants (e.g. age, genotypic information, past and current diagnosis and treatment categories). If you filled out the behavioural & social sciences study design questions and have nothing to add here, write "See above." |
| Recruitment | Describe how participants were recruited. Outline any potential self-selection bias or other biases that may be present and how these are likely to impact results. |
| Ethics oversight | Identify the organization(s) that approved the study protocol. |

Note that full information on the approval of the study protocol must also be provided in the manuscript.

# Field-specific reporting

Please select the one below that is the best fit for your research. If you are not sure, read the appropriate sections before making your selection.

☒ Life sciences    ☐ Behavioural & social sciences    ☐ Ecological, evolutionary & environmental sciences

For a reference copy of the document with all sections, see nature.com/documents/nr-reporting-summary-flat.pdf

# Life sciences study design

All studies must disclose on these points even when the disclosure is negative.

| | |
|---|---|
| Sample size | Sample size were 2 biological replicates for each sample in Hi-C experiment. 2-4 technical replicates from each biological replicate were performed.<br>RNA-seq and immunofluorescence analysis of mutant gonads and WT have more than 3 biological replicates. |
| Data exclusions | Samples were excluded only according to the genotype assessed in control experiments. |
| Replication | All experiments were performed in at least two biological replicates and good reproducibility was assessed by Pearson correlation. |
| Randomization | Randomization was not relevant for this study because all the experiments done had to take in account the genotype and the sex of the sample, and there is no treatment involved or additional covariates expected to introduce biases in development. |
| Blinding | Blinding was not relevant for our study, since all comparisons were performed automatically using statistical software that is not influenced by the investigator. |

# Reporting for specific materials, systems and methods

We require information from authors about some types of materials, experimental systems and methods used in many studies. Here, indicate whether each material, system or method listed is relevant to your study. If you are not sure if a list item applies to your research, read the appropriate section before selecting a response.

## Materials & experimental systems

| n/a | Involved in the study |
|---|---|
| ☐ | ☒ Antibodies |
| ☐ | ☒ Eukaryotic cell lines |
| ☒ | ☐ Palaeontology and archaeology |
| ☐ | ☒ Animals and other organisms |
| ☒ | ☐ Clinical data |
| ☒ | ☐ Dual use research of concern |
| ☒ | ☐ Plants |

## Methods

| n/a | Involved in the study |
|---|---|
| ☒ | ☐ ChIP-seq |
| ☐ | ☒ Flow cytometry |
| ☒ | ☐ MRI-based neuroimaging |

## Antibodies

| | |
|---|---|
| Antibodies used | SOX9 (Merck millipore, AB5535, Lot : 3282152, 1:600)<br>FOXL2 (abcam, ab5096-100ug, Lot:GR270487-1, 1:150)<br>SCYP3 (abcam, ab15093, Lot: GR302438-2, 1:200 )<br>Alexa Fluor 488 donkey anti goat IgG (life technologies, A11055, Lot:2134018, 1:200)<br>Alexa Fluor 555 donkey anti rabbit IgG (life technologies, A31572, Lot: 2088692, 1:200) |
| Validation | Primary antibodies<br>SOX9:<br>Anti-Sox9 Antibody is a well characterized affinity purified Rabbit Polyclonal Antibody that reliably detects Transcription Factor Sox-9. This highly published antibody has been validated in IHC & WB<br>FOXL2:<br>Goat polyclonal to FOXL2, host species:goat, tested applications WB, postivie control WB: K562 and HeLa cell lysates, species reactivity: Human but predicted to work on mouse, rat and cow<br>SCYP3:<br>Rabbit polyclonal to SCP3, Suitable for: IHC-P, ICC/IF,Reacts with: Mouse, Human,  Synthetic peptide (Human) (C terminal), positive control: mouse testis tissue sections, IHC-P<br><br>Secondary antibodies:<br>Alexa Fluor 488 donkey anti goat IgG:<br>Applications: IHC, ICC/IF, Flow. Species reactivity:Goat, Host :Donkey IgG, Class: polyclonal, Type:Secondary antibody. Immunogen: Gamma immunoglobins heavy and light chains. Conjugate: Alexa Fluor 488 |

Alexa Fluor 555 donkey anti rabbit IgG:
Applications: IHC, ICC/IF, Flow. Species reactivity:Rabbit, Host :Donkey IgG, Class: polyclonal, Type:Secondary antibody. Immunogen: Gamma immunoglobins heavy and light chains. Conjugate: Alexa Fluor 555

# Eukaryotic cell lines

Policy information about cell lines and Sex and Gender in Research

| Cell line source(s) | G4F1 (https://doi.org/10.1073/pnas.0609277104)<br>G4F1/del306 (generated in this study)<br>G4F1/del93 (generated in this study)<br>G4F1/del104 (generated in this study) |
|---|---|
| Authentication | Cell lines were not authenticated |
| Mycoplasma contamination | All cells were tested for mycoplasma contamination using Mycoalert detection kit (Lonza) and Mycoalert assay control set (Lonza) |
| Commonly misidentified lines<br>(See ICLAC register) | No commonly misidentified cell lines were used |

# Animals and other research organisms

Policy information about studies involving animals; ARRIVE guidelines recommended for reporting animal research, and Sex and Gender in Research

| Laboratory animals | Mus musculus CD-1, female animals from various ages for donor and embryos retransfered by tetraploid aggregation and E13.5 embryos isolated to perform experimental analysis.<br>Nr5a1-eGFP on a C57BL/6 (B6) background to FACS sort cell populations for  Hi-C experiments (reference stated in methods)<br>Sox9-eCFP on a C57BL/6 (B6) background to FACS sort cell populations for  Hi-C experiments (reference stated in methods)<br>Runx1-GFP also in C57BL/6 (B6) background to FACS sort cell populations for Hi-C experiments (reference stated in methods)<br>Meis1flox on a C57BL/6 (B6) background to obtain Meis1/Meis2 conditional knockout embryos (reference stated in methods)<br>Meis2flox on a C57BL/6 (B6) background to obtain Meis1/Meis2 conditional knockout embryos (reference stated in methods)<br>Stra8Cre on a C57BL/6 (B6) background to obtain Meis1/Meis2 conditional knockout embryos (reference stated in methods)<br>Zp3Cre on a C57BL/6 (B6) background to obtain Meis1/Meis2 conditional knockout embryos (reference stated in methods) |
|---|---|
| Wild animals | No wild animals were used in this study |
| Reporting on sex | Sex was considered for the study. The embryos from Sox9-eCFP line were males and the ones from Runx1-GFP females, since we were interested in the populations from testis and ovaries respectively. |
| Field-collected samples | This study did not involve field-collected samples |
| Ethics oversight | Mice used for tetraploid complementation assays were handled according to institutional guidelines under an experimentation licence (G0111/17 and G0051/22) approved by the Landesamt fuer Gesunheit und Soziales (Berlin, Germany), or according to the Spanish law and EU Directive 2010/63/EU, with approvals from the University Pablo de Olavide Ethics Committee and the Junta de Andalucía, with reference 09/02/2024/024.<br><br>Reporter mice used for FACS of the cell populations of the gonad were handled in accordance with National Institutes of Health guidelines and with the approval of the Duke University Medical Center Institutional Animal Care and Use Committee  (A089-20-04 9N).<br><br>Mice used for the generation of the Cre/lox lines were handled in accordance with CNIC Ethics Committee, Spanish laws and the EU Directive 2010/63/EU for the use of animals in research. All mouse experiments were approved by the CNIC and Universidad Autónoma de Madrid Committees for 'Ética y Bienestar Animal' and the area of 'Protección Animal' of the Community of Madrid with reference PROEX 220/15. |

Note that full information on the approval of the study protocol must also be provided in the manuscript.

# Plants

| | |
|---|---|
| Seed stocks | *Report on the source of all seed stocks or other plant material used. If applicable, state the seed stock centre and catalogue number. If plant specimens were collected from the field, describe the collection location, date and sampling procedures.* |
| Novel plant genotypes | *Describe the methods by which all novel plant genotypes were produced. This includes those generated by transgenic approaches, gene editing, chemical/radiation-based mutagenesis and hybridization. For transgenic lines, describe the transformation method, the number of independent lines analyzed and the generation upon which experiments were performed. For gene-edited lines, describe the editor used, the endogenous sequence targeted for editing, the targeting guide RNA sequence (if applicable) and how the editor was applied.* |
| Authentication | *Describe any authentication procedures for each seed stock used or novel genotype generated. Describe any experiments used to assess the effect of a mutation and, where applicable, how potential secondary effects (e.g. second site T-DNA insertions, mosiacism, off-target gene editing) were examined.* |

# Flow Cytometry

## Plots

Confirm that:

☒ The axis labels state the marker and fluorochrome used (e.g. CD4-FITC).

☒ The axis scales are clearly visible. Include numbers along axes only for bottom left plot of group (a 'group' is an analysis of identical markers).

☐ All plots are contour plots with outliers or pseudocolor plots.

☒ A numerical value for number of cells or percentage (with statistics) is provided.

## Methodology

| | |
|---|---|
| Sample preparation | The Sf1-eGFP (Nr5a1-eGFP) and Sox9-eCFP reporter mouse lines previously generated were maintained on a C57BL/6 (B6) background64,65. The Runx1-GFP reporter mouse was generously gifted by Dr. Humphrey Yao at NIEHS20 (availability at MMRRC_010771-UCD). Timed matings were generated with reporter males and wild-type CD1 females. The morning of a vaginal plug was considered E0.5. Embryos were collected at E10.5 and E13.5, with genetic sex determined using PCR for the presence or absence of the Y-linked gene Uty. Gonads were dissected from E10.5 or E13.5 embryos, and the mesonephros was removed using syringe tips. The gonads were incubated in 500ul 0.05% trypsin for 6-10 min at 37°C, then mechanically disrupted in 1X PBS/10% FCS. The cell suspension was pipetted through a 40µm filter top and the supporting cells were collected with FACS. |
| Instrument | GFP-positive cells were sorted using a Becton Dickinson (BD) DiVa and analyzed with the BD Fortessa X-20. CFP-positive cells were sorted using a B-C Astrios and analyzed with the BD Fortessa X-20 |
| Software | Data was analyzed using the BD FACSDiVa software |
| Cell population abundance | Post-sort purity was determined to be greater than 97% by re-analyzing the post-sort fraction by FACS. Cell population abundance was on average as follows: 2.8% for E10.5 XX, 2.5% for E10.5 XY, 30% for E13.5 XX, and 20% for E13.5 XY. |
| Gating strategy | Cells were gated on an FSC-A and SSC-A plot to select for cells and eliminate debris. Cells were gated on an FSC-H and FSC-A plot to select for singlets and eliminate doublets. Cells were gated on an SSC-A and GFP-A/CFP UV-A plot to identify fluorescent cells of interest. The boundaries between fluorescent and non-fluorescent populations were defined each sort based on a negative control brought with the sample. Gates were established based on the negative control, allowing fluorescent cells to be identified as outside of the negative gates. |

☒ Tick this box to confirm that a figure exemplifying the gating strategy is provided in the Supplementary Information.

