## [Peer Review File · Nature Structural & Molecular Biology]

Chromatin spatial analysis by METALoci unveils sex-determining 3D regulatory hubs

Corresponding Author: Dr Darío Lupiáñez

Version 0:

Decision Letter:

29th May 2025

Dear Dr. Lupiáñez,

Thank you again for submitting your manuscript "Sex-determining 3D regulatory hubs revealed by genome spatial auto-correlation analysis". I apologise for the delay in responding, which resulted from the difficulty in timely obtaining and discussing suitable referee reports. Nevertheless, we now have comments (below) from the 3 reviewers who evaluated your paper. In light of these reports, we remain interested in your study and would like to see your response to the comments of the referees, in the form of a revised manuscript.

You will see that all three reviewers appreciate the potential conceptual novelty and resource impact imparted by the study, but they nevertheless raise numerous issues which must be addressed in a revised manuscript. More specifically, both Reviewer #1 and #3 request further analytical benchmarking of METALoci to existing methods, which we deem as a paramount point, whereas Reviewer #2 requests for further experimental investigation on the roles of Meis1/2, which would significantly boost the functional value of the manuscript. In addition to these points, we strongly encourage you to address all non-textual requests of the referees with additional analyses and experiments to strengthen the work. Moreover, the experts request further discussion at places, figure improvement, clarifications and addressing perceived inconsistencies.

We expect to see your revised manuscript within 3-4 months. If you cannot send it within this time, please contact us to discuss an extension; we would still consider your revision, provided that no similar work has been accepted for publication at NSMB or published elsewhere.

Reporting Summary:

When submitting the revised version of your manuscript, please pay close attention to our [href="https://www.nature.com/nature-portfolio/editorial-policies/image-integrity">Digital Image Integrity Guidelines. and to the following points below:](https://www.nature.com/nature-portfolio/editorial-policies/image-integrity)

EXTENDED DATA FIGURES

Data availability: this journal strongly supports public availability of data. All data used in accepted papers should be available via a public data repository, or alternatively, as Supplementary Information. If data can only be shared on request, please explain why in your Data Availability Statement, and also in the correspondence with your editor. Please note that for some data types, deposition in a public repository is mandatory - more information on our data deposition policies and available repositories can be found below:

<https://www.nature.com/nature-research/editorial-policies/reporting-standards#availability-of-data>

Link Redacted

Sincerely,

Dimitris Typas
Senior Editor
Nature Structural & Molecular Biology
ORCID: 0000-0002-8737-1319

Reviewers' Comments:

Reviewer #1 (Remarks to the Author):

Overview:

This study identifies new genes and distal enhancers involved in mouse gonadal sex determination, aiming to improve our understanding of transcriptional regulation in this process and its relevance to human disorders. The authors generated Hi-C maps from FACS-purified bipotential progenitors (E10.5 XX and XY embryos) and differentiated Sertoli or granulosa cells (E13.5 embryos). Conventional Hi-C analyses showed minimal TAD-level changes (Fig. 1). To detect higher-order 3D

genome changes, the authors developed METALoci, a computational tool combining Hi-C and H3K27ac ChIP-seq data to identify active 3D regulatory hubs (“metaloci”) with coordinated H3K27ac gains during gonad differentiation. METALoci confirmed known distal enhancer-promoter interactions at Sox9 (Fig. 2). When applied to all genes, METALoci identified new genes that engage in active 3D hubs upon Sertoli or granulosa cell differentiation, expanding the list of candidate genes involved in sex determination (Fig. 3). Running METALoci on simulated genomic deletions uncovered a broad region containing new Fgf9 distal enhancers, validated in vivo via large deletions in mice that resulted in partial Fgf9 downregulation and sex reversal phenotypes (Fig. 4). Finally, the authors integrated METALoci analysis with the published SCENIC pipeline and published ATAC-seq and scRNA-seq datasets to identify transcription factors (TFs) enriched at metaloci. This analysis identified candidate TFs that may control gonad differentiation, including Meis1/2 whose deletion caused sex differentiation defects, uncovering a novel role for these TFs (Fig. 5).

Key strengths of this study include the development of METALoci, the validation of a new Fgf9 regulatory region required for sex determination (but not other Fgf9 functions), and the identification of new transcriptional regulators of sex determination like Meis1/2. The engineered mouse models showed convincing sex reversal phenotypes. The authors argue that their new computational approaches, which are outside the scope of my expertise, accelerated the discovery of relevant genes and regulatory elements compared to traditional genetic approaches. The findings are relevant to a broad audience interested in developmental gene regulation and sex determination.

Limitations are summarized in my major comments below:

1. It was not fully clear how METALoci compares to existing methods in pinpointing enhancers, such as performing differential analysis of H3K27ac ChIP-seq or ATAC-seq peaks between undifferentiated versus differentiated gonads. In the case of Fgf9, METALoci identified a region of hundreds of kb as containing a candidate enhancer, as opposed to precisely pinpointed candidate enhancer peaks.
2. It was not fully clear which H3K27ac ChIP-seq datasets were used for METALoci analysis. It seems essential for the analysis that the ChIP-seq datasets were generated at the same developmental time points and cell types analyzed here by Hi-C, but this was not explicitly stated and should be clarified.

Minor comments:

1. Lines 188-192: The text can appear contradictory. “Metaplot analysis revealed that insulation at boundary regions increased during the transition from bipotential to the differentiated stage” suggests a global effect on TAD boundaries. “Pairwise comparisons revealed that only 1.49-1.84% of TAD boundaries changed their insulation significantly” instead describes a negligible effect that should not be visible in a metaplot analysis.
2. Line 218: I was not convinced that METALoci “quantif(ies) gene regulation”.
3. Paragraph starting at line 333: There is another apparent contradiction in the text: “During the differentiation from bipotential to granulosa cells, most female-specific genes acquired an active regulatory environment (...)” This sentence seems to contradict the following one: “The majority of male genes (46 out of 55) gain an active regulatory environment during Sertoli cell differentiation. In contrast, this mechanism was not as common for female-specific genes during granulosa cell differentiation (14 out of 27)”. 14 out of 27 is not accurately described as “most female-specific genes”.
4. Line 427: Explain that mice were derived from the engineered ESCs, for better flow in the text.
5. Line 473: Could the authors narrow down candidate Fgf9 enhancers by differential analysis of H3K27ac ChIP-seq peaks between Sertoli and granulosa cells?
6. Fig. 4A: The reader would benefit from more extensive explanations on how to interpret the METALoci analysis output. What do yellow and blue lines under the Hi-C maps represent? How should the values of LM’s I be interpreted?
7. Fig. 4A: Please show H3K27ac ChIP-seq data for E10.5.
8. Line 495: The reader would benefit from a clearer conceptual explanation (potentially a schematic) of how METALoci and SCENIC pipelines were integrated for the analysis.
9. Line 496: Please specify what kind of cells were analyzed in the published scRNA-seq dataset.
10. Fig. 5B: please label the cell types above the plots and provide a color legend for red/blue color coding.
11. A general comment is that the main text would benefit from proofreading to correct minor mistakes throughout.

Reviewer #2 (Remarks to the Author):

In this very interesting paper, the authors performed a comprehensive analysis of the changes in 3D chromatin structure and chromatin interactions characterizing the supporting cell lineage of the mouse gonads of both sexes, across the window of sex determination. Using a combination of innovative epigenomics techniques, novel in-silico approaches, mouse transgenesis to validate their findings and analysis of multiple deposited datasets beyond those produced within this study, the paper illustrates the following results:

- 1) When comparing the two critical timepoints of E10.5 (before sex determination) and E13.5 (after) between the female (XX) and male (XY) sexes by Hi-C, TAD structure is quite conserved across this developmental window, indicating that the conformation of the chromatin in the gonads is established before sex determination and maintained afterwards. Despite this stability in overall TAD organization, analysis of switches between compartments (with A enriched for euchromatic active regions and B representing the heterochromatic inactive regions), revealed that a number of genomic regions switch compartments after sex determination, even though this does not result into changes in gene expression, but only in a changed epigenetic status.
- 2) A new analysis tool was developed called METALoci. This method allowed the authors to use Hi-C data to build maps of interactions within specific loci and then map epigenomics data namely ChIP-Seq for histone marks onto these maps to visually assess recruitment of regulatory regions to a specific gene promoter. A nice example which validates this method is provided for the Enh13 and TESCO enhancers of Sox9, and for BMP2 for the female pathway amongst others. The tool was

then applied to each gene of the mouse genome showing a similar level or rewiring of regulatory regions in both sexes (Extended Figure 10 and Figure 3). However, when focusing on a subset of 27 female/55 male -enriched genes, this revealed a more prominent rewiring of regulatory regions associated with male genes than with female's.

3) METALoci was then used to predict the effects of deleting specific genomic regions. The authors successfully validated the approach to known regulatory regions of Sox9 and applied this to Fgf9, a gene very important for testis development but whose gene regulation is still not understood. They identified a region 250 kb downstream of Fgf9 with a potential regulatory role.

4) A mouse model with a deletion encompassing this region (d360) was generated and validated its putative regulatory role as it led to downregulation of Fgf9, upregulation of meiotic markers typical of ovarian tissues, and development of ovotestis and ovary-like structures rather than testes, as expected from a Fgf9 KO. Unlike the Fgf9 full KO, the d360 mutants survived postnatally, supporting the tissue-specific role of this regulatory region.

5) Analysis of mice carrying two smaller deletions (d93 and d104), revealed that d93 was more like the d306 phenotype morphologically and molecularly (impaired fg9 and sox9 expression), however some did develop testes implying that sex reversal was less efficient in this context. This suggested that the d93 region is not the minimal required region to support Fgf9 expression and testis development, but rather than the regulatory potential of Fgf9 is spread across multiple redundant enhancers across d306.

6) Gene regulatory networks were inferred by adapting a scRNA-Seq specific pipeline (SCENIC) to use output derived from the METALoci approach. An increase in the number of regulatory modules (regulons) after sex determination was identified, notably including several TF known to regulate the process.

7) Additionally, novel TF not previously associated with sex determination were identified namely MEIS 1 and 2, with a predicted role in both sexes. Validation analyses using transgenic mice demonstrated that these were indeed involved at least partially into cell fate regulation, perhaps more evidently in the testes than the ovary.

The paper is clearly presented, very well and concisely written. The figures are clear and well organized, although some of the extended ones could have bigger font text to aid reading. The methods were applied appropriately and rigorously. The validity and potential of the novel approaches developed in this study is appropriately discussed in the context of other methods published. The authors do an admirable job of integrating published data with their own, using their newly developed pipeline of metaloci to identify successfully gene regulatory networks in sex determination. They go beyond validating their approach on known regulators of sex determination to identify novel players which are tested and confirmed in vivo. For Fgf9, it is great to discover the first regulatory region of this gene in the context of gonadal development. The fact that this region was identified in silico means that many more potential regulatory regions could be identified this way before generating animal models, which is highly desirable to reduce animal usage. I believe the conclusions are well supported by the data, which are validated computationally and in vivo. The omics data generated will also be the source of several potential future studies, so are of extreme interest to the sex determination community.

Importantly, the application of this pipeline is going to be extremely useful also for other fields, where a plethora of published omics data is often available but still lacks a robust gene regulatory network analysis. The ability to identify successfully regulatory regions which are tissue-specific also offers the benefit of bypassing the embryonic lethality of a specific deletion, which would be an attractive model for many labs.

Therefore, I believe this paper will be of interest to a very broad readership beyond developmental biology, besides constituting on its own a powerful framework for the sex determination field and for the field of non-coding genome regulation in other biological systems.

Main comments

-The first part of the introduction could be expanded to incorporate and discuss latest finding regarding female sex determination namely the discovery of -KTS WT1 which supports an active process rather than a default one. Wt1 is also featured highly amongst the regulons reported by this study.

-Related to the above, it may be worth discussing also the fact that perhaps changes in granulosa cells may be better assessed by comparing E10.5 with a later timepoint E14.5 rather than E13.5, this may in part be responsible for the lesser degree of gene transitions observed compared to the male? Or it may be possible that a widespread rewiring of the chromatin may not be necessary to drive cell fate in the female context, and that rather the precise remodelling of specific loci may be more critical. It may be beyond the scope of this work but may be worth considering these aspects in the discussion as I think the view of female sex determination as default state is no longer valid given current knowledge.

-Can the authors expand on the the role Meis1 may have on granulosa cell identity? When looking at figure extended 23 is not immediately clear why the same genotype across the two sexes have not been compared. What happens when two copies of Meis1 are deleted in XX for example?

- Can comment on the focus on MEIS1/2 which seem to be relevant for both sexes. Why not focusing on something for example female-specific where much more is yet to be understood? Or is this plan for future work? Discussing and justifying the choice a bit more maybe providing examples of other factors female or male specific that could be important to investigate further would strengthen the rationale.

Minor comments

-Extended data Fig 11, text label is very small, that's the case for other images throughout the manuscript especially when GO analyses are reported. I know it's hard to show them properly, maybe separating some of them in 2 figures would increase readability.

-Line 433 a reference to the figure with Stra8 would help.

Reviewer #2 (Remarks on code availability):

The code seems to contain everything needed for reproducibility by other labs, and it is also well annotated. Please note I did not test it.

Reviewer #3 (Remarks to the Author):

Mota-Gómez et al. explore the 3D regulatory landscape of mammalian sex determination in vivo using high-resolution chromatin interaction maps of the mouse gonadal supporting lineage, before and after sex determination. They integrate Hi-C and epigenetic data using METALoci, a novel analytical method that applies a spatial autocorrelation framework within a chromatin interaction network, to quantify regulatory environments genome-wide, based on data from Hi-C and H3K27ac ChIP-seq experiments.

In summary, the results are convincing, the paper is generally well written, and the novel analytical methods are interesting and generally applicable. I recommend the paper for publication in Nature Structural & Molecular Biology, pending minor revisions.

The paper is interesting and offers a new perspective on sex-determining regulatory hubs. It also presents an innovative method for leveraging cooperative interactions from Hi-C data to detect functional states within regulatory hubs. The usage of interaction cooperativity between spatially related loci to enhance functional predictions of regulatory hubs seems useful and METALoci could be an easy-to-use approach to integrate Hi-C maps with a variety of functional signals for dissecting otherwise overlooked functional changes during differentiation. The authors did show prominent rewiring of regulatory hubs during sex determination. The genome-wide changes in regulatory hubs are convincing, including the curated set of female/male-development genes.

The authors discovered substantial changes in the 3D regulatory hub organization for hundreds of genes that display sex- and temporal specificity. As an example, the authors detect significant changes in the signal status of the Sox9 locus environment during Seratoli differentiation, confirming overall expectations. They also detected state transitions of the Bmp2 promoter from non-active (LL) to an active (HH) state in granulosa cells. Both these examples demonstrate the applicability of METALoci.

Also, the detection of Enh13 and TESCO enhancers through in silico perturbations in the METALoci analysis is interesting and validated by deletion experiments.

1) The authors claim that conventional Hi-C analysis suggested that chromatin structures remain largely unchanged during sex determination. However, they focus their analysis only at the level of TAD boundaries, and A/B compartments. However, A/B compartment detection may be dominated by long-range relationships, potentially overlooking subtle changes in shorter range regulatory interactions. A meaningful comparison might also involve using a loop detection algorithm to test for differences in the number of specific enhancer-promoter loops, or loops between promoters and H3K27ac peak regions, for a selected set of genes across the developmental stages. This could be done for the Sox9 locus or selected set of relevant genes.

A comparison may highlight genes whose regulatory rewiring may be detected by METALoci but may be missed by Hi-C-based promoter-enhancer loop analysis, as alternative enhancer interactions might not be captured by statistically significant promoter-enhancer pairs alone. This could enhance the relevance of the METALoci analysis.

The authors should at least add a discussion about the strength or weaknesses of METALoci in comparison to more traditional enhancer-promoter loop detections.

2) Since the Kamada-Kawai algorithm seeks only an approximate optimum, variations in arbitrary parameters could result in distinct yet equally plausible graph layouts. I assume the authors have explored the robustness of their approach. It may be useful to add a discussion about the robustness of the results with respect to alternative graph layouts. Or potentially indicate if some loci's LM's I predictions may be more affected by potential robustness issues than others.

3) Although some of the graphics are visually appealing, I had great problems to read the text and distinguish some of the color schemes. Partially also due to the relative low resolution in the embedded figures. But I assume the figures would be further scaled down in size in the published manuscript. Especially the distinction between a faded and deeper color to distinguish samples at different developmental time points during development (e.g., E10.5 and E13.5). Some of the fonts were ridiculously small. The authors should improve the readability of labels, figure details and make sure the color scheme used for the 4 samples is clearly distinguishable.

Version 1:

Decision Letter:

Our ref: NSMB-A50899A

19th Nov 2025

Dear Dr. Lupiáñez,

Thank you for submitting your revised manuscript "Sex-determining 3D regulatory hubs revealed by genome spatial auto-

correlation analysis" (NSMB-A50899A). It has now been seen by the original referees and their comments are below. The reviewers find that the paper has improved in revision, and therefore we can now accept it in principle in Nature Structural & Molecular Biology, pending minor revisions to satisfy the referees' final requests and to comply with our editorial and formatting guidelines.

We are now performing detailed checks on your paper and will send you a checklist detailing our editorial and formatting requirements in about 2 weeks. Please do not upload the final materials and make any revisions until you receive this additional information from us.

To facilitate our work at this stage, it is important that we have a copy of the main text as a word file. If you could please send along a word version of this file as soon as possible, we would greatly appreciate it; please make sure to copy the NSMB account (cc'ed above).

Sincerely,

Dimitris Typas
Senior Editor
Nature Structural & Molecular Biology
ORCID: 0000-0002-8737-1319

Reviewer #1 (Remarks to the Author):

The authors addressed my comments and the revised manuscript now more clearly presents the novel METALoci analysis pipeline as a powerful new tool to identify candidate regulatory regions of developmental genes, in this study compellingly applied to discover new regulatory elements and genes involved in male and female sex determination. The work is of high quality and broad relevance. I only have two final minor comments.

1. It is premature to invoke a functional role of 3D genome folding in sex determination. The authors have not formally demonstrated that genome folding is required for sex determination as opposed to simply reflecting gene regulation. The following sentences should therefore be toned down:
 - line 131: "guiding" (suggesting an active role) could for example be replaced with "informing on"
 - line 621: "affecting" could for example be replaced with "correlating with"
2. Reply to minor point 6: Thank you for the explanation but I think my question was misunderstood. It did not relate to how to interpret the scatter plots in Fig. 3a, but rather how to interpret the "Moran local I" y axis scale under the Hi-C plots in Fig. 4a. There is no comparable scale (from 0 to -1.5) in the scatter plot in Fig. 3a. I imagine that smaller values here somehow indicate a transition to a "HH" state (?). It would be great if the authors could clarify this in the figure legend.

Reviewer #2 (Remarks to the Author):

I appreciate the authors' thorough responses to the comments raised in the initial review. The revised manuscript demonstrates significant improvements in clarity and data presentation, with a more comprehensive discussion section. I am satisfied that the authors have adequately addressed all major and minor points I raised and I believe the manuscript is now suitable for publication in its current form.

Reviewer #2 (Remarks on code availability):

The code link seems to contain all necessary items including the README file. Code seems to be adequately annotated. Please note I have not tested the code as this is out of my expertise.

Reviewer #3 (Remarks to the Author):

The authors addressed all comments and I recommend the paper for publication.

Version 2:

Decision Letter:

14th Jan 2026

Dear Dr. Lupiáñez,

We are now happy to accept your revised paper "Chromatin spatial analysis by METALoci unveils sex-determining 3D regulatory hubs" for publication as an Article in Nature Structural & Molecular Biology.

Your paper will be published online soon after we receive proof corrections and will appear in print in the next available issue. You can find out your date of online publication by contacting the production team shortly after sending your proof corrections.

Authors may need to take specific actions to achieve compliance with funder and institutional open access mandates. If your research is supported by a funder that requires immediate open access (e.g. according to [Plan S principles](https://www.springernature.com/gp/open-science/plan-s-compliance) or the [NIH public access policy](https://www.springernature.com/gp/open-science/us-federal-agency-compliance)) then you should select the gold OA route, and we will direct you to the compliant route where possible. Because authors warrant under our subscription licensing terms that they haven't committed to licensing any version of their article under a licence inconsistent with the terms of our agreement – including the applicable embargo period – publication under the subscription model isn't suitable for authors whose funders require no embargo.

Sincerely,

Dimitris Typas
Senior Editor
Nature Structural & Molecular Biology
ORCID: 0000-0002-8737-1319

Sex-determining 3D regulatory hubs revealed by genome spatial auto-correlation analysis

Irene Mota-Gómez^{1,2}, Juan Antonio Rodríguez^{3*@}, Shannon Dupont⁴, Alicia Hurtado^{1,2}, Vanessa Cadenas⁵, Leo Zuber³, Iago Maceda³, Oscar Lao⁶, Johanna Jedamzick¹, Ralf Kuhn⁷, Scott Lacadie⁸, Sara Alexandra García-Moreno^{4\$}, Miguel Torres⁵, Francisca M. Real², Rafael D. Acemel^{1,2}, Blanche Capel^{4#}, Marc A. Marti-Renom^{3,9,10#}, and Darío G. Lupiáñez^{1,2#}*

Responses to reviewer's comments

Reviewer #1

This study identifies new genes and distal enhancers involved in mouse gonadal sex determination, aiming to improve our understanding of transcriptional regulation in this process and its relevance to human disorders. The authors generated Hi-C maps from FACS-purified bipotential progenitors (E10.5 XX and XY embryos) and differentiated Sertoli or granulosa cells (E13.5 embryos). Conventional Hi-C analyses showed minimal TAD-level changes (Fig. 1). To detect higher-order 3D genome changes, the authors developed METALoci, a computational tool combining Hi-C and H3K27ac ChIP-seq data to identify active 3D regulatory hubs ("metaloci") with coordinated H3K27ac gains during gonad differentiation. METALoci confirmed known distal enhancer-promoter interactions at Sox9 (Fig. 2). When applied to all genes, METALoci identified new genes that engage in active 3D hubs upon Sertoli or granulosa cell differentiation, expanding the list of candidate genes involved in sex determination (Fig. 3). Running METALoci on simulated genomic deletions uncovered a broad region containing new Fgf9 distal enhancers, validated in vivo via large deletions in mice that resulted in partial Fgf9 downregulation and sex reversal phenotypes (Fig. 4). Finally, the authors integrated METALoci analysis with the published SCENIC pipeline and published ATAC-seq and scRNA-seq datasets to identify transcription factors (TFs) enriched at metaloci. This analysis identified candidate TFs that may control gonad differentiation, including Meis1/2 whose deletion caused sex differentiation defects, uncovering a novel role for these TFs (Fig. 5).

Key strengths of this study include the development of METALoci, the validation of a new Fgf9 regulatory region required for sex determination (but not other Fgf9 functions), and the identification of new transcriptional regulators of sex determination like Meis1/2. The engineered mouse models showed convincing sex reversal phenotypes. The authors argue that their new computational approaches, which are outside the scope of my expertise, accelerated the discovery of relevant genes and regulatory elements compared to traditional genetic approaches. The findings are relevant to a broad audience interested in developmental gene regulation and sex determination.

We thank the reviewer for the positive comments on our manuscript and for highlighting the significance of the work.

Limitations are summarized in my major comments below:

1. It was not fully clear how METALoci compares to existing methods in pinpointing enhancers, such as performing differential analysis of H3K27ac ChIP-seq or ATAC-seq peaks between undifferentiated versus differentiated gonads. In the case of *Fgf9*, METALoci identified a region of hundreds of kb as containing a candidate enhancer, as opposed to precisely pinpointed candidate enhancer peaks.

Following the reviewer comments, we performed differential analyses of H3K27ac ChIP-seq comparing cells from undifferentiated and differentiated gonads. As this analysis does not rely on 3D chromatin interaction data, we assigned peaks to genes by establishing a distance threshold of +/- 500Kb from the transcription start site (TSS), which is a common strategy used in the field. Focusing on the same gene set shown in Figure 2 from the main text, we identify a higher number of differential peaks during Sertoli differentiation than during granulosa differentiation, even for female-biased genes (Reviewer Fig. 1). Specifically, we find a median of 20.76 peaks associated with male-biased and 19 associated with female-biased genes during Sertoli differentiation. In contrast, we observe a median of 7.76 peaks for male-biased and 9.56 for female-biased genes during granulosa differentiation. According to this analysis, genes would be associated with a higher number of enhancers during Sertoli than granulosa differentiation, independently if those genes acquire a female- or male-biased expression. As such, the assignment of peaks based on fixed genomic distances and without considering 3D chromatin interaction data results in enhancer-promoter associations that do not reflect known transcriptional dynamics during sex determination. Rather, these results just seem to simply reflect the fact that many more peaks are identified during Sertoli compared to granulosa differentiation at a genome-wide level (72,025 vs 27,103, respectively).

a

"Female" genes

b

"Male" genes

Reviewer Figure 1. Comparison between METALoci and a differential peak approach. *a.* Comparison of methods for genes displaying female-biased (a) or male-biased expression (b) during XX granulosa differentiation and XY Sertoli differentiation. METALoci analysis shows individual gene transitions for each female and male-specific genes during sex-determination. Genes are grouped by unsupervised clustering based on gene transitions from E10.5 to E13.5 in female and male differentiation (see also Main Fig.3). H3K27ac ChIP-seq and ATAC-seq analysis shows the number for differential peaks detected on a +/-500Kb window around the transcriptional start site (TSS) of each analyzed gene. Note that these two analyses detect a higher number of differential peaks for nearly all female-biased genes in the differentiation process of the opposite sex (that is XY Sertoli differentiation) compared to the same sex (XX granulosa differentiation). In contrast, METALoci detects a higher regulatory activity in XX granulosa compared to XY Sertoli differentiation for almost every female-biased genes, which is consistent with known transcriptional dynamics.

As an example, we show the analysis at the *Fgf9* locus, which yielded up to 37 potential putative enhancers upon Sertoli differentiation (Reviewer Fig. 2). Please note that the utilization of the fixed +/-500Kb threshold is also problematic, as it fails to identify the regulatory elements validated by our 104Kb deletion because this region is further away from the TSS. Similarly, this threshold also assigns 42 putative enhancers for *Sox9* but fails to identify the known *Enh13* as a candidate enhancer, as this element is located 800 Kb away from the TSS (data not shown). In addition, many genes with regulatory domains that are smaller than the size threshold yield associations with putative enhancers that are outside of their regulatory landscapes, thus rendering many false positives. Similar results were obtained when analyzing ATAC-seq data.

Reviewer Figure 2. ATAC and H3K27ac differential peak analysis during Sertoli cell differentiation. The figure displays the entire 4 Mb region surrounding the *Fgf9* gene (chr14:56,070,000–60,060,000), with the gene promoter positioned at the center of the visualized region. This genomic window corresponds to the one used in the METALoci analysis. However, the differential peak analysis focuses specifically on regions proximal to the promoter—defined here as ± 500 kb from the *Fgf9* transcription start site. The data show a marked increase in both ATAC and H3K27ac signal near the *Fgf9* locus, with additional peaks identified in Sertoli cells across 19 and 29 distinct 10 kb bins (indicated by green in the figure, respectively). To ensure a fair comparison, all analyses were conducted at a consistent resolution of 10 kb, matching the resolution used in the METALoci framework. Note that the fixed ± 500 kb threshold excludes the 104Kb region, which has been demonstrated to contain *Fgf9* enhancers.

In contrast, *METALoci* integrates 3D chromatin interaction data, helping to define the regulatory landscape of each studied gene. When these analyses are applied to the same gene set, *METALoci* identifies regulatory activation patterns that are specific of sex, and that recapitulate the observed transcriptional changes. More specifically, female genes gain regulatory activity during granulosa differentiation and male genes during Sertoli differentiation, as expected. These results demonstrate that integrating 3D chromatin interaction data helps to better capture the regulatory aspects of the studied biological process. It also avoids the arbitrariness of fixed-distance thresholds, which cannot account for the ample variability in regulatory domain sizes (which can range from few kilobases to Megabases, in extreme cases). It is also important to mention that *METALoci* provides more interpretable results than traditional strategies on Hi-C data analysis that focus on predetermined structures, such as loop calling (see response to Reviewer 3)

The advantage of using *METALoci* can be further highlighted by the identification of the known *Sox9* gonadal enhancers. Previously, Gonen *et al.* performed an extensive screening to identify such elements, by using a combination of ATAC-seq, DNase-seq and H3K27ac ChIP-seq data. Up to 33 putative enhancers were identified in the adjacent gene desert to *Sox9* and 16 of them were validated *in vivo*. Although this peak-based strategy was successful and led to the identification of Enh13 as a regulatory element required for *Sox9* expression, it also required the generation of numerous transgenic mice lines for enhancer screening. Such resources are not readily accessible for many research groups. In contrast, *METALoci* accurately identified two critical regions for *Sox9* regulation in its adjacent gene desert: the Enh13 and the TESCO enhancer. Therefore, the integration of H3K27ac and Hi-C datasets in our spatial autocorrelation analyses served to pinpoint the elements that have been demonstrated to be critical for *Sox9* regulation. Applied to other loci or developmental systems, *METALoci* may simplify enormously the identification of candidate enhancers for further functional validation.

At the *Fgf9* locus, we observe that its downstream adjacent >1Mb gene desert is rich in putative enhancers: H3K27ac ChIP-seq and ATAC-seq data reveal around 20 candidate regions that could be potential enhancer elements (see Figure 4 from main text). The results derived from *METALoci* guided us to a 306 Kb subregion of this large gene desert. The deletion of this region in transgenic mice led to significant reduction in *Fgf9* expression and sex reversal, suggesting that critical regulatory elements lie within this region. Further, mice carrying deletions of the 94 Kb and 104 Kb subregions exhibit sex reversal phenotypes of reduced severity, confirming the presence of redundant enhancers distributed across the 306 Kb region. Gene regulation by redundant enhancers is rather common for many developmental genes (*Hoxd* genes in Andrey et al. *Science* 2013; alpha-globin in Hay et al., *Nat Genet.* 2016), including *Fgf8*, a paralogous gene to *Fgf9* whose limb expression is also controlled by several enhancers distributed over large genomic regions (Hörnblad et al, *Nat Commun.* 2021). This supports the idea that the large regulatory domains predicted by *METALoci* at the *Fgf9* locus reflect the existence of enhancer redundancy, rather than resulting from a lack of precision of the method. Of course, once a critical regulatory region has been identified, it is relatively straight-forward to assign activities to the individual ATAC-seq/H3K27ac ChIP-seq peaks contained within (see also the response to the minor comment #5).

Overall, we would like to note that *METALoci* is not specifically focused on pinpointing individual enhancer peaks but rather to define regulatory landscapes based on spatial epigenomic organization. As such, our approach offers a biologically meaningful framework that complements traditional peak-based analyses and enables targeted functional studies with greater efficiency. This aspect allows us to investigate the regulation of sex determination under a new perspective. It is also important to note that, despite the development of various methods aimed at globally identifying correlations between epigenetics and 3D genome structure, *METALoci* remains, to our knowledge, the only approach specifically designed to locally identify regulatory landscapes surrounding a gene of interest. Consequently, benchmarking *METALoci* against similar methods is not straightforward, and its performance can only be meaningfully evaluated against experimental data, as demonstrated in our manuscript.

2. It was not fully clear which H3K27ac ChIP-seq datasets were used for *METALoci* analysis. It seems essential for the analysis that the ChIP-seq datasets were generated at the same developmental time points and cell types analyzed here by Hi-C, but this was not explicitly stated and should be clarified.

We apologize for the lack of clarity in this matter. H3K27ac ChIP-seq data were obtained from Garcia-Moreno. et al. *Dev Biol*, 2019. In this manuscript, the progenitor supporting cells from both sexes at E10.5 were obtained from the same Sf1-eGFP line that we employ in our study and at the same developmental stages. In the case of XY E13.5 Sertoli cells, H3K27ac ChIP-seq data was obtained by using a TESCO-CFP line, while here we employed a Sox9-eGFP. Both lines have been widely used in the literature to isolate Sertoli cells, and our cell collections were obtained at the same developmental timepoints. In the case

of XX 13.5 granulosa cells, H3K27ac data was obtained from a Runx1-eGFP line. Similarly, both lines have been used to isolate granulosa cells, and our cell collections were obtained at the same developmental timepoints. We have added this relevant information in the Material and Methods section corresponding to the CHIP-seq analysis.

Minor comments:

1. Lines 188-192: The text can appear contradictory. “Metaplot analysis revealed that insulation at boundary regions increased during the transition from bipotential to the differentiated stage” suggests a global effect on TAD boundaries. “Pairwise comparisons revealed that only 1.49-1.84% of TAD boundaries changed their insulation significantly” instead describes a negligible effect that should not be visible in a metaplot analysis.

These apparently contradictory findings correspond to a generalized gain of insulation across most boundaries in the transition from bipotential to differentiated. However, such changes are most of the times of low magnitude and do not pass a statistical test when individual boundaries are compared between samples. Therefore, although there is a general increase in insulation upon differentiation, sharp changes at individual loci or the appearance/disappearance of boundaries are rare events.

We have modified the text accordingly (page 8, lines 194-195) to better explain these apparently counterintuitive findings.

“Yet, this increase in insulation is of low magnitude for most regions, as pairwise comparisons revealed that only 1.49-1.84% of TAD boundaries changed their insulation significantly between sexes or timepoints (Fig. 1g).”

2. Line 218: I was not convinced that METALoci “quantif(ies) gene regulation”.

We have reworded the sentence to state that we employed *METALoci* to “infer gene regulatory activity”, which we believe may better describe the purpose of the analysis (page 9, line 223).

“We repurposed this analysis to infer gene regulatory activity, as CREs and genes cluster in the 3D nuclear space while displaying similar epigenetic properties.”

3. Paragraph starting at line 333: There is another apparent contradiction in the text: “During the differentiation from bipotential to granulosa cells, most female-specific genes acquired an active regulatory environment (...)” This sentence seems to contradict the following one: “The majority of male genes (46 out of 55) gain an active regulatory environment during Sertoli cell differentiation. In contrast, this mechanism was not as common for female-specific genes during granulosa cell differentiation (14 out of 27)”. 14 out of 27 is not accurately described as “most female-specific genes”.

We apologize for the lack of clarity in this section. We have modified the text accordingly (page 14, lines 348-352).

“During the differentiation from bipotential to granulosa cells, many female-specific genes acquired an active regulatory environment, while their male-specific counterparts either lost it or displayed minor changes (Extended Data Fig. 12). The opposite trend was observed during the differentiation of Sertoli cells, with the acquisition of active regulatory activity for many male-specific genes and a general loss in their female-specific counterparts (Extended Data Fig. 12).”

4. Line 427: Explain that mice were derived from the engineered ESCs, for better flow in the text.

We have modified the text accordingly (page 17, line 438-439).

5. Line 473: Could the authors narrow down candidate *Fgf9* enhancers by differential analysis of H3K27ac ChIP-seq peaks between Sertoli and granulosa cells?

Following the reviewer’s suggestion, we performed a differential peak analysis for both ATAC and H3K27ac signals within ± 1 Mb of the *Fgf9* locus, comparing granulosa and Sertoli cells (i.e., XX and XY samples at embryonic day 13.5). The analysis revealed a gain of multiple peaks in males relative to females, specifically 58 new H3K27ac peaks and 36 new ATAC peaks (see Reviewer Figure 3). These findings highlight the strength of the *METALoci* approach in prioritizing, from among many peaks, those most likely to influence the expression of a target gene. Without this prioritization, CRISPR-based functional validation would have required dozens of individual experiments. Instead, *METALoci* effectively narrowed the focus to a small subset of key regulatory regions.

When focusing on the 306Kb region identified by *METALoci* the analysis revealed up to 13 differential peaks within as relevant for *Fgf9* gonadal regulation. Most of these peaks (8 out of 13) were also identified when performing a differential ATAC-seq analysis. These analyses are in line with our observations from transgenic mice experiments, supporting the existence of functional redundancy between several enhancer elements. We have included this new figure in the manuscript (Extended Data Fig. 26) and referenced it in the main text (page 26, line 694-695)

Reviewer Figure 3. H3K27ac ChIP-seq and ATAC-seq differential peak analysis between Sertoli and granulosa cells at embryonic day 13.5 (E13.5). The figure displays the entire 4 Mb region surrounding the *Fgf9* gene (chr14:56,070,000–60,060,000), with the gene promoter positioned at the center of the visualized region. This genomic window corresponds to the one used in the METALoci analysis. However, the differential peak analysis focuses specifically on regions that locate ± 1 Mb from the *Fgf9* transcription start site. Unique peaks in Sertoli cells are marked with green rectangles. Asterisks indicate 8 unique peaks overlapping between H3K27ac ChIP-seq and ATAC-seq data signal, within the 306 Kb region that was deleted in transgenic mice.

6. Fig. 4A: The reader would benefit from more extensive explanations on how to interpret the METALoci analysis output. What do yellow and blue lines under the Hi-C maps represent? How should the values of LM's I be interpreted?

We are thankful to the reviewer for bringing our attention to this detail. We have provided additional explanations in the yellow/blue tracks, as wells for LM's I value interpretations in the figure legend.

“The LM scatter plot can be interpreted as the correlation between local signal at the node (x-axis) of interest and that of the neighborhood of the node (y-axis). For examples, if a genomic bin is high in H3K27ac and its neighborhood is also high, the node will be placed in the HH quadrant and colored red.”

7. Fig. 4A: Please show H3K27ac ChIP-seq data for E10.5.

We have modified the figure accordingly. Specifically, we have added the H327ac ChIP-seq XY 10.5 track in substitution of the previous ATAC XY E13.5, since this one is shown also in the new Extended Data Fig. 26.

8. Line 495: The reader would benefit from a clearer conceptual explanation (potentially a schematic) of how METALoci and SCENIC pipelines were integrated for the analysis.

Following the reviewer's suggestion, we have created a new Extended Data Figure containing a schematic of the integration between both approaches (Extended Data Fig. 21). We have also made the corresponding reference in the main text (page 19, lines 516-517).

9. Line 496: Please specify what kind of cells were analyzed in the published scRNA-seq dataset.

We have modified the main text accordingly to include this information (page 19, lines 510-513).

“We used previously published scRNA-seq datasets to extract single-cell transcriptional information from the four studied cell types (XY and XX E10.5 Nr5a1-expressing cells; XY E13.5 Sox9-expressing cells; and XX E13.5 Runx1-expressing cells).”

10. Fig. 5B: please label the cell types above the plots and provide a color legend for red/blue color coding.

The figure has been modified accordingly.

11. A general comment is that the main text would benefit from proofreading to correct minor mistakes throughout.

We have reviewed the manuscript to streamline it and to remove inconsistencies or minor mistakes.

Reviewer #2

In this very interesting paper, the authors performed a comprehensive analysis of the changes in 3D chromatin structure and chromatin interactions characterizing the supporting cell lineage of the mouse gonads of both sexes, across the window of sex determination. Using a combination of innovative epigenomics techniques, novel in-silico approaches, mouse transgenesis to validate their findings and analysis of multiple deposited datasets beyond those produced within this study, the paper illustrates the following results:

1) When comparing the two critical timepoints of E10.5 (before sex determination) and E13.5 (after) between the female (XX) and male (XY) sexes by Hi-C, TAD structure is quite conserved across this developmental window, indicating that the conformation of the chromatin in the gonads is established before sex determination and maintained afterwards. Despite this stability in overall TAD organisation, analysis of switches between compartments (with A enriched for euchromatic active regions and B representing the heterochromatic inactive regions), revealed that a number of genomic regions switch compartments after sex determination, even though this does not result into changes in gene expression, but only in a changed epigenetic status.

2) A new analysis tool was developed called METALoci. This method allowed the authors to use Hi-C data to build maps of interactions within specific loci and then map epigenomics data namely ChIP-Seq for histone marks onto these maps to visually assess recruitment of regulatory regions to a specific gene promoter. A nice example which validates this method is provided for the Enh13 and TESCO enhancers of Sox9, and for BMP2 for the female pathway amongst others. The tool was then applied to each gene of the mouse genome showing a similar level or rewiring of regulatory regions in both sexes (Extended Figure 10 and Figure 3). However, when focusing on a subset of 27 female/55 male -enriched genes, this revealed a more prominent rewiring of regulatory regions associated with male genes than with female's.

3) METALoci was then used to predict the effects of deleting specific genomic regions. The authors successfully validated the approach to known regulatory regions of Sox9 and applied this to Fgf9, a gene very important for testis development but whose gene regulation is still not understood. They identified a region 250 kb downstream of Fgf9 with a potential regulatory role.

4) A mouse model with a deletion encompassing this region (d360) was generated and validated its putative regulatory role as it led to downregulation of Fgf9, upregulation of meiotic markers typical of ovarian tissues, and development of ovotestis and ovary-like structures rather than testes, as expected from a Fgf9 KO. Unlike the Fgf9 full KO, the d360 mutants survived postnatally, supporting the tissue-specific role of this regulatory region.

5) Analysis of mice carrying two smaller deletions (d93 and d104), revealed that d93 was more like the d306 phenotype morphologically and molecularly (impaired fg9 and sox9 expression), however some did develop testes implying that sex reversal was less efficient in this context. This suggested that the d93 region is not the minimal required region to support Fgf9 expression and testis development, but rather than the regulatory potential of Fgf9 is spread across multiple redundant enhancers across d306.

6) Gene regulatory networks were inferred by adapting a scRNA-Seq specific pipeline (SCENIC) to use output derived from the METALoci approach. An increase in the number of regulatory modules (regulons) after sex determination was identified, notably including several TF known to regulate the process.

7) Additionally, novel TF not previously associated with sex determination were identified namely MEIS 1 and 2, with a predicted role in both sexes. Validation analyses using transgenic mice demonstrated that these were indeed involved at least partially into cell fate regulation, perhaps more evidently in the testes than the ovary.

The paper is clearly presented, very well and concisely written. The figures are clear and well organized, although some of the extended ones could have bigger font text to aid reading. The methods were applied appropriately and rigorously.

The validity and potential of the novel approaches developed in this study is appropriately discussed in the context of other methods published. The authors do an admirable job of integrating published data with their own, using their newly developed pipeline of metaloci to identify successfully gene regulatory networks in sex determination. They go beyond validating their approach on known regulators of sex determination to identify novel players which are tested and confirmed in vivo. For Fgf9, it is great to discover the first regulatory region of this gene in the context of gonadal development. The fact that this region was identified in silico means that many more potential regulatory regions could be identified this way before generating animal models, which is highly desirable to reduce animal usage. I believe the conclusions are well supported by the data, which are validated computationally and in vivo. The omics data generated will also be the source of several potential future studies, so are of extreme interest to the sex determination community.

Importantly, the application of this pipeline is going to be extremely useful also for other fields, where a plethora of published omics data is often available but still lacks a robust gene regulatory network analysis. The ability to identify successfully regulatory regions which are tissue-specific also offers the benefit of bypassing the embryonic lethality of a specific deletion, which would be an attractive model for many labs.

Therefore, I believe this paper will be of interest to a very broad readership beyond developmental biology, besides constituting on its own a powerful framework for the sex determination field and for the field of non-coding genome regulation in other biological systems.

We are grateful to the reviewer for the positive remarks and for acknowledging the contribution and relevance of our study.

Main comments

-The first part of the introduction could be expanded to incorporate and discuss latest finding regarding female sex determination namely the discovery of -KTS WT1 which supports an

active process rather than a default one. Wt1 is also featured highly amongst the regulons reported by this study.

We completely agree with the reviewer in that the introduction should be expanded to include these important findings. We have revised the Introduction to clearly acknowledge the role of the –KTS isoform of WT1 as a female-determining factor (page 3, lines 72-74).

“In XX individuals, which lack Sry, ovarian development is actively driven by the expression of the ovarian-determining factor Wilms Tumor suppressor (WT1) -KTS isoform, as well as of several members of the WNT pathway, ...”

We also appreciate the reviewer’s observation that Wt1 appears as a top female regulon in our analysis. This point is now explicitly mentioned in the Results section (page 25, lines 647-650).

“Similarly, although the scRNA-seq data does not allow to discriminate between isoforms, Wt1 also appears among the top regulons in female-specific networks, which is consistent with the ovarian-determining function of its -KTS variant.”

-Related to the above, it may be worth discussing also the fact that perhaps changes in granulosa cells may be better assessed by comparing E10.5 with a later timepoint E14.5 rather than E13.5, this may in part be responsible for the lesser degree of gene transitions observed compared to the male? Or it may be possible that a widespread rewiring of the chromatin may not be necessary to drive cell fate in the female context, and that rather the precise remodelling of specific loci may be more critical. It may be beyond the scope of this work but may be worth considering these aspects in the discussion as I think the view of female sex determination as default state is no longer valid given current knowledge.

We fully agree with the reviewer in that it is plausible that granulosa cells may experience further relevant changes at later stages. For example, it is known that granulosa undergo a squamous-to-cuboidal transition at postnatal stages. We have expanded the discussion to acknowledge such possibility (page 25, lines 641-644).

“Yet, it is also known that granulosa cells undergo a squamous-to-cuboidal transition upon ovarian follicle activation, at prenatal stages. Thus, it is plausible that active changes in regulation may become more obvious at later timepoints, beyond E13.5.”

We also thank the reviewer for pointing out that the wording in some sections of the text may suggest the old idea that female sex determination is a default state. We have carefully revised the text and remove such instances.

-Can the authors expand on the the role Meis1 may have on granulosa cell identity? When looking at figure extended 23 is not immediately clear why the same genotype across the two sexes have not been compared. What happens when two copies of Meis1 are deleted in XX for example?

Regarding the immunofluorescence experiments displayed in Extended Data Fig. 24, which are resulting from the breeding of floxed *Meis1* and *Meis2* mice, we have now included data for the missing genotype (XX *Meis1*^{hom}; *Meis2*^{het}). In this mutant, we observe the same effect as in the other mutant with 3 inactive alleles (XX *Meis1*^{het}; *Meis2*^{hom}). Specifically, we observe that male differentiation is triggered on a limited number of XX supporting cells, as denoted by SOX9 expression. These new analyses support our previous observations that the *Meis1* and *Meis2* are functionally redundant during sex determination, in both sexes.

Please note that *Meis1*^{hom} embryos could not be obtained with our breeding strategy, as we had to employ a complex *Cre-lox* strategy involving multiple transgenic lines. This strategy combined a simultaneous maternal and paternal deletion of *Meis1flox/flox* and *Meis2flox/flox* alleles, using the maternal *Zp3Cre* and the paternal *Stra8Cre*. This complex approach was necessary to circumvent the pleiotropic effects of *Meis* genes, which often lead to early embryonic lethality and have historically limited functional studies on these factors. Unfortunately, the large number of mutations can yield a broad range of allele combinations, resulting in major difficulties to obtain embryos with specific genotypes. This problem is further exacerbated by the fact that female and male embryos are analyzed separately (effectively dividing the probability to obtain a particular genotype by half). Therefore, we focused on a mating scheme that maximized the possibilities of obtaining the genotypes that were essential to investigate *Meis* paralog redundancy. That is double heterozygotes, triple allele knockouts in different combinations, and quadruple knockouts (although these last ones did not progress to relevant stages). Thus, our mating strategy did not yield female embryos that are homozygous for *Meis1*.

Please also note that our initial experiments on *Meis1* homozygous inactivation (Figure 5C, second image) were conducted by generating homozygous XY mESCs, followed by the derivation of mutant mice using tetraploid aggregation. Since these stem cells are XY, our investigation was limited to embryos of this sex, which already exhibited signs of altered differentiation in a subset of cells. We could not perform similar experiments because XX mESCs perform poorly in tetraploid aggregation protocols, as they experience aberrant X chromosome inactivation in culture.

Nevertheless, we were able to analyze double heterozygous female embryos derived from our *Cre-lox* breeding strategy. Based on the data obtained in their male counterparts (where double heterozygotes phenocopy the full *Meis1* knockout), as well as existing literature demonstrating the functional overlap among *Meis* genes, we would predict similar effects on *Meis1* KO in females as in double heterozygous. That is, we would expect that SOX9 positive cells are only observed in female mutant gonads upon inactivation of 3 *Meis* alleles, suggesting that this phenotype may be less severe as in males.

Nevertheless, the effect of *Meis1* inactivation in females may need to be confirmed experimentally, which we are planning to do in the future together while addressing other open questions regarding the role of *Meis* genes in sex determination. For instance, an interesting observation is that triple allele knockout embryos exhibited gonadal growth defects, suggesting that *Meis* genes may play distinct roles in gonadal organogenesis vs differentiation (as has been demonstrated for other factors like *Sf1*, see Ikeda *et al. Sci Rep* 2021). We are

currently exploring this possibility by using conditional Cre lines for temporal and tissue-specific deletions. Additionally, we are performing ChIP-seq experiments to validate direct targets of *Meis* (which is relevant to explore a potential repressive function, as mentioned before) and to explore the degree of overlapping on chromatin regulation between the two paralogs. While these questions are important and actively being pursued by the Lupiáñez group, we feel that they fall beyond the scope of the current manuscript that is centered on the 3D regulatory dynamics of sex determination and on *METALoci*. In any case, we believe that the current data included in the manuscript support our claims. That is the first evidence for an involvement of *Meis* genes in sex determination and the redundant function that certain factors may have in this process. This concept is particularly relevant for our field, in the search for novel genes.

We have included the new data from the XX *Meis1^{hom}*; *Meis2^{het}* embryos in Extended Data Fig. 24, which we have restructured for a more direct comparison between genotypes. Accordingly, the results from XY *Meis1^{hom}* embryos obtained from tetraploid aggregation have been removed from this Extended Data Fig. and kept in Figure 5C from the main text.

- Can comment on the focus on MEIS1/2 which seem to be relevant for both sexes. Why not focusing on something for example female-specific where much more is yet to be understood? Or is this plan for future work? Discussing and justifying the choice a bit more maybe providing examples of other factors female or male specific that could be important to investigate further would strengthen the rationale.

As mentioned above, we were particularly intrigued by the fact that *Meis1* was listed among the top list of negative regulators in our gene regulatory network analysis, in both sexes. Traditionally, emphasis has been put on studying genes that can induce cell differentiation, but not so much on factors that can repress those processes. The possibility that *Meis1* could be performing a repressive role during female and male differentiation was intriguing and prompted us to explore this aspect further. In fact, our results on *Meis* inactivation raise the possibility that these factors could be repressing the alternative sex pathway, although such mechanisms would need to be characterized in detail (which we are planning to perform in future works).

Nevertheless, *METALoci* identified several other factors that have not been yet directly involved in sex determination. Some of them are indeed associated with female differentiation and, as the reviewer mentioned, there is much knowledge that can be gained from their detailed study. The Lupiáñez lab is currently generating mouse models from these candidates, which may have a potential role in sex determination.

As per reviewer's suggestion, we have expanded the text to better explain our rationale to focus on the study of *Meis* genes (page 20, lines 533-535).

"In particular, the identification of negative regulons may reveal repressors of lineage differentiation, a class of factors that have been traditionally less studied than activators."

Minor comments

-Extended data Fig 11, text label is very small, that's the case for other images throughout the manuscript especially when GO analyses are reported. I know it's hard to show them properly, maybe separating some of them in 2 figures would increase readability.

We have revised all figures of the manuscript, including the mentioned one, to increase text font and improve readability.

-Line 433 a reference to the figure with Stra8 would help.

A reference to the figure has now been added.

Reviewer #3:

Mota-Gómez et al. explore the 3D regulatory landscape of mammalian sex determination in vivo using high-resolution chromatin interaction maps of the mouse gonadal supporting lineage, before and after sex determination. They integrate Hi-C and epigenetic data using METALoci, a novel analytical method that applies a spatial autocorrelation framework within a chromatin interaction network, to quantify regulatory environments genome-wide, based on data from Hi-C and H3K27ac ChIP-seq experiments.

In summary, the results are convincing, the paper is generally well written, and the novel analytical methods are interesting and generally applicable. I recommend the paper for publication in Nature Structural & Molecular Biology, pending minor revisions.

The paper is interesting and offers a new perspective on sex-determining regulatory hubs. It also presents an innovative method for leveraging cooperative interactions from Hi-C data to detect functional states within regulatory hubs. The usage of interaction cooperativity between spatially related loci to enhance functional predictions of regulatory hubs seems useful and METALoci could be an easy-to-use approach to integrate Hi-C maps with a variety of functional signals for dissecting otherwise overlooked functional changes during differentiation. The authors did show prominent rewiring of regulatory hubs during sex determination. The genome-wide changes in regulatory hubs are convincing, including the curated set of female/male-development genes.

The authors discovered substantial changes in the 3D regulatory hub organization for hundreds of genes that display sex- and temporal specificity. As an example, the authors detect significant changes in the signal status of the Sox9 locus environment during Seratoli differentiation, confirming overall expectations. They also detected state transitions of the Bmp2 promoter from non-active (LL) to an active (HH) state in granulosa cells. Both these examples demonstrate the applicability of METALoci.

Also, the detection of Enh13 and TESCO enhancers through in silico perturbations in the METALoci analysis is interesting and validated by deletion experiments.

We appreciate the reviewer's positive feedback on our manuscript and the recognition of the importance of our work.

1) The authors claim that conventional Hi-C analysis suggested that chromatin structures remain largely unchanged during sex determination. However, they focus their analysis only at the level of TAD boundaries, and A/B compartments. However, A/B compartment detection may be dominated by long-range relationships, potentially overlooking subtle changes in shorter range regulatory interactions. A meaningful comparison might also involve using a loop detection algorithm to test for differences in the number of specific enhancer-promoter loops, or loops between promoters and H3K27ac peak regions, for a selected set of genes across the developmental stages. This could be done for the Sox9 locus or selected set of relevant genes.

A comparison may highlight genes whose regulatory rewiring may be detected by METALoci but may be missed by Hi-C–based promoter-enhancer loop analysis, as alternative enhancer interactions might not be captured by statistically significant promoter-enhancer pairs alone. This could enhance the relevance of the METALoci analysis.

The authors should at least add a discussion about the strength or weaknesses of METALoci in comparison to more traditional enhancer-promoter loop detections.

Following the reviewer comments, we have performed analyses on promoter-enhancer loops. We again focused on the same set of female and male genes analyzed in Main Figure 3 from the main text. To identify putative enhancers, we detected differential H3K27ac ChIP-seq peaks anchoring a Hi-C chromatin loops that has the other anchor at a gene promoter. This combined analysis only retrieved a very reduced number of putative enhancer-promoter loops, without any meaningful output for most of the studied genes (Reviewer Figure 4). In particular, the analysis could not identify any “looped” enhancers at the *Fgf9* locus, contrasting with the predictions from *METALoci*, which identified the critical region containing those. The analysis, however, identified H3K27ac peaks at the bases of anchor loops involving the *Sox9* promoter, including the well-characterized *Enh13*, during Sertoli differentiation.

Reviewer Figure 4. Comparison between METALoci and a chromatin loop-based approach. a. Comparison of methods for genes displaying female-biased (a) or male-biased expression (b) during XX granulosa differentiation and XY Sertoli differentiation. METALoci analysis shows individual gene transitions for each female and male-specific genes during sex-determination. Genes are grouped by unsupervised clustering based on gene transitions from E10.5 to E13.5 in female and male differentiation (see also Main Fig.3). H3K27Ac ChIP-seq

analysis shows the number for differential peaks that overlap with a chromatin loop that has a gene promoter in the other anchor. Loop analyses were performed with different algorithms, and we proceeded with the results from MUSTACHE (0.05 FDR; Roayaei Ardakany et al, Genome Biol., 2020), which identified a higher number of chromatin loops. Note that the loop approach does not detect any differentially “looped” H3K27Ac ChIP-seq peak for most of the studied genes.

Our results underscore the relevance of *METALoci* in identifying putative enhancers, as demonstrated in the *Sox9* locus. While similar findings can, to a certain extent, be reproduced by identifying chromatin loops, this approach presents challenges as loop-calling is very dependent on the method used to detect loops. Moreover, those approaches rely on the assumption that enhancer-promoter interactions are rather stable and highly frequent, a notion that has been challenged by emerging models of gene regulation (see Karr et al., *Genes & Dev*, 2022; or Bickmore, *Nat Rev Genet.*, 2025) As such, *METALoci* adds value in a large number of loci such as *Fgf9*, where no loops are detected between the promoter (or nearby regions) and enhancers, better capturing the regulatory dynamics underlying sex differentiation. Overall, the output retrieved from *METALoci* agrees with those alternative models of enhancer-promoter communication. As suggested by the reviewer, we have added this figure to the manuscript (Extended Data Fig. 25) and a paragraph in the discussion about the potential advantages of employing *METALoci* (pages 25-26, lines 670-683)

“Strategies for enhancer identification based on chromatin interaction data generally rely on the detection of chromatin loops, which can be challenging as loop-calling is very dependent on the analytical method used. More importantly, these approaches work under the assumption that enhancer-promoter interactions are relatively stable and highly frequent. However, emerging models of gene regulation have challenged this notion, suggesting that productive enhancer-promoter communication may rely on transient interactions or even physical proximity, rather than direct contact. When applied to our datasets, an enhancer detection strategy combining H3K27ac ChIP-seq and chromatin loop information proved largely uninformative for most genes exhibiting sex-biased expression, including Fgf9 (Extended Data Fig. 25). In contrast, METALoci, which does not rely on the identification of predefined structures like chromatin loops, captured the regulatory dynamics underlying sex differentiation most effectively, thus supporting such alternative models of enhancer-promoter communication.”

2) Since the Kamada-Kawai algorithm seeks only an approximate optimum, variations in arbitrary parameters could result in distinct yet equally plausible graph layouts. I assume the authors have explored the robustness of their approach. It may be useful to add a discussion about the robustness of the results with respect to alternative graph layouts. Or potentially indicate if some loci’s LM’s I predictions may be more affected by potential robustness issues than others.

The Kamada-Kawai layout algorithm is a force-directed graph drawing method that positions nodes such that their geometric (Euclidean) distances are proportional to their graph-theoretic distances. Given an input graph, a set of initial positions, and a deterministic numerical solver, the algorithm will consistently produce the same layout in a deterministic manner. However, the reviewer is correct in noting that *METALoci* parameters can affect the final layout produced by the Kamada-Kawai algorithm. Indeed, the potential impact of

parameters in *METALoci* is not directly on the functioning of the Kamada-Kawai algorithm itself, but rather on the construction of the initial graph. Specifically, *METALoci* includes two parameters that can influence the final layout. The first is the selection of interactions from the Hi-C input matrix. The second is the force applied to consecutive nodes in the graph, which ensures both connectivity and, to some extent, bendability (approximation of the persistence length of the polymer).

Currently, *METALoci* allows users to specify these parameters in two ways: through command-line inputs (e.g., `--cutoff` to set the fraction of top Hi-C interactions to use, and `--pl` to define the “persistence length” or connectivity force), or via an optimization procedure that identifies the optimal parameters based on the input Hi-C matrix. As suggested by the reviewer, we have clarified this point in the Methods section of the manuscript and have added a paragraph discussing the limitations of the approach (page 34-35, lines 914-930).

3) Although some of the graphics are visually appealing, I had great problems to read the text and distinguish some of the color schemes. Partially also due to the relative low resolution in the embedded figures. But I assume the figures would be further scaled down in size in the published manuscript. Especially the distinction between a faded and deeper color to distinguish samples at different developmental time points during development (e.g., E10.5 and E13.5). Some of the fonts were ridiculously small. The authors should improve the readability of labels, figure details and make sure the color scheme used for the 4 samples is clearly distinguishable.

We have revised the figures and others to fix the issues raised by the reviewer, including font sizes and color schemes.

Sex-determining 3D regulatory hubs revealed by genome spatial auto-correlation analysis

Irene Mota-Gómez^{1,2}, Juan Antonio Rodríguez^{3*@}, Shannon Dupont⁴, Alicia Hurtado^{1,2}, Vanessa Cadenas⁵, Leo Zuber³, Iago Maceda³, Oscar Lao⁶, Johanna Jedamzick¹, Ralf Kuhn⁷, Scott Lacadie⁸, Sara Alexandra García-Moreno^{4\$}, Miguel Torres⁵, Francisca M. Real², Rafael D. Acemel^{1,2}, Blanche Capel^{4#}, Marc A. Marti-Renom^{3,9,10#}, and Darío G. Lupiáñez^{1,2#}*

Responses to reviewer's comments

Reviewer #1

The authors addressed my comments and the revised manuscript now more clearly presents the novel METALoci analysis pipeline as a powerful new tool to identify candidate regulatory regions of developmental genes, in this study compellingly applied to discover new regulatory elements and genes involved in male and female sex determination. The work is of high quality and broad relevance. I only have two final minor comments.

We thank the reviewer for the positive comments on our manuscript and for highlighting the significance of the work.

1. It is premature to invoke a functional role of 3D genome folding in sex determination. The authors have not formally demonstrated that genome folding is required for sex determination as opposed to simply reflecting gene regulation. The following sentences should therefore be toned down:
- line 131: "guiding" (suggesting an active role) could for example be replaced with "informing on"
- line 621: "affecting" could for example be replaced with "correlating with"

We have toned down the statements in the main text, as suggested by the reviewer.

2. Reply to minor point 6: Thank you for the explanation but I think my question was misunderstood. It did not relate to how to interpret the scatter plots in Fig. 3a, but rather how to interpret the "Moran local I" y axis scale under the Hi-C plots in Fig. 4a. There is no comparable scale (from 0 to -1.5) in the scatter plot in Fig. 3a. I imagine that smaller values here somehow indicate a transition to a "HH" state (?). It would be great if the authors could clarify this in the figure legend.

The reviewer makes a point about clarifying the MLI values in Figure 4a. These are not the values for Moran Local I (LMI) for each locus in the "scanned" region. To the

contrary, those correspond to the MLI value of the *Fgf9* containing bin once the bin the x-axis has been removed from the layout. Therefore, the distribution of the values does not need to match that of the genome-wide values of LMI. We have modified the figure legend for the panel 4a to reflect this better.

a. *Upper panel. Predictive scanning analysis at the Fgf9 locus. Hi-C and ChIP-seq tracks are displayed for XY E10.5 and XY E13.5. Each dot in the scatter plot indicates the value of the Moran local I for Fgf9 after the deletion of a particular bin set in the scanned region. Vertical blue lines mark regions whose deletion is predicted to decrease the LM's I of the Fgf9 metaloci below one standard deviation (below grey area, see Methods). Red transparent shape indicates the region containing most regulatory potential within the Fgf9 TAD. Lower panel. Zoom in of the Fgf9 TAD region. ATAC-seq and H3K27 ChIP-seq tracks are shown for Sertoli E13.5 XY E10.5 and E13.5 cells (Sertoli). Note the abundance of ATAC peaks across the adjacent gene desert to Fgf9. Deleted regions are indicated below.*

Reviewer #2

I appreciate the authors' thorough responses to the comments raised in the initial review. The revised manuscript demonstrates significant improvements in clarity and data presentation, with a more comprehensive discussion section. I am satisfied that the authors have adequately addressed all major and minor points I raised and I believe the manuscript is now suitable for publication in its current form.

We are grateful to the reviewer for the positive remarks.

Reviewer #3:

The authors addressed all comments and I recommend the paper for publication.

We appreciate the reviewer's positive feedback on our manuscript.